# Development of Aptamer-DNAzyme based metal-nucleic acid frameworks for gastric cancer therapy

Jiaqi Yan [1,2,3], Rajendra Bhadane[2,4,5], Meixin Ran[2,3], Xiaodong Ma[2,3], Yuanqiang Li[2,3], Dongdong Zheng [6], Outi M. H. Salo-Ahen [2,5] & Hongbo Zhang [1,2,3] ✉

The metal-nucleic acid nanocomposites, first termed metal-nucleic acid frameworks (MNFs) in this work, show extraordinary potential as functional nanomaterials. However, thus far, realized MNFs face limitations including harsh synthesis conditions, instability, and non-targeting. Herein, we discover that longer oligonucleotides can enhance the synthesis efficiency and stability of MNFs by increasing oligonucleotide folding and entanglement probabilities during the reaction. Besides, longer oligonucleotides provide upgraded metal ions binding conditions, facilitating MNFs to load macromolecular protein drugs at room temperature. Furthermore, longer oligonucleotides facilitate functional expansion of nucleotide sequences, enabling disease-targeted MNFs. As a proof-of-concept, we build an interferon regulatory factor-1(IRF-1) loaded $Ca^{2+}$/(aptamer-deoxyribozyme) MNF to target regulate glucose transporter (GLUT-1) expression in human epidermal growth factor receptor-2 (HER-2) positive gastric cancer cells. This MNF nanodevice disrupts GSH/ROS homeostasis, suppresses DNA repair, and augments ROS-mediated DNA damage therapy, with tumor inhibition rate up to 90%. Our work signifies a significant advancement towards an era of universal MNF application.

The integration of metals with DNA structures can lead to the creation of exceedingly powerful materials[1,2]. One noteworthy example is deoxyribozyme (DNAzyme), a therapeutical material capable of targeting gene regulation through the interaction with metal ions[3]. Their safety and efficacy in human have already been validated[4]. Expanding upon this, in 2019, the metal-nucleic acid coordination methods were further utilized to build metal-DNA nanocomposites, to co-delivery small molecule drugs and CpG oligonucleotides[5,6]. Based on its structure and composition, here we first define this material as Metal-Nucleic Acid Frameworks (MNFs).

In recent years, MNFs have gradually gained momentum in the field of cancer treatment by combining the catalytic functions of metal ions with the therapeutic functions of nucleic acid drugs[7-10]. MNFs are composed of metal ions and DNA sequences, resembling another famous material "Metal-Organic Frameworks (MOFs)" that consist of metal ions and organic ligands[11]. One of the primary advantages of MNFs over MOFs is the multifunctionality of DNA sequences[12] compared to organic ligands. Specifically, DNA sequences in MNFs can serve as therapeutic nucleic acid drugs[13], facilitate specific cell targeting through adapter sequences[14], and enable disease gene diagnosis

[1]Department of Orthopaedics Shanghai Key Laboratory for Prevention and Treatment of Bone and Joint Diseases Shanghai Institute of Traumatology and Orthopaedics Ruijin Hospital Shanghai Jiao Tong University School of Medicine 197 Ruijin 2nd Road, Shanghai 200025, PR China. [2]Pharmaceutical Sciences Laboratory, Faculty of Science and Engineering, Åbo Akademi University, Turku, Finland. [3]Turku Bioscience Centre, University of Turku and Åbo Akademi University, Turku, Finland. [4]Institute of Biomedicine, University of Turku, Turku, Finland. [5]Structural Bioinformatics Laboratory, Biochemistry, Åbo Akademi University, 20520 Turku, Finland. [6]Department of Ultrasound, Fudan University Shanghai Cancer Center, Shanghai 200032, PR China. ✉e-mail: hongbo.zhang@abo.fi

**a**   Aptamer-DNAzyme/Ca biomineral preparation process and GLUT-1 mRNA regulation

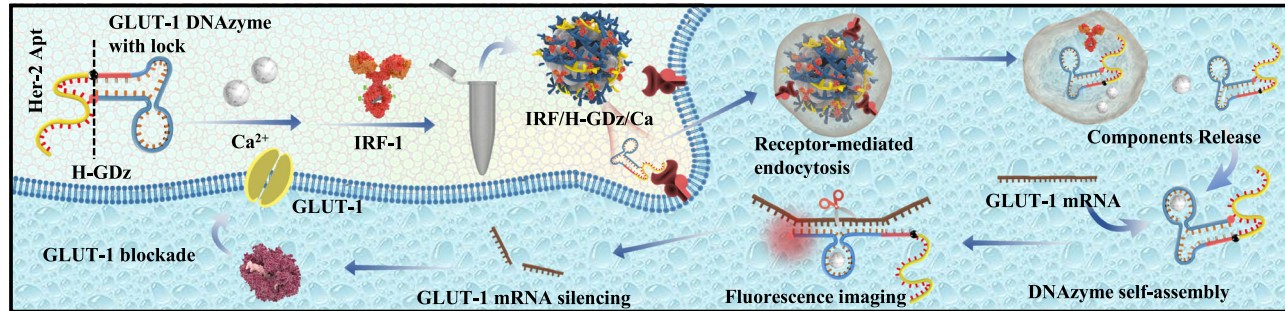

**b**   GLUT-1 inhibition-caused GSH depletion and Ca²⁺-mediated ROS up-regulation

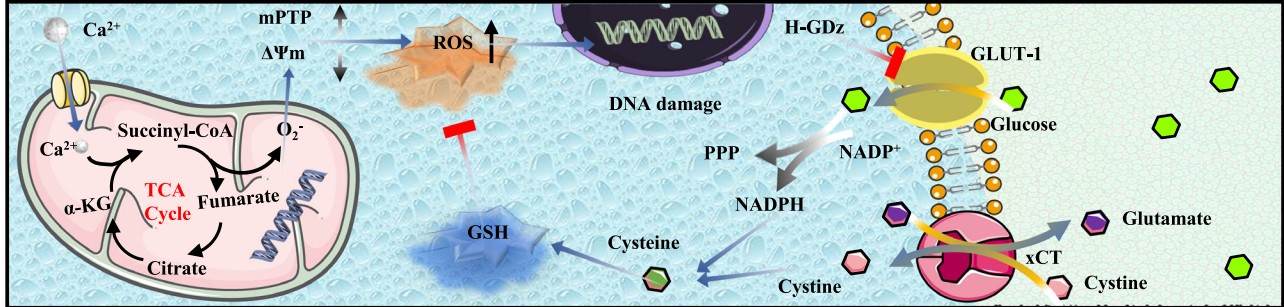

**c**   Augmentation of ROS-mediated DNA damage contributed by ATP depletion and IRF-1 based RAD51 inhibition

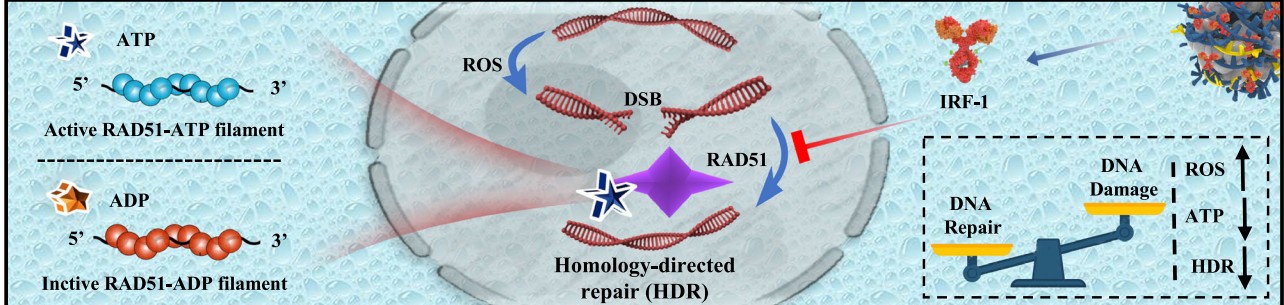

**d**   Systematic energy exhaustion through glycolysis and oxidative phosphorylation inhibition

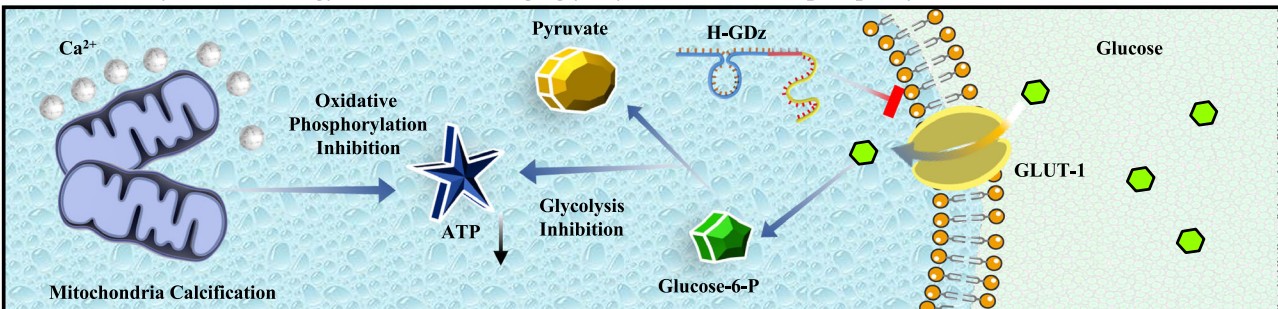

**Fig. 1 | Illustration depicting the process of preparing the streamlined MNF and its therapeutic mechanism for ROS-mediated gastric cancer therapy. a** The fabrication of IRF/H-GDz/Ca MNFs and its GLUT-1 monitoring and regulation function. **b** Inhibition of GLUT-1 impacts the homeostasis of GSH/ROS, enhancing mitochondrial calcium overload and subsequent damage to the cell nucleus. **c** IRF-1 inhibition suppresses the DNA homologous recombination repair function of RAD51, thereby enhancing ROS-mediated DNA damage. **c, d** Inhibition of GLUT-1 leads to ATP depletion and synergistically downregulates the function of RAD51. Abbreviations:mitochondrial permeability transition pore (MPTP), cystine/gluta-mate antiporter SLC7A11 (xCT), Alpha-ketoglutarate (α-KG).

via the construction of molecular fluorescence beacons[15,16]. Consequently, the utilization of MNF materials for drug delivery not only exploits the inherent properties of the drug but also harnesses the diverse functionalities of the DNA sequences.

However, despite its advantages characteristics combining those of both metal-organic-framework (MOFs) and DNA nanotechnology, MNFs have not been fully developed to unleash their tremendous potential. One of the major drawbacks is the stringent synthesis

conditions required for MNF materials, while also necessitate surface modifications to enhance stability or achieve further cellular targeting[8,9]. For example, the most extensively studied Fe²⁺/nucleic acid MNF was prepared under conditions 95 °C for 3 h, but it degraded completely within 2 hours in PBS and required further MOF layering to protect it[5,6]. Furthermore, for metals such as calcium, which do not easily form MNF structures with nucleic acids, the synthesis process required 120 °C for 3 days[10], which makes it difficult to load

macromolecular active drugs. Another limitation is that the functional oligonucleotide sequences currently used to construct MNFs are relatively simple, consisting of only around 20 bases/base pairs of CpG, DNAzymes, siRNA, etc., and have not fully leveraged the advantages of DNA nanotechnology. Hence, it is crucial to improve the preparation into mild conditions and guarantee efficient intracellular delivery of macromolecular drugs using multifunctional DNA sequences to broaden the applications of MNFs.

Currently, the factors that affect MNFs synthesis can be summarized as the type of nucleic acid bases, and the concentration of nucleic acid and metal ions. While different nucleic acid bases exhibit varying binding affinities with different metal ions, the ability to construct MNFs by using any nucleic acid bases remains unaffected, demonstrating its universality. Therefore, the concentration ratio of nucleic acids and metal ions has always been an important means to change the size of MNFs. However, the attention to the length of the nucleic acid sequence has been largely overlooked in the construction of MNF materials. Oligonucleotides lack base stacking forces and have flexible spatial conformations[17]. Hence increasing the length of oligonucleotide sequences, unlike increasing nucleotide concentration, theoretically helps to improve the folding and even entanglement probability of oligonucleotide sequences during the reaction process[18,19]. This, in turn, should increase the synthesis efficiency and stability of MNFs. Moreover, the entanglement of long oligonucleotide chains may counteract the repulsive forces between DNA molecules and enhances the attraction of DNA towards calcium ions, thereby increasing the yield of MNFs. Besides, from the perspective of DNA nanotechnology, longer DNA chains help to realize the functional expansion of nucleic acid sequences. Therefore, we hypothesize that using longer oligonucleotides (more than 20 bases[5,10]) with multifunctional segments to bind metal ions will help to increase the likelihood of successful MNFs synthesis, facilitate mild condition preparation, and enable efficient loading of macromolecular active drugs into MNF construct.

Combining the catalytic activity of metal ions with the therapeutic function of nucleic acids has long been a major design advantage of MNFs in cancer treatment[20–22]. In this study, we incorporated a HER-2 targeted aptamer into the $Ca^{2+}$-dependent GLUT-1 DNAzyme[23], thereby elongating the oligonucleotide length to achieve $Ca^{2+}$-assisted self-mineralization under room temperature, rendering visualized and silenced treatment of GLUT-1 mRNA (Fig. 1a). The HER-2 targeting aptamers were selected through SELEX (Systematic Evolution of Ligands by Exponential enrichment) studies by previous researchers[24–26]. This approach not only increases the MNFs synthetic capacity but achieves targeted delivery of MNFs to HER-2 over-expressing cells. The choice of calcium ions during MNF material preparation was based on several factors. Firstly, the responsiveness of the DNAzyme to calcium ions enabled the provision of essential metal cofactors for the DNAzyme following material degradation. Secondly, the diverse anti-cancer functions of calcium ions were a focal point of our study, particularly their role in mitochondrial calcification. Thirdly, considering material construction, previous research and our own investigations have demonstrated the challenging nature of achieving robust self-assembly between calcium ions and DNA to form MNF materials. Therefore, our use of calcium ions highlights how elongated DNA sequences can enhance the binding capacity of DNA sequences with metal ions.

Moreover, inhibiting glucose uptake at an early step may enhance the effectiveness of therapies that induce oxidative stress compared to targeting downstream processes[27]. Hence, by specifically silencing GLUT-1 in cancer cells, we can effectively disrupt the pentose phosphate pathway (PPP) and NADPH synthesis, leading to a reduction in cysteine synthesis and causing GSH/ROS homeostasis disruption[28] (Fig. 1b). This disruption greatly amplifies the ROS levels generated by mitochondrial calcification, resulting in efficient DNA damage within

cancer cells[29] (Fig. 1b). However, gastric cancer cells exhibit elevated expression of RAD51[30–32], a protein involved in homologous recombination DNA repair, which relies on ATP[33–35] for rapid DNA repair and provides protection to cancer cells. To overcome this challenge, we employed a mild condition MNF preparation system to load interferon regulatory factor-1 (IRF-1)[36–38] (abbreviated as IRF/H-GDz/Ca), to suppress RAD51 expression[32] (Fig. 1c). In combination with silencing GLUT-1 to achieve ATP depletion (Fig. 1d), we synergistically downregulated the DNA homologous recombination repair capacity within gastric cancer cells, thereby enhancing ROS-mediated DNA damage therapy.

In this study, our innovation advances in regulating the length of oligonucleotides to facilitate the mild-condition preparation of MNFs materials, allowing efficient encapsulation of macromolecular drugs. Meanwhile, from the therapeutic viewpoint, we hypothesize that by effectively cellular delivery of our IRF-1 loaded H-GDz/Ca MNF nanoplatform (IRF/H-GDz/Ca), the therapeutic efficacy of mitochondrial calcification strategy could be greatly enhanced through GSH/ROS homeostasis disruption, intracellular ATP depletion, and RAD51 suppression. These innovations hold potential for improving the effectiveness of ROS-mediated DNA damage therapy and have significantly expanded the medical application scope of MNFs.

## Results

In order to investigate whether multi-segmented (longer) sequences possess stronger MNF synthesis capability, we designed three functional DNA fragments of different lengths (20 nucleotides, 41 nucleotides, and 83 nucleotides) at equimolar concentrations (83 mM nucleotides). Among them, the sequence comprising 41 nucleotides represented GLUT-1 DNAzymes with a fluorescence lock, referred to as GDz (Fig. 2a1). The sequence consisting of 83 nucleotides corresponded to GDz extended by HER-2 aptamer, denoted as H-GDz (Fig. 2a1). Lastly, the sequence comprising 20 nucleotides represented the GLUT-1 substrate strand. The findings revealed that H-GDz exhibited to effectively bind with calcium ions, resulting in a 20% higher yield of nanoparticles compared to GDz with 41 bases (Fig. 2a2, Supplementary Fig. 1). However, unexpectedly, the 20-base sequence was incapable of binding with calcium ions and forming nanoparticles (NPs) (Fig. 2a2, Supplementary Fig. 1). These observations provided macroscopic evidence supporting the superior MNF synthesis capability of longer DNA sequences compared to shorter ones.

Subsequently, to further investigate the structural composition of the MNF, we analyzed the DNA and calcium elements content in the generated nanomaterials (Table S2). We observed that for the H-GDz sequence, when the input amount of DNA was approximately 52.39 μg, the actual amount of DNA in the obtained product was around 50.66 μg, indicating a high utilization efficiency of the DNA sequence (96.7%). Similarly, for the GDz sequence, the input amount of DNA was approximately 51.5 μg, and the utilization efficiency of DNA in the product was also high (92.2%), with a content of around 47.5 μg. However, the difference between the two lies in the utilization of calcium ions. The product of the H-GDz group contained around 10.6 μg of calcium (calcium mass ratio of 17.3%), while the GDz group had only about 5.8 μg of calcium (calcium mass ratio of 10.8%). These phenomena highlighted that, although the utilization efficiency of DNA remained consistent for both GDz and H-GDz during MNF synthesis, longer DNA sequences exhibited a stronger binding capacity towards calcium ions. Interestingly, when the DNA sequence was shortened to 20 bases, despite an input amount of approximately 50.8 μg, the resulting product was barely perceptible to the naked eye. Further analysis revealed a DNA content of only 1.36 μg, indicating a utilization efficiency of merely 2.6%. Moreover, the calcium element content was measured at 0.08 μg, corresponding to a calcium mass ratio of 5.7%. These findings underscored the drastic difference in product yield and highlighted the pivotal role of DNA sequence length in maximizing the binding capacity with calcium ions.

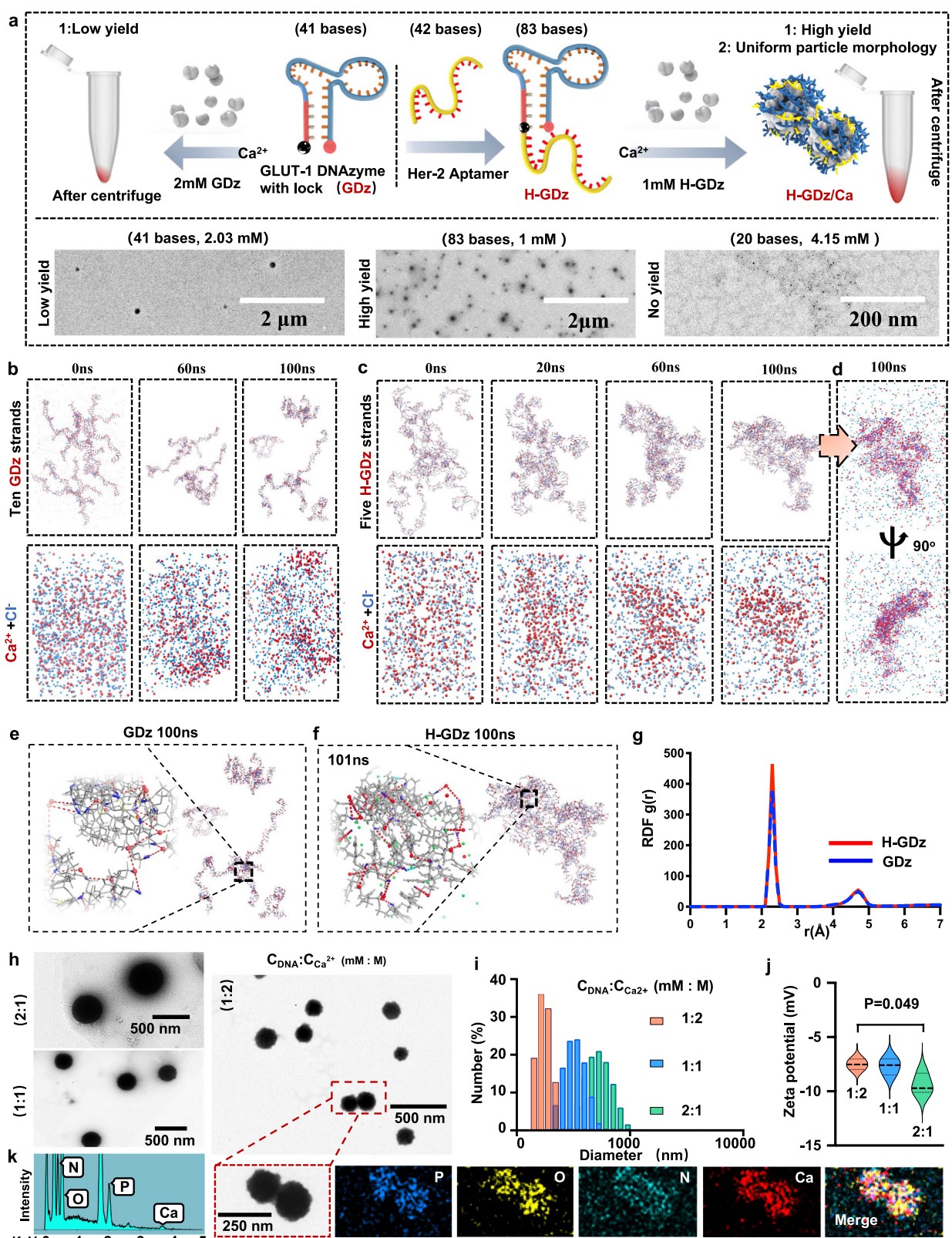

The observed results can be attributed to the inherent characteristics of oligonucleotides, which are known to lack strong base stacking forces and exhibit flexible spatial conformations[18]. Consequently, increasing the length of oligonucleotide sequences (which is different from augmenting the nucleotide concentration), holds the potential to enhance the folding and entanglement probability of the sequences during the reaction process[39,40]. Hence, we hypothesize that

the entanglement of longer oligonucleotide chains endows the intertwined DNA clusters with a higher potential, which in turn enhances the affinity of DNA for calcium ions, thus, obtaining higher MNF production.

To obtain molecular level insight into the aforementioned hypothesis, molecular dynamics (MD) simulations were employed, where two DNA systems of identical nucleotide counts were

**Fig. 2 | Mechanism of MNF synthesis based on long-chain DNA binding with calcium ions and regulation of MNF properties. a** MNF synthesis process of GDz and H·GDz, and the MNF TEM results for GDz (41 bases), H·GDz (83 bases) and the substrate strand (20 bases) ($n = 3$ independent experiments with similar results). **b, c, d** Molecular dynamics (MD) simulation results of 10 GDz and 5 H·GDz with the nearly same nucleotide content and the same concentration of calcium ions. Calcium ions are depicted as red spheres, while chloride ions are represented by blue spheres. **e, f** Electrostatic binding interactions between calcium ions and the oxygen atoms of the phosphate backbone in the DNA sequences. Atom color code: gray – DNA bases; blue – oxygen atom in phosphate backbone; Red color spheres depict $Ca^{2+}$; green color spheres depict $Cl^-$; red dashed lines denote electrostatic interactions. **(g)** The radial distribution function (RDF) analysis of GDz and H·GDz sequences. The horizontal axis represents the distance between calcium ions and the negatively charged oxygen atoms in the DNA phosphate backbone, while the vertical axis represents the probability (density) of finding a calcium ion at a certain distance from a negatively charged phosphate oxygen atom. **h** The transmission electron microscopy (TEM) images of MNF nanoparticles formed under different DNA/Ca ratio conditions including 1:2, 1:1, and 2:1 ($C_{DNA}:C_{Ca}^{2+}$, mM:M) ($n = 3$ independent experiments with similar results). **i, j** Dynamic light scattering (DLS) data of MNF nanoparticles at different DNA/Ca ratio ($n = 3$ independent experiments). The bold dashed line in the middle is the median, and the unbold line is the quartiles. **k** Energy-dispersive X-ray spectroscopy (EDS) and elemental mapping analysis of MNF nanoparticles ($n = 3$ independent experiments with similar results). Statistics in (**j**) were calculated using one-way ANOVA, followed by the Tukey post-hoc test for multiple comparisons. The source data from (**i, j, k**) are provided as a Source Data file.

constructed (Fig. 2b, c). System 1 consisted of ten GDz sequences, while system 2 comprised five H·GDz sequences. Throughout the simulated reaction process ranging from 0 ns to 100 ns, it was observed that the shorter GDz sequences (strand length 273 Å) persisted in an unstable and dispersed state, exhibiting minimal entanglement between the sequences even at 100 ns. Conversely, in the longer H·GDz system (strand length 645 Å), the DNA fragments displayed a pronounced intertwining phenomenon, forming a densely packed aggregation by 100 ns (Fig. 2d). This uncovering suggests that longer DNA sequences are more inclined to undergo entanglement and intertwining. To further demonstrate this hypothesis, we calculated another parameter, persistence length (Lp), which is used to measure the mechanical bending stiffness of polymeric chains. A smaller Lp value indicates higher flexibility and, consequently, a higher degree of folding. The results indicate that the long-chain DNA, H·GDz, has an Lp value of 5.476 Å, while the short-chain DNA, GDz, has an Lp value of 9.527 Å (Supplementary Fig. 2). Therefore, H·GDz is indeed softer than GDz, with a higher probability of folding. Moreover, in both systems, the calcium ions were observed to exhibit a notable affinity towards the DNA sequences, forming strong binding interactions, while the sodium and chloride ions inside medium did not exhibit a discernible binding tendency with the DNA sequences (Fig. 2b, c and Supplementary Fig. 3). Subsequent analysis revealed that the calcium ions predominantly interacted with the oxygen atoms of the DNA phosphate backbone, forming salt bridges (Fig. 2e, f).

To investigate whether the longer DNA sequences have a greater tendency to bind with the calcium ions, we further analyzed the radial distribution function (RDF) of calcium atoms with respect to oxygen atoms in the DNA phosphate backbone (Fig. 2g). The RDF provide information about the probability distribution of distances between specific pairs of atoms in the system. The results show that long-chain H·GDz attracts a higher density of calcium ions than short-chain GDz. This may be because long-stranded DNA accumulates high negative charge after being entangled, so it can attract more calcium ions. These results shed light on the calcium binding behavior of longer DNA sequences and their potential in the synthesis of calcium-based MNFs.

Therefore, by combining the practical experiments and computational approaches, we have obtained convincing evidence on the significant role of DNA sequence length in the synthesis of MNFs. By increasing the length of DNA oligonucleotide sequences, we can achieve the stable synthesis of calcium-based MNFs at room temperature, thereby imparting them with a broader range of potential applications. Subsequently, building upon the utilization of multi-segmented DNA sequences (H·GDz), we conducted further investigations into the impact of the DNA/$Ca^{2+}$ concentration ratio on the resulting nanoparticle size (Fig. 2h). Interestingly, we observed that when the DNA: $Ca^{2+}$ concentration ratio (mM:M) was lower (1:2), the resulting nanoparticles exhibited smaller particle sizes around 130 nm, while the 1:1 and 2:1 presented as around 190 nm and 500 nm (Fig. 2h, i), aligning with findings reported in other studies[5]. Furthermore, it was noteworthy that as the particle size increased, the nanoparticle's electrostatic potential decreased, indicating a higher proportion of negatively charged DNA in larger particles (Fig. 2j). We further verified the stability of the nanomaterials in DMEM culture medium and found that their morphology remained unchanged within DMEM for one week, ensuring the stability of MNF (Supplementary Fig. 4).

In the final step, we conducted Energy-dispersive X-ray spectroscopy (EDS) analysis specifically on the MNFs with a DNA/$Ca^{2+}$ ratio of 1:2 (Fig. 2k). The results confirmed the presence of the target elements in the nanoparticles, including calcium (Ca), phosphorus (P), nitrogen (N), and oxygen (O). This analysis provided direct evidence for the successful synthesis of MNFs.

Following the successful synthesis of stable calcium-based H·GDz/Ca NPs at room temperature, a comprehensive investigation was undertaken to explore their GLUT-1 gene imaging performance, regulation functionality, and cell-targeting capability towards HER-2 positive gastric cancer. First, glut-1 is generally described as ubiquitous present, but like other glucose transporter subtypes, its expression has tissue specificity[41]. In addition to specific high glucose uptake tissues, such as astrocytes, high levels of GLUT-1 expression have been an indicator of carcinogenesis (GLUT-1 can be detected in the necrotic areas of many human tumor types, but not in normal tissues)[42,43]. Hence, here we validated the detection capability of the H·GDz molecular beacon for GLUT-1 mRNA (Fig. 3a). This molecular beacon was composed of a dual-labeled fluorescent probe, FAM/BHQ-1, where the fluorescence of FAM was effectively quenched by BHQ-1 in the absence of GLUT-1 mRNA through forster resonance energy transfer (FRET) effect. When the target gene was encountered, the molecular beacon underwent a conformational change, resulting in the release of FAM fluorescence and the visualization of the target gene, marked as red color.

For mRNA detection, the current detection limit is approximately 100 nM[44]-1 µM[45], depending on the detection system and system concentration. For intracellular mRNA detection, single-molecule fluorescence in situ hybridization (FISH) technology is now capable of detecting individual mRNA in fixed cells[46,47]. Various techniques for live-cell real-time imaging can amplify mRNA detection signals and achieve low-abundance mRNA detection[48,49]. In this work, the preparation of DNAzyme molecular beacons served to demonstrate the persistent activity of functionalized DNAzymes within MNF materials after cell internalization, validate their target substrate (GLUT-1 mRNA) recognition capability, and harness DNA nanotechnology for diverse MNF applications. From the results, we observed that the fluorescence intensity of the system increased with increasing concentrations of GLUT-1 mRNA substrate (Fig. 3a), which indicated that the H·GDz molecular beacon could specifically recognize and bind to GLUT-1 mRNA, leading to an enhanced fluorescence signal in the presence of the target mRNA.

Subsequently, to investigate the cleaving capability of H·GDz on GLUT-1 mRNA at different calcium ion concentrations (ranging from 0

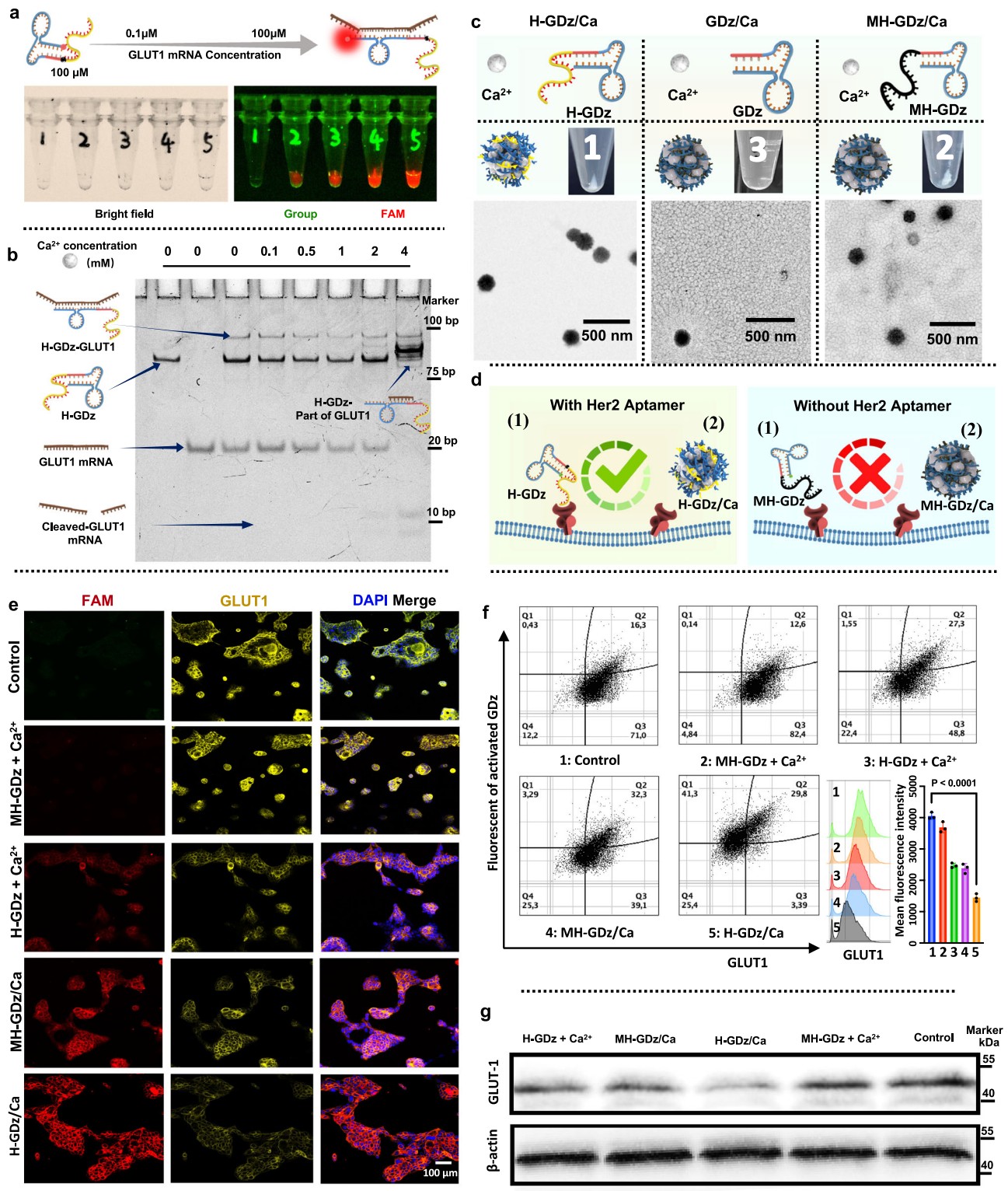

to 4 mM), PAGE (polyacrylamide gel electrophoresis) analysis was employed (Fig. 3b). The results demonstrated that, under Ca²⁺ free situation, when H-GDz was co-incubated with GLUT-1 mRNA, a new delayed shifted band was observed, indicating the successful binding of the two sequences (Fig. 3b and Supplementary Fig. 5). However, due to the absence of calcium ions, the mRNA was not cleaved. As the calcium ion concentration was gradually increased, a noticeable decrease in the intensity of the GLUT-1 mRNA band was observed at 0.5 mM (Supplementary Fig. 6). At a Ca²⁺ concentration of 2 mM, the

appearance of a cleaved GLUT-1 mRNA band was observed, indicating the successful cleavage of the target gene. Furthermore, at a Ca²⁺ concentration of 4 mM, the target GLUT-1 gene was completely silenced (Fig. 3b and Supplementary Fig. 6). We observed a new band between H-GDz-GLUT1 and H-GDz at calcium ion concentrations exceeding 2 mM (indicating extensive cleavage of the GLUT-1 substrate). Its molecular weight falls between that of H-GDz-GLUT1 and H-GDz, possibly suggesting the presence of a fragment of the cleaved substrate still attached to the DNAzyme. These findings elucidated that

**Fig. 3 | Validation of the targeting ability of each material towards NCI-N87 cells and characterization of their detection and regulation capabilities towards GLUT-1.** **a** PAGE analysis to validate the GLUT-1 recognition and cleavage capabilities of H-GDz sequences ($n = 3$ independent experiments with similar results). **b** The responsiveness of H-GDz towards GLUT-1. The system containing 100 μM of H-GDz and GLUT-1 mRNA substrates were added ranging from 0.1 μM to 100 μM ($n = 3$ independent experiments with similar results). **c** The design and synthesis of targeted materials ($n = 3$ independent experiments with similar results). **d** The schematic illustration of the targeting mechanism of HER-2 modified material. **e** Fluorescence confocal microscopy was employed to evaluate the detection and regulation capabilities of various materials on the GLUT-1 gene in the cellular context ($n = 3$ independent experiments with similar results). The intracellular GLUT-1 protein was labeled with antibody and represented as yellow fluorescence, the DNAzyme was labeled as red, while the cell nuclei were stained with DAPI and shown as blue color. **f** Images obtained from flow cytometry analysis to assess the silencing efficacy of different groups on GLUT-1 expression ($n = 3$ independent experiments with similar results). The x-axis represented the fluorescence detection intensity of GLUT-1, while the y-axis represented the fluorescence of activated GDz, upon hybridization between GDz and GLUT-1. **g** Western blot analysis to demonstrate the silencing efficacy of different groups on GLUT-1 expression ($n = 3$ independent experiments with similar results). Statistics in (**f**) were calculated using two-tailed paired *t* tests. The source data from (**f**, **g**) are provided as a Source Data file.

the depleting activity of H-GDz on GLUT-1 mRNA was influenced by the concentration of calcium ions, with higher concentrations leading to more pronounced gene regulation effects.

Furthermore, we investigated the targeting ability of H-GDz towards HER-2 positive NCI-N87 gastric cancer cells. To comparatively evaluate the functionality of the HER-2 aptamer, we designed H-GDz sequences containing the HER-2 aptamer, MH-GDz sequences containing a Mutated-HER-2 aptamer, and GDz sequences without the HER-2 aptamer, for the preparation of MNF materials (Fig. 3c). The choice of mutant bases in the Mutated-Her-2 aptamer was driven by two key considerations. Firstly, altering the order of GC pairs has a profound impact on the secondary structure of the aptamer, and this was confirmed through simulations using NUPACK software (Supplementary Fig. 7). Secondly, our experimental goal was to investigate the influence of DNA sequence length on the construction ability of MNF. Hence, we only modified the arrangement of GC pairs, without introducing any new base or new sequence, to ensure a focused exploration of length-related impacts during MNF construction. The results showed that the production yields of H-GDz/Ca and MH-GDz/Ca nanomaterials were similar, while the GDz group without the HER-2 sequence exhibited a significantly lower production yield. Therefore, for the design of nanosystems, H-GDz/Ca and MH-GDz/Ca groups were selected for the further comparative analysis and validation.

The NCI-N87 cell line was derived from the stomach of a male patient with gastric carcinoma in 1976. Due to its high expression of the HER-2 receptor, it has been utilized for subsequent experiments involving HER-2 targeted delivery of MNF. The targeting effect of H-GDz towards NCI-N87 cells was manifested in two aspects: the targeting ability of the MNF materials and the DNA sequence itself (Fig. 3d). The endocytic ability of each material group can be reflected by the detected GLUT-1 mRNA (represented by the red signal of FAM), as higher endocytic intensity corresponds to a stronger GLUT-1 detection signal. Moreover, from the results of lysosomal escape (Supplementary Fig. 8), it can be observed that there is a better overlap between MNF and lysosomes at 2 h and reduced overlap at 8 h, reflecting that the MNF nano-systems were able to effectively escape from the lysosomes at 8 h. Therefore, 10 h timepoint was selected, and fluorescent confocal microscopy (Fig. 3e) and flow cytometry (Fig. 3f) were used to evaluate the nanomaterial groups (H-GDz/Ca and MH-GDz/Ca) as well as the pure DNA sequence groups (H-GDz + Ca²⁺ and MH-GDz + Ca²⁺) gene regulation ability. The concentration of free calcium ions was set to be consistent with the nanomaterials.

From the results in Fig. 3e and higher magnification images in Supplementary Fig. 9, compared with MH-GDz/Ca NPs group, the H-GDz/Ca NPs group exhibited 2.95 times (Supplementary Fig. 10) higher efficient GLUT-1 detection, suggesting great cell internalization of H-GDz/Ca NPs. In contrast, the cellular uptake of free DNA sequences was significantly diminished comparing with NPs groups, due to their high negative charge, as less GLUT-1 signal can be detected. However, remarkably, even in the presence of the highly negative charge, we observed a higher imaging capability for GLUT-1 in the H-GDz + Ca²⁺

group compared to the MH-GDz + Ca²⁺ group, highlighting the crucial targeting role of HER-2 aptamer. Furthermore, as the gene detection signal increased, the GLUT-1 mRNA was depleted, leading to a downregulation of GLUT-1 protein expression. The results indicated that compared to the control group, the H-GDz + Ca²⁺ group experienced a 10.3% reduction in GLUT-1 protein expression, while the H-GDz/Ca MNF NPs group exhibited a significant 63.6% downregulation. These fluorescence confocal findings demonstrated that the nanomaterial group possessed superior gene-regulating capabilities compared to free DNA sequences, and the presence of HER-2 aptamer enhanced the internalization ability of NCI-N87 cells towards the materials.

For the purpose of validating the robustness of the aforementioned findings, we conducted a replicate verification utilizing an additional HER-2 positive gastric cancer cell line SNU216. The results similarly confirm that H-GDz/Ca materials, when possessing the correct HER-2 sequence, exhibit significant internalization in SNU216 cells (Supplementary Fig. 11a, b), comparing with the MH-GDz/Ca group. Furthermore, the internalization of MNF materials is notably higher compared to the individual H-GDz sequence. In addition to validating the impact of the aptamer sequence's accuracy on endocytosis, we also examined the internalization of MNF materials by N87 cells versus HER-2-negative normal human dermal fibroblasts (NHDF) (Supplementary Fig. 12). The results similarly demonstrated the selectivity of MNF materials for HER-2-positive cell lines.

Flow cytometry analysis further confirmed the cell-targeting ability of the HER-2 integrated materials (Fig. 3f and Supplementary Fig. 13). Compared with control group, GLUT-1 expression was reduced by approximately 37.5% in the H-GDz + Ca²⁺ group, while significantly reduced by up to 70% in the H-GDz/Ca NPs group. Simultaneously, a noticeable shift of cells from the Q3 region (71.0 %) characterized by high GLUT-1 expression and low GLUT-1 detection towards the Q1 region (41.3 %) was observed. Therefore, these findings provided compelling evidence that the transformation of H-GDz into nanostructured MNF materials through self-mineralization techniques can significantly enhance its intracellular gene monitoring and regulating capabilities.

Western blotting was also utilized to further validate the expression of GLUT-1 within the N87 and SNU216 cells (Fig. 3g and Supplementary Fig. 11c). The results showed that the HER-2 modified H-GDz + Ca²⁺ group exhibited a slight decrease in GLUT-1 protein expression, surpassing the effect observed in the HER-2 mutated MH-GDz + Ca²⁺ group. Of note, the H-GDz/Ca nanomaterial group exhibited the strongest inhibitory effect on GLUT-1, indicating again that the nanomaterials possess higher GLUT-1 expression regulatory capabilities compared to free DNA strands.

After confirming the superior gene regulation capability of HER-2 aptamer-contained MNF nano-system (H-GDz/Ca), we embarked on an investigation into its loading content of the DNA homologous recombination repair protein RAD51 inhibitor, IRF-1. We initially characterized the size and morphology of MNF materials after loading them with protein drugs (Supplementary Fig. 14). DLS analysis showed that the average particle size of the IRF/H-GDz/Ca increased from

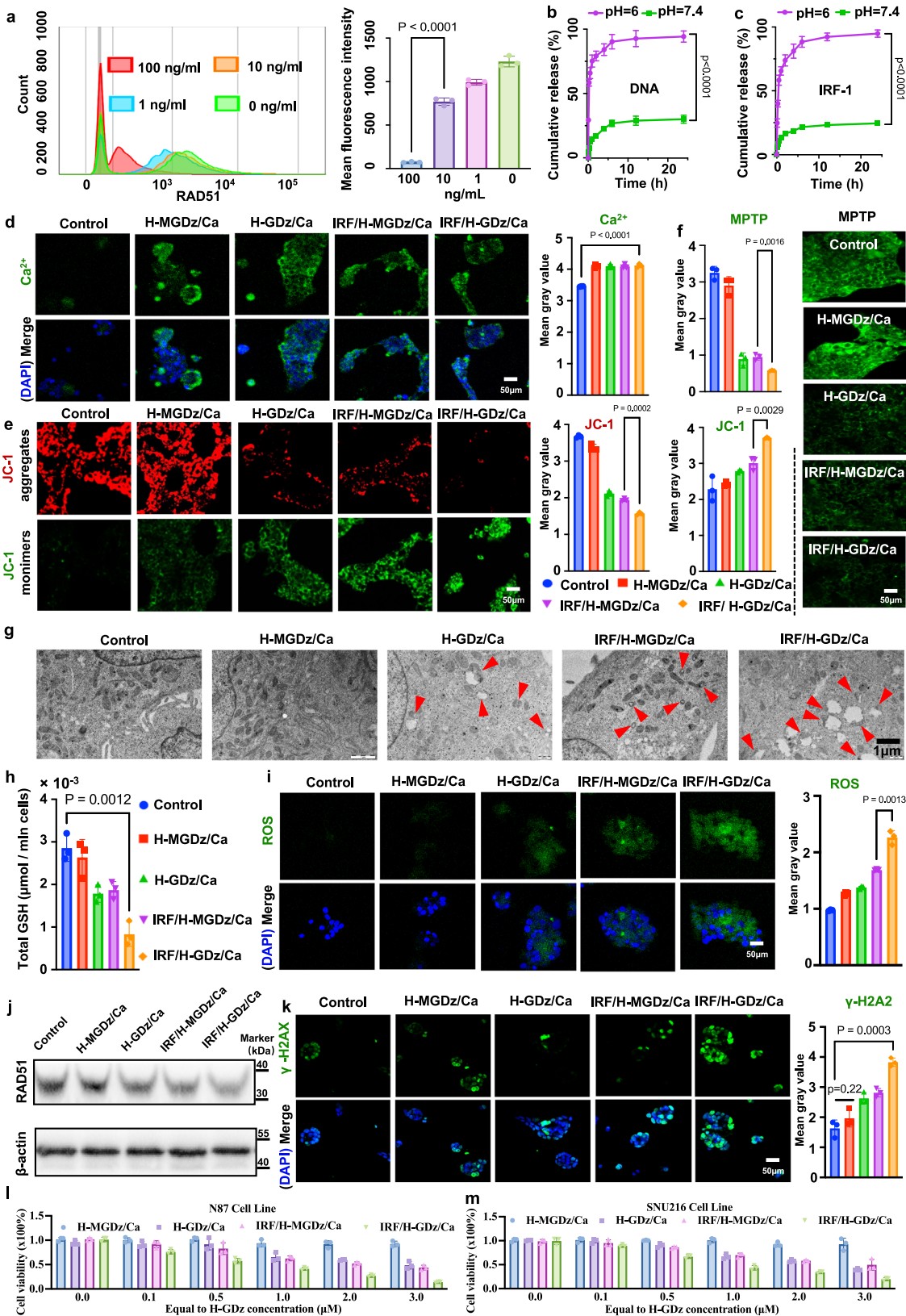

around 130 nm (H-GDz/Ca) to about 200 nm. Additionally, TEM images further confirmed these observations, illustrating the uniform spherical structure and distribution of the IRF-1 loaded particles. Subsequently, employing flow cytometry techniques, the inhibitory efficacy of IRF/H-GDz/Ca system on RAD51 were examined (Fig. 4a). The findings unveiled a substantial reduction in RAD51 expression

upon achieving an IRF-1 concentration of 100 ng/mL. Consequently, guided by the therapeutic dose of IRF-1 and the requisite DNAzyme dosage (500 nM)[50], we determined the optimal loading amount of IRF-1 as 0.4 µg within 2 nmol H-GDz system.

Additionally, the release of DNA sequences (Fig. 4b) and IRF-1 (Fig. 4c) from IRF/H-GDz/Ca NPs under varying pH conditions were

**Fig. 4 | Investigation of the synergistic therapeutic effects of IRF-1 in conjunction with GLUT-1 inhibition materials. a** The regulatory effect of IRF/H-GDz/Ca MNF NPs loaded with different concentrations of IRF-1 (ranging from 0 to 100 ng/mL) on RAD51 in NCI-N87 cells ($n = 3$ independent experiments and the data are presented as mean values ± SD). **b, c** The release of DNA and IRF-1 from IRF/H-GDz/Ca MNF NPs in different pH release media ($n = 3$ independent experiments and the data are presented as mean values ± SD). **d** Intracellular $Ca^{2+}$ content measurement ($n = 3$ independent experiments and the data are presented as mean values ± SD). **e** Mitochondrial membrane potential assessment ($n = 3$ independent experiments and the data are presented as mean values ± SD). JC-1 dye formed aggregates that emitted a red fluorescence signal. Conversely, JC-1 remained in its monomeric form and emitted a green fluorescence signal. **f** A combination indicator comprising Calcein AM and $CoCl_2$ for the evaluation of mitochondrial membrane permeability, In the presence of a closed MPTP, $CoCl_2$ quenched the cytoplasmic Calcein fluorescence, resulting in exclusive green fluorescence emitted by the mitochondria ($n = 3$ independent experiments and the data are presented as mean values ± SD). **g** Bio-TEM images of mitochondria in various cellular conditions ($n = 3$ independent experiments with similar results). **h** Quantification of intracellular glutathione (GSH) levels ($n = 3$ independent experiments and the data are presented as mean values ± SD). **i** Measurement of intracellular reactive oxygen species (ROS) levels, by using the green fluorescence probe 2',7'-Dichlorodihydrodrofluorescein diacetate (DCFH-DA) ($n = 3$ independent experiments and the data are presented as mean values ± SD). **j** Western blot analysis confirming the modulation of intracellular RAD51 by different nanomaterials ($n = 3$ independent experiments and the data are presented as mean values ± SD). **k** Assessment of nuclear DNA damage. The Mean Gray Value (average gray value within the selection) is calculated using ImageJ to determine the average intensity of pixels within the selected area, and it is independent of cell density ($n = 3$ independent experiments and the data are presented as mean values ± SD). **l, m** Cytotoxicity experiments of each MNF material group on N87 and SNU216 cells ($n = 3$ independent experiments and the data are presented as mean values ± SD). Statistical comparisons were performed using one-way ANOVA, followed by the Tukey post-hoc test for multiple comparisons. The source data underlying figures (**a, b, d, e, f, h, i, k, l, m**) are available in the provided Source Data file.

studied. The results demonstrated the remarkable pH-sensitive release capability of the nanosystem, with both the H-GDz sequence and IRF-1 protein exhibiting comparable release efficiencies. In a pH = 6 release medium, complete release of both DNA and IRF-1 was achieved within 24 h, while only approximately 20% of the payload was released within the same timeframe under pH = 7.4 condition. These findings highlighted the exquisite pH-responsive release properties of the nanosystem, indicating its potential for targeted and controlled drug release profiles.

Following this, we proceeded to validate the DNA damage therapeutic effects of the formulation. The ROS mediated damage was induced by mitochondrial calcium overload, the disruption of ROS/GSH homeostasis mediated by GLUT-1 inhibition, and the inhibition of DNA repair protein RAD51 by IRF-1. Five groups including Control, H-MGDz/Ca, H-GDz/Ca, IRF/H-MGDz/Ca, and IRF/H-GDz/Ca were designed, and MGDz sequence is a mutated sequence of GLUT-1 DNAzymes with same nuclear bases. All the other groups possess the HER-2 aptamer, while the distinction lies in the presence or absence of GDz and IRF. First, the intracellular calcium levels were characterized (Fig. 4d). The results showed that the four nano-groups exhibited similar intracellular calcium levels compared to the control group, demonstrated that HER-2 incorporated MNF system possessed comparable internalization capabilities, unaffected by GDz sequence variations or protein loading.

Subsequently, the health status of mitochondria was assessed by monitoring the mitochondrial membrane potential (ΔΨm) (Fig. 4e), and JC-1 dye was served as a key indicator[51]. The results indicated that in the control group, JC-1 mainly presented as red fluorescence signals in the form of aggregates. When H-MGDz/Ca NPs was added, no significant enhancement of free JC-1 green fluorescence was observed. This suggested that there was no apparent mitochondrial disruption effect without GLUT-1 regulation. Conversely, in the H-GDz/Ca treatment group, we observed a 45.9% decline in JC-1 red signal intensity, accompanied by an 27.6% increase in green fluorescence, indicating a reduction in mitochondrial membrane potential. This could be attributed to the inhibition of GLUT-1, which lead to ROS/GSH imbalance, causing elevated ROS to further induce mitochondrial damage[52]. Additionally, in the IRF/H-GDz/Ca group, upon loading IRF-1 in the MNF system, 60.5% of the JC-1 aggregate signal was substantially diminished, replaced by a strong green fluorescence from free JC-1. These results demonstrated that the addition of IRF−1 could suppress mitochondrial function, leading to a further decrease in mitochondrial membrane potential, as reported in other study[53].

Another indicator used to assess mitochondrial status was the mitochondrial permeability transition pore (MPTP). The opening of MPTP increased following mitochondrial damage (Fig. 4f). From the results, the fluorescence intensity of Calcein in the mitochondria

decreased by around 68% in the H-MGDz/Ca and IRF/H-MGDz/Ca groups, indicating a significant increase in mitochondrial permeability. Furthermore, in the IRF/H-GDz/Ca NPs group, the fluorescence intensity of Calcein in the mitochondria decreased by 78.1%, demonstrating a pronounced elevation in mitochondrial membrane permeability.

With Bio-TEM, the morphology of mitochondria under different therapeutic groups were further examined (Fig. 4g). The results revealed that the mitochondria in the H-MGDz/Ca group appeared like those in the control group, with well-defined cristae and intact structures. However, in the H-GDz/Ca group, the mitochondria showed evident damage, characterized by disrupted membranes and the presence of cavity-like structures. The IRF/H-MGDz/Ca group also exhibited noticeable mitochondrial impairment. Notably, the mitochondria in the IRF/H-GDz/Ca group displayed a significant number of damaged mitochondria with hollow-like cavities, indicating that the combination effects of GLUT-1 inhibition and IRF-1 introduction.

It has been reported that the inhibition of GLUT-1 results in a decrease in the levels of the pentose phosphate pathway (PPP) intermediate, 6-phosphogluconate (6PG), and a specific increase in the $NADP^+/NADPH$ ratio[28]. Furthermore, the NADPH downregulation will also block the cysteine synthesis. To investigate these alterations, we utilized the 6-phosphogluconate assay kit, $NADP^+/NADPH$ quantification kit, and the cysteine assay kit to assess the levels of 6PG, $NADP^+/NADPH$ ratio and cysteine concentration (Supplementary Fig. 15) in N87 cell line. From the results, we found a decreased of the 6PG concentration, while the $NADP^+/NADPH$ ratio was increased for the H-GDz/Ca and IRF/H-GDz/Ca groups. Meanwhile, cysteine concentration decreased for H-GDz/Ca and IRF/H-GDz/Ca groups, which proves that the GLUT-1 inhibition will block the PPP metabolism and further affects the cysteine synthesis. This series of metabolic outcomes was also validated in another SNU216 gastric cancer cell line (Supplementary Fig. 16a).

The cellular GSH content served as a critical indicator regulated by GLUT-1 inhibition. Hence, the GSH levels (μmol per million cells) were determined (Fig. 4h and Supplementary Fig. 16a). The results demonstrated that both H-GDz NPs and the IRF/H-MGDz/Ca group could effectively decreased intracellular GSH content by 35.7% and 32.1% in GSH levels, compared to the control group. The results supported the notion that inhibiting GLUT-1 or introducing IRF-1 could led to a notable reduction in intracellular GSH levels. The decrease observed in the GLUT-1 inhibition group could be attributed to the impaired GSH synthesis[28]. On the other hand, the decrease caused by IRF-1 could be attributed to the IRF-1 induced ROS generation and GSH depletion[54]. Notably, in the IRF/H-GDz/Ca group, a substantial decrease of 71.4% in GSH content was observed, highlighting the synergistic effect of GDz and IRF-1 in reducing the GSH content.

To verify whether the inhibition of GLUT-1 expression would result in a significant increase in intracellular ROS levels, ROS were directly detected as green fluorescent (Fig. 4i). Specifically, in the H-GDz/Ca group, the fluorescence signal of ROS increased by 16.7%. It is noteworthy that IRF-1 also demonstrated the ability to elevate ROS levels. In the IRF/H-MGDz/Ca group, the ROS signal increased by approximately 21.8%, consistent with previous research findings[54]. Furthermore, the ROS signal intensity in the IRF/H-GDz/Ca group increased by approximately 40.5%, providing strong evidence of the synergistic effect between IRF-1 and H-GDz in effectively enhancing intracellular ROS levels.

In addition to its discovered functions in enhancing mitochondrial damage and increasing ROS content, IRF-1 played a pivotal role in downregulating the expression of RAD51. This downregulation magnified the ROS-mediated damage to nuclear DNA, thereby amplifying the cellular DNA injury. Therefore, western blot analysis was performed to examine the expression levels of RAD51 in different treatment groups (Fig. 4j). To our surprise, both H-GDz/Ca and IRF/H-MGDz/Ca group were observed a decrease in RAD51 expression, showing that the inhibition of GLUT-1 also had a downregulation effect on RAD51. Of note, we found a remarkable suppression of RAD51 expression for IRF/H-GDz/Ca group, which could be attributed to the synergistic effect of GLUT-1 downregulation and the inhibitory action of IRF-1 on RAD51.

Later, immunofluorescence experiments using γ-H2AX green fluorescent antibody were performed to assess the extent of DNA damage within the N87 cell nucleus (Fig. 4k). As anticipated, the H-MGDz/Ca formulation did not induce much nuclear damage. However, when H-GDz/Ca NPs was used, the signal intensity of γ-H2AX fluorescence in the cell nucleus increased by 44.4%, indicating GLUT-1 inhibition can enhance DNA damaging efficacy. Meanwhile, the signal intensity of γ-H2AX fluorescence in the IRF/H-MGDz/Ca group also increased by 55.6%. This can be attributed to the dual effects of IRF-1, which enhances ROS levels and inhibits RAD51 expression. Of note, in the IRF/H-GDz/Ca group, the intensity of γ-H2AX fluorescence significantly increased by 110.5%, owing to the synergistic effects of downregulating GLUT-1 expression and IRF-1introduction. The damage to the cell nucleus for different groups was also observed in the bio-TEM images (Supplementary Fig. 17). Additionally, within the SNU216 cell line, we observed that IRF/H-GDz/Ca materials efficiently mediated the expression of γ-H2AX (Supplementary Fig. 16b). Therefore, our rationally designed IRF/H-GDz/Ca MNF demonstrated significant power for ROS augmentation triggered DNA damage.

Finally, we conducted cell toxicity experiments in both N87 and SNU216 cell line, and the results (Fig. 4l, m) indicate that after 24 h of treatment, IRF/H-GDz/Ca MNF material exhibited stronger cytotoxicity compared to other materials, including H-MGDz/Ca, H-GDz/Ca, and the IRF/H-MGDz/Ca group. Therefore, owing to the effective GLUT-1 silencing facilitated by GLUT-1 DNAzyme and the enhanced nuclear damage effect mediated by IRF, IRF/H-GDz/Ca MNF demonstrates heightened potential for therapeutic applications.

Encouraged by the robust inhibitory effects of IRF/H-GDz/Ca MNF materials in gastric cancer cell experiments, their tumor-targeting specificity and anti-tumor efficacy were further evaluated in vivo. In the initial step, we first tested the blood circulation time of MNF materials in vivo. Following intravenous administration, the initial rapid decline in serum concentration is ascribed to drug distribution, defining the distribution half-life ($T_{1/2}$, α). Subsequently, after the completion of distribution, a relatively slow descent rate in serum concentration occurs, characterized by the elimination half-life ($T_{1/2}$, β). We calculated the elimination half-lives ($T_{1/2}$, β) by fitting the experimental data using GraphPad Prism 10, two phase decay model (Supplementary Fig. 18). The results indicate that the elimination half-life of IRF/H-GDz/Ca and IRF/MH-GDz/Ca in vivo is approximately 600 min. It is noteworthy that 94 to 97% of a drug will be eliminated after 4 to 5 elimination half-lives, which means the plasma concentrations of a given drug will be below a clinically relevant concentration and thus will be considered eliminated[55,56]. Therefore, at approximately 3000 min, which is around 50 h, the MNF is essentially cleared from the bloodstream. Thus, selecting time points around and above 50 h will be suitable for the observation of stable drug accumulation in each organ without significant non-specific signals from transient drug residues in the body.

As illustrated in Fig. 5a and Supplementary Fig. 19, the in vivo distribution experiment of the MNF materials in mice was conducted for 96 h, and the MNF were labeled with Cy5.5/BHQ-3 probes. Of note, the functionality of Cy5.5/BHQ-3 probes primarily serves to validate the binding between GDz and GLUT-1, rather than achieving specific imaging of the tumor.

The results revealed that MNF NPs constructed with the mutated HER-2 sequence (IRF/MH-GDz/Ca) exhibited substantial kidney accumulation at 48h time point, which reflected from the lateral and prone positions results. This selective accumulation could be attributed to the inherent renal interception of negatively charged nanomaterials[57,58], or negatively charged oligonucleotides[57,59,60]. Conversely, it was evident that IRF/H-GDz/Ca exhibited substantial accumulation in the tumor region within the stomach, and fluorescence signals were observed even at 96 h time point. This result highlighted the remarkable role of the HER-2 aptamer within the MNF material, facilitating targeted delivery of the MNF to the tumor site.

At the designated 48 and 96-hour time points, the mice were euthanized, and ex-vivo analysis was conducted to examine the fluorescence distribution of the nanodevice within various organs (Fig. 5b and Supplementary Fig. 19). At 48h time point, consistent with the findings from in vivo imaging, the accumulation of MNFs was prominently observed in the kidney, particularly in the IRF/MH-GDz/Ca group, with the kidneys exhibiting the highest accumulation degree. Notably, the accumulation of IRF/H-GDz/Ca NPs in the kidneys was reduced by approximately 50% compared to the IRF/MH-GDz/Ca group. Additionally, the fluorescence intensity in the tumor region of the IRF/H-GDz/Ca group was found to be 2.15 times higher than that of the IRF/MH-GDz/Ca group. At the 96-hour time point, we observed only fluorescence signals in the tumor, and no fluorescence signals were detectable in the kidneys. This indicates that the material can indeed accumulate at the tumor site and be completely cleared from the body without causing major organ burden. Besides, these ex-vivo findings further emphasized the enhanced tumor-targeting ability of the nanomaterials achieved through the incorporation of the HER-2 sequence.

Furthermore, tumor sections were prepared to examine the Cy5.5 fluorescence intensity of the nanomaterials (in cyan) within the tumor tissue (Fig. 5c and Supplementary Fig. 20). The results suggested that the tumor tissue in the IRF/H-GDz/Ca NPs group exhibited a significant fluorescence signal from the nanomaterials, indicating the successful binding of GDz to GLUT-1 mRNA for gene regulation. In contrast, only a minimal amount of Cy5.5 fluorescence was observed in the IRF/MH-GDz/Ca NPs group, further emphasizing the role of the HER-2 segment in enhancing the targeting capability of the nanosystem.

Afterwards, to further validate the therapeutic efficacy of our designed MNF materials, in vivo antitumor experiments were conducted, where the luciferin fluorescent intensity was used to monitor the progression of orthotopic tumor growth (Fig. 5d). By calculating the average radiance, we observed that the tumor size gradually increased over time in the control group and H-MGDz/Ca group (Fig. 5e), and the luciferin fluorescence intensity at day 21 was approximately 2.5 times higher than that of the first day (Fig. 5f). In contrast, the H-GDz/Ca group and IRF/H-MGDz/Ca group achieved significant tumor reduction, with approximately 60% and 75% decrease in tumor size, respectively (Fig. 5e). Furthermore, in the final treatment group, IRF/H-GDz/Ca, the tumor was nearly completely eradicated at day 21, exhibiting a remarkable tumor shrinkage of over 90% (Fig. 5f).

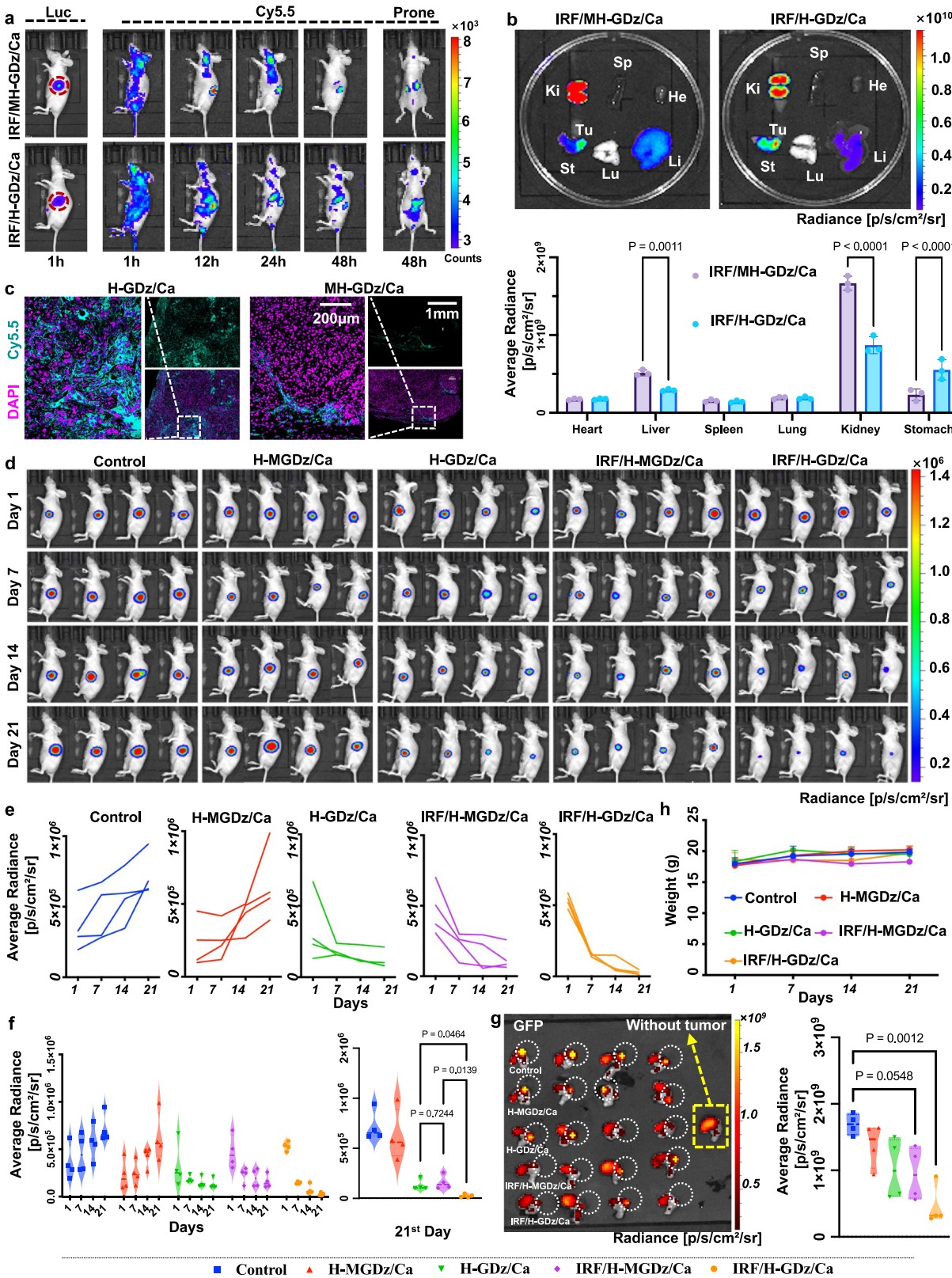

The tumor tissue slides showed all therapeutic groups contained sufficient amount of MNFs inside the tumor (Supplementary Fig. 8), and the remarkable therapeutic outcome can be attributed to the synergistic effects of GLUT-1 inhibition and the increase in IRF-1, which leads to downregulation of RAD51 expression (Fig. 6a).

Later, eGFP was utilized for ex-vivo visualization of the gastric tumor (Fig. 5g and Supplementary Fig. 21). The ex-vivo eGFP

fluorescence imaging results of the gastric tumor were consistent with the in vivo fluorescence results, and the IRF/H-GDz/Ca showed best therapeutic effect. Different functional sequences were successfully combined to construct MNF nanomaterials through a one-step method at room temperature, and each of their function were verified. These results demonstrated the flexibility and effectiveness of DNA sequence design in MNF construction,

**Fig. 5 | Distribution and therapeutic efficacy of nanomaterials in mice.** To focus on the regulatory capacity of GLUT-1 and the functionality of IRF-1, five groups including Control, H-MGDz/Ca, H-GDz/Ca, IRF/H-MGDz/Ca, and IRF/H-GDz/Ca group were carried out. **a** Intravenous administration of MNF materials and their distribution in mice ($n = 3$ independent experiments with similar results). **b** Imaging of the materials in ex vivo organs, including the heart, liver, spleen, lungs, kidneys, and stomach ($n = 3$ independent experiments with similar results). **c** Imaging of the materials in the tumor site of mice, DAPI staining was employed to label the cell nuclei ($n = 3$ independent experiments with similar results). **d** In vivo antitumor experiment in mice. **e** Time-dependent changes in Average Radiance in the tumor site of individual mice. **f** Violin plots illustrating the average changes in Average Radiance in the tumor site for each group of mice. **g** eGFP fluorescence imaging and quantification of gastric tumor in mice. Spontaneous fluorescence was detected from the surrounding gastric contents. Therefore, to ensure accurate analysis, the examination specifically we focused on the circled tumor region. **h** Body weight changes during the treatment period. In (**b**, **h**), data are presented as mean ± SD. Statistics (**b**, **f**, **g**) were calculated using one-way ANOVA using a Tukey post hoc test. The source data from (**b**, **e**–**h**) are provided as a Source Data file.

laying a solid foundation for further development of MNF materials.

Following that, the safety profile of each group was thoroughly assessed. Firstly, the body weight changes of the mice during the treatment period were closely monitored, and no significant differences were observed between the treatment groups and the control group, indicating the absence of adverse effects on the overall health of the animals (Fig. 5h). Additionally, histological analysis of vital organs using H&E staining, and GLUT-1 staining of liver and kidney was conducted to evaluate potential tissue damage or abnormalities (Supplementary Fig. 22 and Fig. 6b). Remarkably, the results of H&E staining demonstrated the normal morphology and histology of the organs, and no difference of GLUT-1 expression in both the treatment and control groups, further supporting the safety of the formulations.

To further enhance the translational potential of MNF nanomaterials, we established a subcutaneous Patient-Derived Xenograft (PDX) tumor model. The HER-2 positive characteristics of the tumor were first verified (Supplementary Fig. 23). Then, tumor inhibition experiments on the previous materials were conducted (Fig. 6c and Supplementary Fig. 24), and the results indicated that the final formulation, IRF/H-GDz/Ca, could significantly inhibit tumor growth, and the body weight of mice in all groups did not decrease.

Subsequently, we investigated the intratumoral levels of ROS (Fig. 6d). We observed that the IRF/H-GDz/Ca group exhibited the highest ROS content within the tumor. This can be attributed to the inhibition of GLUT-1 and the ROS generation and GSH depletion mediated by IRF-1[54].

Tumor cell nucleus damage is the most indicative parameter of the efficacy of our designed MNF material. We performed continuous slicing on tumors from each group and utilized immunofluorescence staining to label the nuclear damage marker γ-H2AX, as depicted by the yellow signal in Fig. 6e. The results indicate that IRF/H-GDz/Ca MNF NPs can mediate comprehensive nuclear damage throughout the three-dimensional tumor, providing strong evidence of the therapeutic effectiveness of the material.

On the other hand, the regulation of the tumor microenvironment is crucial for the treatment of PDX tumor models. Cancer-associated fibroblast (CAF) cells in the tumor were labeled using α-smooth muscle actin (α-SMA) (green signal) and fibroblast activation protein (FAP) (yellow signal). We observed that the inhibition of GLUT-1 downregulates the expression of CAF cells; however, IRF-1 does not inhibit CAF (Fig. 6f). This indicates that our designed MNF materials can achieve the inhibition of CAFs by suppressing GLUT-1, thereby enhancing therapeutic efficacy.

Moreover, it has been reported that the presence of calcium ions and IRF-1 in our IRF/H-GDz/Ca MNF material can inhibit P-glycoprotein (P-gp), thereby addressing drug resistance. Specifically, the accumulation of calcium ions induces mitochondrial damage, leading to reduced ATP synthesis and subsequent P-gp suppression. Additionally, IRF-1 itself has been reported to possess inhibitory effects on P-gp. Therefore, we conducted immunofluorescence staining experiments for P-gp (Supplementary Fig. 25) and observed that the H-MGDz/Ca group showed limited P-gp inhibition, possibly due to insufficient calcium ions to damage mitochondria. In contrast, the H-GDz/Ca group significantly enhanced mitochondrial inhibition, leading to

substantial P-gp downregulation. Notably, the IRF/H-MGDz/Ca and H-GDz/Ca groups had similar P-gp downregulation abilities, with IRF/H-GDz/Ca showing the strongest effect among all formulations. This highlights IRF-1's potent P-gp inhibitory capability.

Finally, we further investigated the safety of the MNF material in vivo. We assessed the blood biochemistry markers for the constructed PDX model, focusing on ALT (Alanine Aminotransferase), AST (Aspartate Aminotransferase), TBIL (Total Bilirubin), and blood creatinine[61]. Elevated ALT and AST levels indicate liver cell damage, while increased TBIL levels suggest issues with liver function and hemolysis. Blood creatinine elevation signifies impaired kidney function. Our nanomaterials (H-MGDz/Ca, H-GDz/Ca, IRF/H-MGDz/Ca, IRF/H-GDz/Ca) showed no significant differences in these markers compared to the healthy untreated group, indicating no adverse effects on physiological health (Supplementary Fig. 26a). In addition, in vitro hemolysis experiments confirmed that our nanomaterials do not cause blood cell rupture (Supplementary Fig. 26b).

In summary, the results from the PDX model demonstrate that our MNF system, loaded with IRF-1, significantly increases ROS levels in tumor tissues, enhances cellular nuclear damage, and exhibits no harm to normal tissues.

## Discussion

The key advantage of MNFs over MOFs lies in the multifunctionality of the DNA sequences compared to the organic ligands. DNA sequences within MNFs can act as nucleic acid drugs with therapeutic functions, construct adapter sequences for targeted cell binding, and form molecular fluorescence beacons for disease gene diagnosis. Therefore, utilizing MNF materials as carriers for drug delivery not only leverages the functionality of the drug itself but also harnesses the multifunctionality of the DNA sequences simultaneously. However, it is imperative to note that MNFs may suffer from certain limitations, including stability concerns compared to MOFs and challenges associated with achieving rapid synthesis at room temperature.

Our work has revealed an improvement in the synthesis of MNF systems by elucidating the crucial role of DNA sequence length, and also unlocked the potential for loading large-molecule active drugs into these advanced nanomaterials. The observed length related results can be attributed to the inherent characteristics of oligonucleotides, which are known to lack strong base stacking forces and exhibit flexible spatial conformations[18]. Consequently, increasing the length of oligonucleotide sequences (which is different from augmenting the nucleotide concentration), holds the potential to enhance the folding and entanglement probability of the sequences during the reaction process[39,40].

The remarkable design flexibility offered by longer DNA sequences empowers us to fully exploit the capabilities of DNA nanotechnology, as demonstrated through the incorporation of GLUT-1 mRNA-targeting DNAzymes, HER-2-targeting DNA aptamers, and molecular beacons, to bestow functional prowess upon the resulting MNF constructs. By embarking on the path of assembling complex DNA materials with metal ions, we have achieved enhanced DNA damage for effective treatment of gastric cancer. Furthermore, we have validated the crucial functional activity of aptamers in

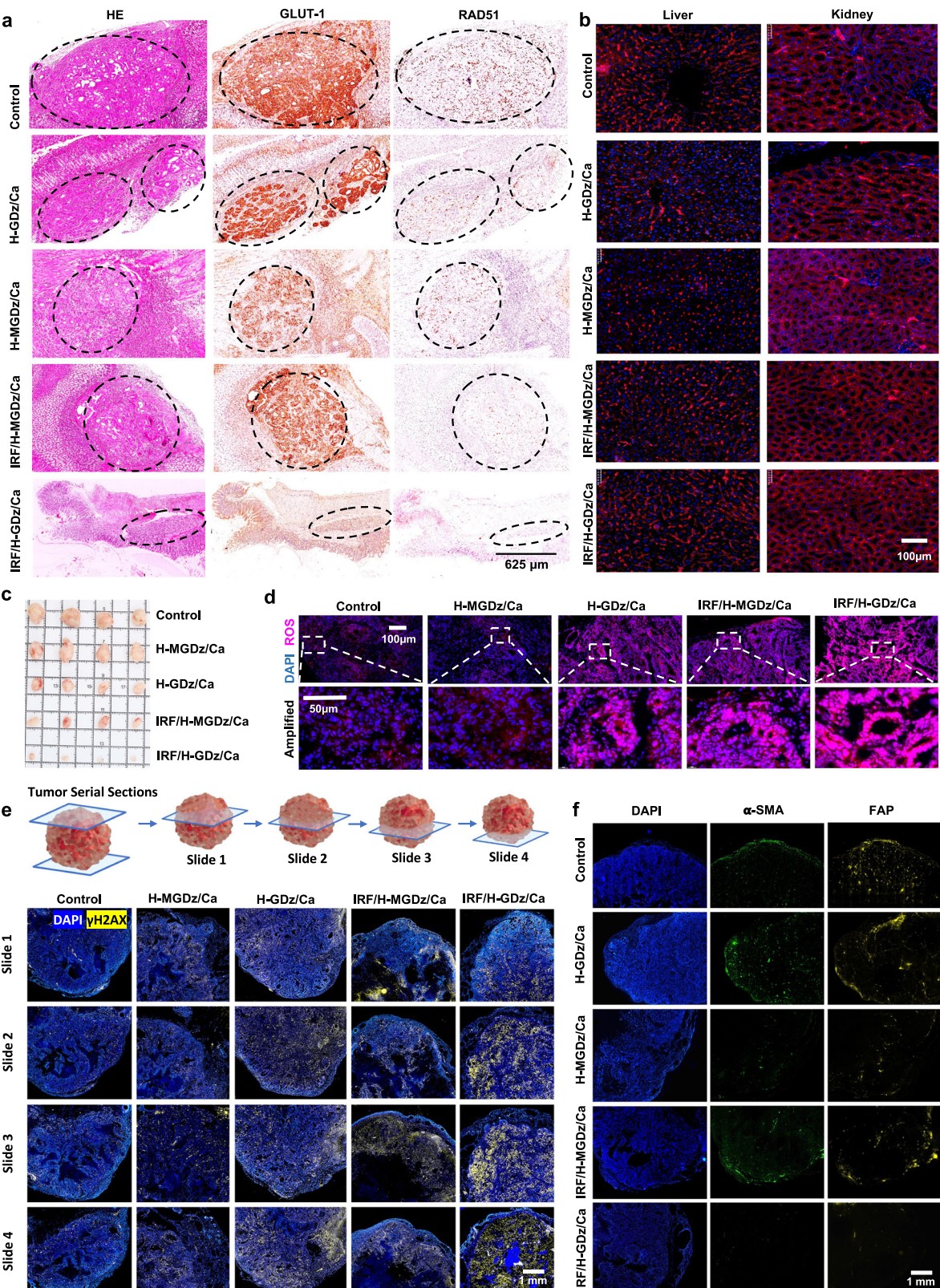

**Fig. 6 | The protein regulation in the in-situ tumor model and the characterization of MNF materials in the PDX model, along with their therapeutic efficacy. a** Immunohistochemical characterization of the expression regulation of GLUT-1 and RAD51 in different treatment groups ($n = 3$ independent experiments with similar results). **b** Expression of GLUT-1 in liver and kidney tissues ($n = 3$ independent experiments with similar results). **c** Therapeutic efficacy of different treatment groups in the PDX model ($n = 4$ mice). **d** Representation of ROS in tumor slice tissues stained with dihydroethidium (DHE), marked in red ($n = 3$ independent experiments with similar results). **e** Characterization of overall three-dimensional nuclear damage in tumor slices ($n = 3$ independent experiments with similar results). **f** Labeling of CAF cells in tumor slices ($n = 3$ independent experiments with similar results).

constructing MNF materials, laying the foundation for flexible design of DNA sequences.

From the perspective of constructing GLUT-1 molecular beacons, the preparation of DNAzyme molecular beacons served three main purposes: confirming their target substrate recognition ability, demonstrating the sustained activity of functionalized DNAzymes in MNF materials' post-cellular internalization, and leveraging DNA nanotechnology for multifaceted MNF applications. While GLUT-1 is present in many cell types, our study provides a practical framework for detecting low-abundance genes in future MNF applications.

From a therapeutic perspective, the current scientific landscape is rich with endeavors to enhance ROS production within the tumor microenvironment for robust anticancer effects[62,63]. In alignment with this research enthusiasm, our simplified MNF material presents distinctive advantages for ROS-induced tumor therapy. First, the MNF boasts a carrier-free therapeutic architecture, integrating therapeutic DNA functional sequences with mitochondrial-disrupting metal ions, thus avoiding the introduction of non-therapeutic components. Second, leveraging the innate targeting proficiency of DNA nanotechnology, MNF requires no exogenous cellular targeting modifications, streamlining the therapeutic approach. Third, protein-loading capability of MNF facilitates the IRF-1 loading and inhibition of nuclear repair proteins RAD51, leading to a significant enhancement of ROS-mediated nuclear DNA fragmentation and thereby amplifying the ROS-driven therapeutic cascade. Besides, we found the MNF metabolism and clearance pathway is kidney based, which could be attributed to the inherent renal interception of negatively charged nanomaterials or negatively charged oligonucleotides. More importantly, our research heralds a new era for MNF materials, brimming with extraordinary possibilities and unparalleled impact in various research and application domains.

Of note, there still exists significant potential for further exploration in the application of DNAzymes outside the scope of this experiment. Several aspects remain unresolved: (1) While DNAzymes are currently predominantly utilized for regulating intracellular mRNA and miRNA, their potential to modulate double-stranded DNA in the nucleus warrants deeper investigation. (2) This study focused on a comprehensive examination of one parameter, DNA length, and its impact on MNF construction. Yet, the influence of other factors, such as the type and proportion of composite metal ions, on the stability of MNF materials remains unknown. Therefore, the future biomedical field offers ample opportunities for continued exploration and refinement of DNAzyme-based MNF materials.

## Methods

### Ethical statement
Our study strictly followed ethical guidelines, with all animal procedures conducted in accordance with the Guidelines for the Care and Use of Laboratory Animals. Approval was obtained from the Institutional Animal Care and Use Committee (IACUC) of Zhejiang Center of Laboratory Animals (ZJCLA), under the reference number ZJCLA-IACUC-20010304. Approval for human experiment was granted by the Ethics Committee of Fudan University Shanghai Cancer Center (Approval No.:050432-4-2108*). Written and informed consent was procured from patient before the collection of tissue samples. Tumor tissue specimens obtained from patients who had not received any treatment other than surgery before the operation were exclusively utilized. Postoperative patient treatment followed standardized care and was unrelated to our study design. All experimental procedures adhered to ethical standards and international guidelines, including the "International Ethical Guidelines for Biomedical Research Involving Human subjects" and "Declaration of Helsinki".

### Materials
All oligonucleotides used in this study were synthesized and purified through high-performance liquid chromatography (HPLC) by Sangon Biotechnology Co., Ltd. (Shanghai, China). Detailed information about the oligonucleotides, including sequences and specifications, can be found in Table S1. Ammonium persulfate (APS) and N, N, N, N-tetramethylethylenediamine (TEMED), calcium chloride, and sodium dihydrogen phosphate were obtained from Sigma Finland. Tris-HCl buffer solutions (pH 8.8 and 6.8), 30% acrylamide/bis, and TBS/Gly/SDS buffer were purchased from Bio-Rad Laboratories, Inc. Finland. Solutions used in microRNA-related experiments were prepared using diethyl pyrocarbonate (DEPC)-treated water (RNase-free) from Sigma, Switzerland. The BCA kit, RIPA buffer, Tris-Triton buffer, protease inhibitor, EDTA, and GelRed were purchased from Thermo-Fisher, Finland. Anti-mouse PTEN antibody, (Catalog no. ab267787), Anti-Rabbit GAPDH, (Catalog no. ab210113), Anti-Glucose Transporter GLUT1 antibody [EPR3915], Anti-Rad51 antibody [EPR4030(3)], 6-Phosphogluconate Assay Kit (Colorimetric) (Catalog no.ab211071) were provided from Abcam. ATP Determination Kit (A22066) and LysoTracker™ Green DND-26, special packaging, (Catalog number: L7526) were purchased from ThermoFisher. Cysteine Assay Kit (Catalog Number MAK255), NADP/NADPH Quantification Kit (Catalog Number MAK038) were bought from Sigma-Aldrich.

### MNF synthesis with different lengths and mass analysis
DNA sequences are ordered directly from the company, and the freeze-dried powder can be prepared at any concentration by adding the required water according to the company's requirements. Specifically, functional DNA fragments of varying lengths (20 nucleotides, 41 nucleotides, and 83 nucleotides) were designed. For the 83-nucleotide DNA, a 1 mM solution was prepared, resulting in a nucleotide concentration of 83 mM. For the 20-nucleotide DNA, a 4.15 mM solution was prepared to achieve a nucleotide concentration of 83 mM. Similarly, for the 41-nucleotide DNA, a 2.03 mM solution was prepared to obtain a nucleotide concentration of 83 mM. The nucleotide concentration is calculated by multiplying the DNA sequence concentration with the number of nucleotides in the sequence. This approach ensures the precise determination of the nucleotide's concentration during MNF construction. The DNA fragments (2 μL, 1 mM) were used for MNF synthesis by incubating them with 1 M $CaCl_2$ in a reaction volume of 200 μL at room temperature. The total weight of the MNF material was measured after freeze-drying. The presence of calcium (Ca) in the MNF material was detected using a Calcium Assay Kit (Colorimetric) from Abcam (ab102505). The DNA content was determined by subtracting the DNA content in the supernatant from the initial amount of DNA before the reaction. The DNA weight was measured using NanoDrop™ 2000/2000c Spectrophotometers (Thermo Scientific™) .

### Molecular dynamics simulations
To obtain molecular-level understanding about the effect of the length of the oligonucleotides on the MNF synthesis, molecular dynamics (MD) simulations of the studied GDz and H-GDz DNA constructs were carried out. The two DNA constructs were modeled using the 3D Builder panel in the Maestro Molecular Modeling Suite (Schrödinger Release 2023-1, New York, USA). The GDz part was sketched from the given sequence (TTGCACCAGC GAGGCTCTCA GCGAGACGAA ATGAGGTGCAT) in 5'→3' direction using the B-helix conformation. The HER-2 aptamer (GCAGCGGTGT GGGGGCAGCG GTGTGGGGGC AGCGGTGTGG GG) was built to the GDz part to obtain the H-GDz structure. The DNA constructs were modeled so that the same bases were pairing and forming the lock (loop structure) as in reality. These modeled 3D structures were then utilized to prepare two amorphous simulation systems using the Disordered System Builder panel of the Schrödinger Material Science Suite (Schrödinger Release 2023-1, New York, USA). Both systems contained 0.15 M calcium chloride solution and roughly an identical count of nucleotides in explicit single-point-charge (SPC) water. System 1 consisted of ten GDz sequences (10 × 41

nt = 410 nt) while system 2 comprised five H-GDz sequences (5 × 83 nt = 415 nt) (Fig. 1b, c). Moreover, the simulation systems were neutralized by adding an appropriate number of sodium ions. Periodic boundary conditions (PBC) were applied with an orthorhombic unit cell. As a reference, system 2 was simulated also with 0.15 M NaCl instead of $CaCl_2$.

The simulation systems were both submitted to a 100-ns MD simulation with Desmond (Schrödinger Release 2023-1: Desmond Molecular Dynamics System, D. E. Shaw Research, New York, NY, USA, 2023; Maestro-Desmond Interoperability Tools, Schrödinger, New York, NY, USA, 2023)[64], using the OPLS4 force field[65]. Before the production simulation, the systems were equilibrated using Desmond's default relaxation protocol (as implemented in Maestro), which includes short simulations (12–24 ps) in both NVT and NPT ensembles, first at 10 K and then at 300 K and 1.01325 bar pressure, both with and without restrains on solute atoms. The production simulation was then performed for 100 ns at 300 K and 1.01325 bar using the Nose-Hoover chain thermostat[66] and Martyna-Tobias-Klein barostat[67] with isotropic coupling. Long-range Coulombic interactions were handled using the U-series method[68], while a cut-off radius of 9.0 Å was applied for short-range interactions. The Radial Distribution Function (RDF) panel of Maestro was used to calculate and plot the RDF between $Ca^{2+}$ ions and the negatively charged phosphate oxygen atoms in the DNA backbone from the MD simulation trajectories.

## Molecular dynamics simulations

The persistence length (Lp) is determined using the following formula: $\langle h^2 \rangle = 2L_p L_0 [1 - (\frac{L_p}{L0})(1 - \exp(-\frac{L0}{L_p}))]$. Here, $\langle h^2 \rangle$ represents the mean squared end-to-end distance, $L_0$ is the extended chain length, and $L_p$ is the persistence length. The end-to-end distance is computed for each selected chain in every trajectory frame, and the values are then averaged over the entire trajectory.

## MNF NPs particle size investigation

The MNF synthesis process involved a reaction between the DNA fragments (2 μL, 1 mM) and various concentrations of $CaCl_2$ (2 M, 1 M, or 0.5 M) in a 200 μL reaction volume at room temperature. Following the reaction, the particles were subjected to centrifugation and washed three times with Milli-Q water. The resulting particles were then stored in anhydrous ethanol. The size distribution of the nanoparticles was determined using dynamic light scattering by dispersing them in pH = 7.4 PBS buffer. The surface charge of the nanocarrier was assessed using disposable cells and measured using the Zetasizer Nano ZS instrument. For TEM imaging, carbon-coated copper grids were used to carefully deposit droplets of the particle solution, which were subsequently air-dried prior to examination. SEM imaging and EDS analysis were performed using the Thermo Scientific Apreo S field-emission scanning electron microscope equipped with the Oxford Instruments UltimMax 100 energy dispersive X-ray spectrometer.

## Polyacrylamide gel electrophoresis (PAGE) assays

A gel electrophoresis setup was prepared by filling a plate with a shim with a 20% gel solution, ensuring a uniform distribution without any trapped air bubbles. The gel was allowed to solidify at room temperature for approximately 30–60 minutes after the insertion of a comb to create wells. Following coagulation, the samples were carefully loaded into a vertical electrophoresis tank, and an initial run of 10 min at 5 V was conducted to establish optimal gel conditions. Oligonucleotide samples were prepared by combining 2 μl of the sample, 8 μl of water, and 2 μl of loading buffer. To ensure accurate analysis, a minimum of 10 μg of oligonucleotide sample was meticulously loaded into the wells, ensuring precise band visualization. The gel was then subjected to electrophoresis at a voltage of 90 V for approximately 2 h. For band visualization, a diluted GelRed solution (1:10,000) was applied for 5 min. Excess staining was carefully removed by immersing the gel in distilled water for 10 min, effectively reducing background interference.

For different groups, the incubation is performed at 37 °C, for 3 h with 1 μM of H-GDz and 2 μM of substrate under different metal ion concentrations (0, 0.1, 0,5, 1, 2, 4 mM).

## GLUT-1 mRNA detection capability

The H-GDz sequence at a concentration of 100 μM was subjected to incubation with GLUT-1 mRNA for efficient binding and interaction. Various concentrations of GLUT-1 mRNA substrates, ranging from 0.1 μM to 100 μM, were added to the reaction mixture. The mixture was heated to 95 °C and subsequently cooled down to room temperature within 3 min, allowing for annealing of the DNA-RNA complex. The incubation period lasted approximately 1 h. The resulting fluorescence signal was detected using a ChemiDoc MP imaging system (Bio-Rad). The tubes containing the reaction mixture were placed in the imaging system for fluorescence visualization and analysis.

## Cell culture and maintenance

The NCI-N87 cells were cultured and maintained in RPMI 1640 medium supplemented with 10% fetal bovine serum (FBS) at a temperature of 37 °C. The cells were passaged 2–3 times per week when they reached 90-100% confluency. The human gastric cancer cell line NCI-N87 was obtained from ATCC®, CRL-5822 ™. The normal human dermal fibroblasts NHDF cells were purchased from PromoCell® and cultured in DMEM medium with 10% FBS. The human gastric cancer SNU-216 cells were purchased from BOHUI Biotechnology Co., Ltd, and were maintained in RPMI 1640 medium supplemented with 10% fetal bovine serum (FBS). The N87 and NFDH cell lines used were directly used without any authentication. The SNU-216 cells were authenticated through Short tandem repeat (STR) testing, performed by Shanghai biowing applied biotechnology Co., Ltd. In detail, DNA extraction will be performed using the Axygen Genomic DNA Extraction Kit. Subsequently, DNA samples will undergo amplification using the 21-STR amplification scheme, followed by the detection of the STR loci and the gender gene Amelogenin on the ABI3730XL genetic analyzer. Appropriate positive and negative controls were run and confirmed for the sample. All the cell lines tested negative for mycoplasma contamination.

## Lysosome escape

For the assessment of lysosomal escape, NCI-N87 cells were cultured overnight on confocal dishes. H-GDz/Ca NPs labeled with Cy5.5 were utilized in the experiment. The MNF material was initially co-incubated with cells for one hour, and it was replaced with blank culture medium at the 1-hour mark. Subsequently, at 0 h, 1.5 h, and 7.5 h, the lysosomes of the cells were stained for a duration of 30 min. Simultaneous imaging of the drug and lysosomes was performed, with imaging time points recorded as 30 min, 2 h, and 8 h. Lysosomes were labeled with LysoTracker™ Green DND-26, special packaging, Catalog number: L7526, exhibiting green fluorescent signals. DNA sequences within MNFs were labeled with Cy5.5, presenting red signals. Cell nuclei were stained with DAPI, appearing blue. The cells were examined using a fluorescence confocal microscope. ImageJ software was employed to carry out the topological analysis of fluorescence intensity along the marked white lines, which provided insights into the degree of overlap between the NPs and lysosomes in each experimental group.

## GLUT-1 detection and down-regulation

The cells were seeded in 6-well plates at a density of $1 \times 10^5$ cells per well and incubated overnight to allow for attachment. Once the cells were attached, the growth media was replaced with different solutions corresponding to the respective treatment groups: free medium

group, H-GDz + Ca$^{2+}$ group, MH-GDz + Ca$^{2+}$ group, H-GDz/Ca group, and MH-GDz/Ca group. The concentration of GDz in all therapeutic groups was maintained at 2 μM, and the cells were incubated at 37 °C. After 10 h of incubation, the cells were observed using confocal laser scanning microscopy (CLSM). NPs were under FAM channel (470/525 nm) and GLUT-1 were under TRITC channel (557/576 nm). Additionally, the cells were detached using trypsin, washed with PBS, and subjected to flow cytometry analysis using a BD LSRFortessa flow cytometer (BD Biosciences). The acquired data were analyzed using Flowjo_V10 software, with gating set to include only live cells, and a total of 10,000 cells were recorded per sample. NPs were under FAM channel (470/525 nm) and GLUT-1 were under PE channel (566/574 nm).

## Western blot assay

Following a 24-h incubation of the cells with various nanomaterial groups, RIPA lysis buffer (2 mL) was added to lyse the cells. The lysis buffer contained 20 μl of 1% protease inhibitor and 20 μl of 1% 0.5 M EDTA per mL of RIPA. After 1 h of lysis, the lysate was centrifuged at 8050 x *g* for 10 min, and the supernatant was collected for further analysis. Cell proteins were separated by SDS-PAGE gradient gel and transferred onto a PVDF membrane. Protein loading buffer was added to the samples, and they were thoroughly mixed and denatured at 98 °C for 10 min. GLUT-1 protein was not denatured since Abcam's instructions mention that GLUT-1 protein may aggregate irreversibly after boiling. Avoiding boiling helps prevent GLUT-1 proteins from entering the gel or forming polymers that are often mistaken for background. Our experimental design takes this factor into consideration to ensure the accuracy and reliability of the experimental results. The PVDF membranes were incubated overnight at 4 °C with primary antibodies against GLUT-1 (dilution: 1:100000; Abcam, Cambridge, UK), RAD51 (Abcam, ab133534, 1: 1000 dilution), and β-Actin (dilution: 1:1000; Abcam, Cambridge, UK). Subsequently, the membranes were incubated with the corresponding secondary antibodies at 37 °C for 1 h. Protein bands were visualized using the GelDoc Go Gel Imaging System (Bio-RAD).

## IRF loading and drug releasing

To investigate the loading capacity of IRF-1 in IRF/H-GDz/Ca NPs, various loading amounts of IRF-1 (0, 1, 10, 100 ng/mL) were used to prepare the NPs. Cell culture systems with different concentrations of IRF-1 were established for comparison. The IRF-1 content in the NPs was measured using an IRF-1 testing kit. The loading content of IRF-1 was determined by subtracting the supernatant IRF-1 content from the input IRF-1 content. For drug release experiments, 1 mL of IRF/H-GDz/Ca NPs (containing 2 μM H-GDz and 2 μg IRF-1) were placed in 1 mL tubes. The tubes were continuously shaken during the release process at 37 degrees Celsius. Two different pH conditions, pH 7 and pH 6, were tested for drug release. At each time point, the samples were centrifuged, and 0.1 mL of the release medium was collected from the supernatant for drug concentration measurement. The collected medium was then added back to the system to maintain the volume and conditions for continuous drug release assessment.

## Ca$^{2+}$ concentration level visualization

NCI-N87 cells were cultured in confocal dishes at a density of $1 \times 10^5$ cells per well. Upon reaching 70–80% confluency, the cells were exposed to different treatment conditions: Control, H-MGDz/Ca, H-GDz/Ca, IRF/H-MGDz/Ca, and IRF/H-GDz/Ca, for an incubation period of 16 h at 37 °C. Subsequently, the cell medium was supplemented with Fluo-3 AM solution at a concentration of 1.0 μM. Fluorescence intensity analysis was conducted using ImageJ software to quantitatively evaluate the cellular response.

## Mitochondrial membrane potential detection by tracking JC-1 monomer and aggregates

In order to evaluate alterations in mitochondrial membrane potential (ΔΨm), NCI-N87 cells were cultured in 6-well plates with a density of $2 \times 10^5$ cells per well, using 2.0 mL of FPS-DMEM as the growth medium. Following a 10-h incubation period, the cells were exposed to different treatment conditions including Control, H-MGDz/Ca, H-GDz/Ca, IRF/H-MGDz/Ca, and IRF/H-GDz/Ca, all at a fixed GDz concentration of 2 μM, for a duration of 12 h. Subsequent to two washes with PBS, the cells were subjected to JC-1 staining and analyzed using confocal laser scanning microscopy (CLSM). For JC-1 staining, a mixture of 1.0 mL JC-1 working solution and 1.0 mL DMEM was added to the cell culture.

## MPTP detection by tracking Calcein AM fluorescence

To assess the opening of the mitochondrial permeability transition pore (MPTP), NCI-N87 cells were plated in a 6-well plate at a density of $2 \times 10^5$ cells per well in 2.0 mL of FPS-DMEM and incubated for 10 h. Subsequently, the growth medium was replaced with 1.0 mL of different treatments including Control, H-MGDz/Ca, H-GDz/Ca, IRF/H-MGDz/Ca, and IRF/H-GDz/Ca, all at an equivalent GDz concentration of 2 μM, and incubated for 12 h. After two washes with PBS, the cells were stained with a mixture of Calcein AM and CoCl$_2$ and analyzed using confocal laser scanning microscopy (CLSM).

## Bio-TEM observation of mitochondrial morphology

To investigate the mitochondrial morphology, NCI-N87 cells were cultured in a 100 mm × 20 mm dish with a density of $3.0 \times 10^6$ cells per dish for a duration of 24 h. Subsequently, the culture medium was replaced with 2.0 mL of various treatment groups, including Control, H-MGDz/Ca, H-GDz/Ca, IRF/H-MGDz/Ca, and IRF/H-GDz/Ca, and incubated for an additional 24 h. After two washes with PBS, the cells were subjected to trypsinization and centrifugation at 200 x *g* to collect the cellular pellet. The collected cells were then embedded, and ultrathin sections were obtained using a Leica EM UC7 ultramicrotome. These sections were subsequently placed on copper grids and stained with uranyl acetate and lead citrate to enhance contrast. The sections were examined using a JEM-1400 Plus Transmission Electron Microscope (TEM) at a magnification of 5000-10000x, capturing at least three random images. The ultrastructure of the mitochondria was analyzed based on the presence or absence of cristae and the morphology of the mitochondrial membrane.

## Metabolic key products characterization

The 6-Phosphogluconate (6-PGA) Assay Kit from Abcam (ab211071) was utilized in the experiment. Following the washing of $3 \times 10^5$ cells with PBS, they were resuspended in 100 μL of extraction buffer for on-ice lysis and subsequent centrifugal extraction. After adding the reaction mixture, cells were incubated at 37 °C for 1 h. The optical density (OD) at 450 nm was measured using an enzyme reader, and the 6-PGA content was determined through a standard curve. For the Cysteine Assay, the kit from Sigma (Catalog Number MAK255) was employed. After washing $3 \times 10^5$ cells with PBS, they were resuspended in 100 μL PierceTM RIPA Buffer for on-ice lysis and centrifugal extraction. The OD was measured at room temperature with excitation at 365 nm and emission at 450 nm. The Cysteine content was determined using a standard curve. The NADP/NADPH Quantification Kit from Sigma (Catalog Number MAK038) was used for the NADP/NADPH assay. After washing $3 \times 10^5$ cells with PBS, they were resuspended in 800 μL extraction buffer, lysed on ice, and centrifugally extracted. The OD at 450 nm for both total NADP and NADPH was measured. Subsequently, the obtained values were applied to their respective standard curves to determine the content of total NADP and NADPH. Finally, the NADP/NADPH ratio was calculated.

## GSH detection

For each experimental group, NCI-N87 cells were distributed evenly in culture dishes and cultured until they reached 80% cell density. The Control group was left untreated, while the remaining groups received different treatments including H-MGDz/Ca, H-GDz/Ca, IRF/H-MGDz/Ca, and IRF/H-GDz/Ca. After a treatment period of 16 h, cells from each group were collected and washed with phosphate-buffered saline (PBS). The extraction of glutathione (GSH) was carried out using a GSH/GSSG Ratio Detection Assay Kit (Fluorometric - Green) (ab138881), following the manufacturer's instructions. The GSH samples extracted from each group were then quantified by measuring the absorbance at an excitation/emission wavelength of 490/520 nm, and the GSH concentrations were determined using a standard curve.

## ROS detection

Upon reaching 70−80% confluence, NCI-N87 cells were cultured in confocal dishes at a density of $1 \times 10^5$ cells per well. The cells were then exposed to various experimental conditions, including the control, H-MGDz/Ca, H-GDz/Ca, IRF/H-MGDz/Ca, and IRF/H-GDz/Ca groups, and incubated at 37 °C for 10 h. After 6 h of incubation, the cell medium was supplemented with dichlorodihydrofluorescein diacetate (DCFH-DA) solution. Quantitative analysis of the reactive oxygen species (ROS) signal was performed using Image J software, allowing for the assessment of intracellular $^1O_2$ levels under different experimental conditions.

## γ-H2AX detection

NCI-N87 cells were cultured in confocal dishes at a density of $1 \times 10^5$ cells per well. The culture medium was aspirated, and the cells were washed with PBS. Subsequently, they were fixed with 1 mL fixation solution and incubated for 10 min. Afterward, the fixation solution was aspirated, and the cells were washed three times with the washing solution for 3−5 minutes each time, ensuring residual liquid was removed while keeping the sample surface slightly moist. Following the washing steps, the cells were treated with the blocking solution for 10−20 min at room temperature. The blocking solution was then aspirated, and the cells were incubated with the γ-H2AX rabbit monoclonal antibody. After incubation, the antibody was carefully aspirated and the cells were subsequently washed again, followed by incubation with the anti-rabbit 488 antibody. After the final wash, the cells were stained with the cell nucleus dye (DAPI) for approximately 5 min.

## Cell toxicity assay

The efficacy of MNF materials on gastric cancer cells (N87 and SNU216 cell lines) was determined through WST-1 cell viability assay. N87 cancer cells and SNU216 cells were seeded in a 96-well plate at a density of 3000 cells per well and incubated in cell growth medium overnight at 37 °C with 5% $CO_2$. On the following day, the cell growth medium was replaced with fresh medium containing different concentrations of H-MGDz/Ca, H-GDz/Ca, IRF/H-MGDz/Ca, and IRF/H-GDz/Ca MNF, and cells were cultured for 24 h. All dilutions for cell viability assessment were prepared in cell growth medium. After incubation, 10 μL of WST-1 reagent was added to each well, and the cells were incubated for an additional 2 h at 37 °C with 5% $CO_2$. Absorbance at 440 nm was measured using a multimode reader (Thermo Scientific Inc., Waltham, MA, USA).

## Gastric cancer orthotopic model

First, a subcutaneous tumor model was established by injecting $1 \times 10^6$ N87 gastric cancer cells labeled with Luciferase/GFP, into the BALBc-nu strain (Female, 6−8 weeks old). Once the tumor volume reached approximately 100 mm³, the tumor was excised and cut into 1 mm³ tumor tissue blocks. Nude mice of the BALBc-nu strain (6−8 weeks old) were anesthetized and positioned in the left lateral decubitus position. After standard skin disinfection, a 1 cm oblique incision was made below the right costal margin to access the abdominal cavity. Subsequently, the abdominal cavity was opened to expose the gastric organs. Tumor tissue blocks were sutured to the gastric surface to minimize tissue damage. If necessary, hemostasis was performed, and the incision was sutured to close the abdominal cavity. Finally, the surgical site was disinfected again, and the mice were placed in a warm incubator or recovery area for postoperative care, adhering to ethical and animal care guidelines. The mice were maintained under specific conditions: a 12/12 day/night cycle, humidity at 50−60%, and a temperature of 22−26 °C. The tumor-bearing mice were randomly allocated into five groups once the tumor signal can be clearly seen. While female animals were selected for the experiments, the results are not gender-restricted. Humane endpoints include conditions such as tumor burden exceeding 10% of normal body weight, weight loss in animals exceeding 20% of their normal weight, and persistent self-harm by the animals. When humane endpoints are reached (when mice around 13−15 weeks in this study), euthanasia is performed through cervical dislocation under deep anesthesia. We confirmed that the tumors in the in vivo experiments did not exceed the maximum tumor size.

## Patient-derived xenografts

We obtained fresh gastric cancer tumor tissue from patients undergoing clinical surgery. The tumor tissue was placed in a tissue storage solution and transported at 4 °C. Tumor modeling was initiated within 2 h. During modeling, necrotic tumor tissue was first removed, and the tumor was cut into small pieces of 3 mm × 3 mm x 3 mm. The tumor tissue blocks were implanted subcutaneously in the right anterior shoulder of the mice. All PDX experiments were conducted in 6−8-week-old female NOG (NOD.Cg-Prkdc$^{scid}$ IL2rg$^{tm1Sug}$/JicCrl) mice provided by Beijing Vital River Laboratory Animal Technology Co., Ltd. Tumor growth was observed daily to obtain the first-generation gastric cancer-bearing mice. When the subcutaneous tumors in NOG mice grew to approximately 300 mm³, the skin was incised, and the subcutaneous tumors were dissected. Part of the tumor tissue was used as a source for passaging. This process was repeated until stable third-generation PDX-bearing mice were obtained for subsequent experiments. The gastric cancer tumor tissue was validated for HER-2 positive characteristics through immunohistochemistry. All mice were housed and bred in a specific pathogen-free (SPF) facility with controlled temperature (22 °C) and lighting (12:12 h light-dark cycle). The tumor-bearing mice were randomly assigned to five groups when the tumor volume reached approximately 100 mm³. The mice had ad libitum access to food. The experiment was approved by the Ethics Committee of Fudan University Shanghai Cancer Center (Approval No.: 050432-4-2108*), and informed, written consent was obtained from patients before collecting tissue. Ethical considerations include defining humane endpoints, which encompass scenarios where the tumor burden exceeds 10% of the normal body weight, weight loss surpasses 20% of the normal weight, or persistent self-harm by the animals is observed. In such cases, euthanasia is carried out through cervical dislocation under deep anesthesia. We confirmed that the tumors in the in vivo experiments did not exceed the maximum tumor size.

## Pharmacokinetics study

In the pharmacokinetics investigation, 6−8-week-old female BALBc-nu mice were randomly allocated into two groups ($n = 3$ mice) and subjected to intravenous administration of distinct formulations: (i) IRF/H-GDz/Ca, (ii) IRF/MH-GDz/Ca. All administered at a dosage of 500 nmol/kg. Blood samples (20 μL) were retro-orbitally withdrawn at predetermined intervals, and hemostasis was promptly applied to mitigate bleeding. Blood samples, upon withdrawal, were

immediately dissolved in 0.1 mL of lysis buffer (1% Triton X-100) through gentle sonication. Cy5.5-siRNAs were subsequently extracted by incubating the blood lysis samples in 0.5 mL of DMSO at room temperature overnight. The Cy5.5 level in the supernatant was quantified by measuring the fluorescence intensity using a microplate reader (Ex = 683 nm, Em = 703 nm).

## In vivo biodistribution

For the assessment of nanodevice biodistribution, the IRF/H-GDz/Ca and IRF/MH-GDz/Ca groups were administered intravenously. The therapeutic dosage in each group was adjusted to GDz: 500 nmol/kg. In vivo imaging was conducted at multiple time points, including 1 h, 12 h, 24 h, 48 h and 96 h after administration. On the 48th and 96th hour, the mice were sacrificed, and the tumor, heart, liver, spleen, lung, and kidney were collected for subsequent ex-vivo fluorescence analysis.

## In vivo anticancer efficacy

The mice were categorized into five groups, namely control, H-MGDz/Ca, H-GDz/Ca, IRF/H-MGDz/Ca, and IRF/H-GDz/Ca groups. The therapeutics were administered at a concentration equivalent to GDz: 500 nmol/kg through tail vein injection. After a 21-day treatment period, the tumors connected to the stomach were collected for comparative analysis.

## GLUT-1, RAD51, α-SMA, FAP, P-gp staining

After gently blotting the tissue slices to remove excess moisture, a histochemical pen was used to outline the tissue and create a barrier to prevent liquid from spreading. Proteinase K was applied within the outlined area to fully cover the tissue and incubated at 37 °C for 25 min. The slides were then placed on a decolorizing shaker and subjected to three washes in PBS (pH 7.4) with 5 min of shaking for each wash. Subsequently, the slides were transferred to a fresh container with PBS (pH 7.4) for an additional three washes on the decolorizing shaker. For GLUT-1 staining, the diluted GLUT-1 antibody (1:100-1:500) was carefully added to the tissue sample. After thorough washing with PBS three times, the slides were incubated with the secondary antibody for 60 min. Similarly, for RAD51 staining, the tumor sections were treated with the anti-RAD51 antibody. Following three washes with PBS (pH 7.4) for 5 min each, the slides were incubated with the secondary antibody, for 60 min. After the slices were slightly dried, add freshly prepared DAB chromogenic solution (diluent and concentrated solution 1000:50) dropwise in the circle, and control the color development time under the microscope. For α-SMA (Abcam, ab5694, 1: 1000 dilution), FAP (Abcam, ab314456, 1: 1000 dilution), P-gp (Abcam, ab261736, 1: 1000 dilution) staining, the process is also similar as GLUT-1 and RAD51. The positive color was brownish yellow, and the sections were rinsed with pure water to stop the color development.

## Statistics and reproducibility

Statistical analyses were conducted using GraphPad Prism 9.0, and the results are presented as mean values ± standard deviation. The normality of the data was first confirmed by Shapiro-Wilk test through using GraphPad Prism 9.0, when the sample size is less than 5000. Student's unpaired $t$ test (two-tailed) was employed for comparison of two groups, while one-way ANOVA using a Tukey post hoc test was employed for comparison of ≥ three groups. A significance level of $p < 0.05$ was considered statistically significant. Measurements were obtained from distinct samples, and no statistical method was used to predetermine the sample size. Data collection and analysis were not performed blind to the experimental conditions, and no data were excluded from the analyses. It is noteworthy that all results can be replicated from the available source data files.

## Reporting summary

Further information on research design is available in the Nature Portfolio Reporting Summary linked to this article.

## Data availability

All data supporting the findings of this study are available within the Article, Supplementary Information, or the Source Data file. The source data generated during this study have been deposited in the Figshare database, https://doi.org/10.6084/m9.figshare.25041767. The source data are provided along with this paper. Source data are provided with this paper.

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

## Acknowledgements

This work was supported by the Ruijin Hospital Guangci Introducing Talent Project (H.Z.); Research Fellow (Grant No. 353146 (H.Z.)), Project (Grant No. 347897 (H.Z.)), Solution for Health Profile (Grant No. 336355 (H.Z.)) and InFLAMES Flagship (Grant No. 337531 (H.Z.)) grants from the Academy of Finland; Finland China Food and Health International Pilot Project (H.Z.) funded by the Finnish Ministry of Education and Culture; the Tor, Joe, and Pentti Borg Memorial Fund (O.S.). Parts of the research used Research Council of Finland Research Infrastructure "Printed Intelligence Infrastructure" (PII-FIRI) (H.Z.). It was also supported by the Shanghai Anticancer Association EYAS PROJECT (Grant No. SACA-CY22C07 (D.Z.)). We thank the Electron Microscopy Laboratory, Institute of Biomedicine, University of Turku, and Biocenter Finland. Also, the Biocenter Finland Bioinformatics, CSC IT Center for Science, and Prof. Mark Johnson and Dr. Jukka Lehtonen are gratefully acknowledged for the excellent computational infrastructure at the Åbo Akademi University. We are grateful to Gösta Branders research fund, Åbo Akademi Research Foundation (Gösta Branders forskningsfond, Stiftelsen för Åbo Akademi).

## Author contributions

J.Y. and H.Z. conceived the idea and wrote the manuscript. J.Y., R.B., M.R. and X.M. were responsible for carried out the experiments, material synthesis and animal experiments. Y.L. and D.Z. assisted in manuscript revision. O.S. and H.Z. supervised the project, provided funding, and helped with language editing. All authors discussed the results and have given approval to the final version of the paper.

## Competing interests

The authors declare no competing interests.
