## [Peer Review File · Nature Communications]

Development of Aptamer-DNAzyme Based Metal-Nucleic Acid Frameworks for Gastric Cancer therapyEditorial Note: Parts of this Peer Review File have been redacted as indicated to remove third-party material where no permission to publish could be obtained.

REVIEWER COMMENTS

Reviewer #1 - metal-nucleic acid frameworks, chemistry (Remarks to the Author):

This work has significant implications on the use of MNFs in cancer therapy. The authors assert that they have achieved a route to MNFs via milder (compared to other approaches for the preparation of MNFs or MOFs) synthesis conditions by incorporating multi-fragmented long DNA, calcium ions, and the therapeutic protein IRF-1. Due to the mild conditions used, the MNFs can effectively encapsulate therapeutic proteins that would otherwise be susceptible to heat-induced degradation.

Moreover, it is worth noting that previous literature primarily relied on single-functional DNA for MNF or MOF preparation, necessitating additional chemical steps to attach targeting moieties to their surfaces. The MNFs discussed in this manuscript feature long DNA strands with a dual function: 1) induce ROS-mediated DNA damage; 2) bind to the HER-2 receptor on gastric cancer cells. As a result, there is no need for an additional step where the surface is modified for cancer cell targeting.

In the manuscript, the authors have developed MNFs comprised of calcium ions, fragmented DNA (including HER-2 aptamer and GLUT-1 DNAzyme), and IRF-1. These MNFs were designed to induce the production of reactive oxygen species (ROS) in gastric cancer cells that overexpress HER-2, leading to an anti-cancer effect. The authors have confirmed that HER-2 promotes MNF internalization in gastric cancer cells and demonstrated that GLUT-1 DNAzyme and IRF-1 trigger ROS production at the cellular level. In addition, the authors have presented evidence of reduced tumor growth in mice following MNF treatment.

Overall, the study presents a multifunctional metal-nucleic acid nanosystem. However, more information about the experimental conditions/methodologies and data analysis are needed to facilitate assessment and interpretation. Furthermore, it is difficult to draw definitive conclusions from the results due to the absence of interpretations grounded in established literature within the Results and Discussion sections. It is worth noting that the manuscript references a total of 38 sources, a number that is significantly lower than what is typically found in articles published in Nature Communications, which often include more than 50 references. It would be beneficial to expand the analysis to include mechanistic insights into the intracellular uptake pathways of MNFs, elucidating how individual components influence the endocytic process. Also, it would be beneficial to add tissue/cellular-level analysis to substantiate the observed tumor reduction in the murine model.

Therefore, this manuscript does not meet the standard for publication in Nature Communications at this time.

Introduction

The authors should clarify the advantages of MNFs over MOFs and explain the rationale behind selecting calcium ions as a component of the MNFs.

Please explain how the HER-2 aptamers were selected to target HER-2 overexpressing gastric cancer cells.

Ensure consistent use of "HER-2" instead of "Her-2."

Please include literature to support the use of IRF-1 and GLUT-1 DNAzyme in cancer therapy.

Figure 1

A(b) – Correct the typo "Low yeild" to "Low yield".

Clarify which figures are referenced in the paragraph on Pg. 6, Line 116 - Pg.7, Line 133.

Pg. 7, Lines 134-142 – Please provide a citation to support this claim.

Figure 2

Pg.11 – Please clarify the meaning of GLUT-1imaging performance.

In Figure 2A, the cleaved-GLUT1 mRNA band is difficult to visualize at 2 mM Ca²⁺. Please provide a densitometric analysis of the bands and clarify the identity of the band between H-GDz-GLUT1 and H-GDz. Please also cite references for the detectable range of GLUT-1 mRNA and its physiological relevance.

The authors should consider switching the order of Figs. 2A and 2B. In Figure 2E, please indicate what the red color represents in the figure legend.

The data regarding the lysosomal escape of the MNFs (Figure S4) is difficult to interpret. Please include data to show the intracellular trafficking of the MNFs to confirm whether they will be localized in the lysosomes.

Please clarify why the F-intensity of H-GDz is increasing at the 8-h time point and provide details on how this assay was performed. Were the MNFs removed from the cells at a fixed time before analysis at the 2-h and 8-h time points? Since the F-intensity highly influences the Pearson's correlation for colocalization, it is unclear whether the colocalization analysis is valid and that the lysosomal escape of the MNFs can be concluded. Please provide the sample size used to draw this claim and add scale bars to the images.

Please provide higher magnification images to support the claims associated with Figure 2F;

specify the sample size and the number of biological replicates in the data associated with Figure 3.

Please include particle characterization after IRF-1 loading (e.g., size and morphology).

Pertaining to Fig. 3D, explain why the Ca²⁺ levels are similar between MNF with and without the HER-2 aptamer.

Please clarify how the mean gray values of the images were determined. Some of the images (IRF/H-GDz/Ca (Fig. 3E) and controls (Fig. 3D, 3I and 3K) appear to have different cellular densities. Please clarify what threshold and sample size were used to collect the images.

Please consider investigating the uptake of the MNFs in non-HER-2 expressing cell lines in order to make the claim that the specific targeting of gastric cancer cells and elucidate the cellular uptake pathway analysis to increase the quality of the manuscript.

Please specify the sample size and number of biological replicates to understand the reproducibility of the results.

Figure 4

There is a lack of information in the Materials and Methods for the in vivo studies, which makes it difficult to evaluate the results. What is the strain/age/sex? Inoculation of human cells in naïve mice will cause host responses and rejection thus the results may not be valid. Is a mice xenograft model used? How long was the tumor incubated before treatment?

Fig.4 D, E, F and G represent the same results (Tumor reduction). The authors should include data on the effect of this treatment on the tumor microenvironment.

The data associated with Figure S7 is key to assessing the functionality of MNFs (GLUT-1 and RAD51 inhibition) in vivo. Figure S7 should be placed in the main manuscript. The image of IRF/H-GDz/CA should also represent the area of tissue as other groups. Please indicate what

the dotted circle represents.

Including ROS measurement/detection in vivo would increase the quality of the manuscript.

This could help with interpretation of the in vivo results.

Please specify the sample size and number of biological replicates for reproducibility of results.

Conclusion

Since many effective methods have been reported to stimulate ROS production and an anti-cancer effect, additional discussion is required to clarify the advantages of the MNFs as compared to other nanosystems.

Reviewer #2 - DNAzymes, cancer therapy (Remarks to the Author):

This is an interesting manuscript that describes the potential of metal-nucleic acid frameworks (MNFs) to serve as functional nanomaterials. However, MNF production have in the past generally needed harsh generative conditions (heat, time, small oligos). DNAzymes are DNA-based cleaving agents that degrade mRNA. Here, the authors use a multi-segmented oligonucleotide (MSO) approach to provide proof-of-principle evidence for the efficacy of an interferon regulatory factor-1 (IRF-1) loaded Ca²⁺/aptamer-DNAzyme MNF to target regulate GLUT-1 mRNA expression in human EGFR-2 (Her-2)-positive gastric cancer cells. The authors provide evidence of GSH depletion and ROS homeostasis, among other measures of efficacy. Tumor growth was inhibited by 90% in the RF/H-GDz/Ca group.

The work may of significance to the field and related fields particularly if there is added evidence for the generality of the observations, but currently this is just focussed on a gastric cancer model. The work generally supports the conclusions and claims, but would be strengthened by incorporation of the following additional items:

The Abstract should mention efficacy data in tumor bearing mice.

The legends in the Suppl Materials should indicate how many 'n' (biologically independent determinations) the plots and images are meant to represent.

What was the scientific rationale for the choice of mutant bases in MH-GDz sequences

containing Mutated-Her-2 aptamer?

While the targeting effect of H-GDz was tested in Her-2 positive NIC-N87 gastric cancer cells, similar experiments should be performed in non-gastric cancer cells or Her-2 negative/low gastric cancer cells to demonstrate specificity beyond mutant MNF.

The MNFs accumulated in the kidney and liver. The mice did not demonstrate overt adverse effects. Did the authors investigate whether there were changes in liver and kidney biomarkers?

Tumor growth was inhibited by 90% in the IRF/H-GDz/Ca group. Were there statistically significant effects observed in the H-GDz/Ca and IRF/H-MGDz/Ca groups?

Student's t test was used for comparisons. The authors should indicate if the data was normal. Why was ANOVA (if data was normal) not used for the multi-group studies?

There is no Discussion, only a Conclusions section. The Discussion also should refer to limitations using DNAzymes and limitations in this study.

Reviewer #3 DNAzymes, orthotopic models (Remarks to the Author):

I find this paper well-researched, simple to follow, and to the point.

There are a few issues to ponder, which I trust will enhance the quality of the paper.

Specifics:

Line 45: correct spelling is 'guarantee'

Line 89: define what 'IRF' is at first mention.

Scheme 1, part D: correct spelling is 'Mitochondria'. Also, Title should have 'Inhibition' with lower case 'i' ... ie. 'inhibition'.

Line 112: change 'to our astonishment' to 'unexpectedly' as the former is journalistic.

Line 185: correct spelling is 'synthesis'.

Line 228: correct spelling is 'comparatively'.

Line 236: introduce what type of cell line NIC-N87 is, and why it was chosen for this study.

Line 268: correct word is 'validate'.

Fig 3A: Where are the MFI error bars?

Fig 4b: remove dotted/dashed line around figure 4B. It is distracting, and does not really

help with segregating B from A, C or D.

Line 665: correct term is 'TRITC'.

Line 679: explain why samples were not denatured when doing the GLUT-1 protein electrophoresis.

Line 690: mention whether in the release study, the sample tubes were mixed constantly or stagnant.

Line 787: how was treatment administered in vivo, route?

Reviewer #4 - Gastric cancer, targeting GLUT (Remarks to the Author):

In this manuscript, the authors reported an interferon regulatory factor-1 (IRF-1) loaded Ca^{2+} /(aptamer-deoxyribozyme) MNF to target regulate glucose transporter (GLUT-1) expression in human epidermal growth factor receptor-2 (Her-2) positive gastric cancer cells, which could disrupt GSH/ROS homeostasis, suppress DNA repair, and augment ROS-mediated DNA damage therapy. Although there are interesting parts in the contents of the study, in its current form the data are preliminary, and the conclusions require further validations. The direction of the study has a certain suspicion of artificial selection and lacks objective logic, some details and relevant discussions are missed. Specifically, the authors should address the following points to improve the quality of the paper

1. The author verified the binding capacity of DNA sequences of different lengths with calcium ions, since the synthesis conditions of metal-DNA nanocomposites are complex and easily degradable, how to ensure the synthesis efficiency and stability in vitro and in vivo experiments; how to determine the concentration of the transferred oligonucleotide sequence to be 83 mM when constructing? During the synthesis process, does the folding of longer oligonucleotide chains and spatial conformation changes affect the affinity between the DNA sequence and calcium ions, Will MNF production also be affected as a result? The authors need to provide additional explanations on these aspects.

2. When constructing MOFs, why did the authors choose calcium ions, which do not easily form MNF structures with nuclear acids, instead of other metal materials with milder synthesis conditions? The authors need to supplement the binding affinity of other metal

ions with longer DNA sequences.

3.The toxicity of nanocarriers is essential for delivery. The cell viability of IRF/H-GDz/Ca MNF with different dosages should be measured.

4.GLUT-1, disrupt pentose phosphate pathway (PPP) and the synthesis of NADPH, leading to a decrease in cysteine synthesis and disruption of GSH/ROS homeostasis. The authors need to supplement the changes in metabolic key products and enzymes in five groups including Control, H-MGDz/Ca, H-GDz/Ca, IRF/H-MGDz/Ca, and IRF/H-GDz/Ca in Figure 3.

5.Multiple clinical drug studies targeting HER2 positive advanced gastric cancer patients have shown certain therapeutic efficacy, while the drug resistance could not be ignored. Whether IRF/H-GDz/Ca MNF targeting HER2 positive gastric cancer cells described in the article can effectively improve drug resistance can be explained by constructing preclinical models such as PDX and organoid models.

6.Since nanoparticle behavior in mice bloodstream is more complex than in vitro, this study raises serious concerns: What is the blood circulation time of MNF in vivo? The authors need to clarify how to determine the observation time point of 48 hours in vivo experiments. Is there clear comparison of the theoretical and experimental data (e.g, curves)?

7.There is no discussion of different cancer lines selection for in vitro and in vivo experiments. The authors need to supplement the relevant experiments of another cell line, such as SNU216, to increase the reliability of the experimental results.

8.In Figure 4C, the authors detected the fluorescence intensity of nanomaterials in tumor tissue, and the authors need to provide continuous slices that reflect DNA damage indicators; Similarly, in the IRF/H-GDz/Ca group, the tumor was almost completely eradicated at day 21, exhibiting over 90% significant tumor reduction (Figure 4D). This significant treatment outcome can be attributed to the synergistic effect of GLUT-1 inhibition and IRF-1 increase (S7). The authors also need to provide the fluorescence intensity values of nanomaterials in these five groups.

9.In the section of in vivo biosafety study, although the authors investigated the changes of

mouse body weight, more biosafety experiments need to be provided, for example, major organ pathological analysis, liver function evaluations, the biosafety of kidney, blood compatibility, etc.

Point-by-point responses to reviewers

Dear reviewers, we sincerely appreciate the opportunity to revise our manuscript in light of your insightful comments. Your feedback is invaluable, and we are fully committed to addressing each of the concerns in a direct and concise manner. We are appreciating to the reviewers for their valuable feedback and constructive criticism. We are confident that your insights will significantly improve the quality and impact of our work. In the attached point-by-point responses file, you will find **your comments are highlighted in bold black text**, and **our responses are provided in blue**. **We have made revisions to the main text and Supporting Information, highlighted in yellow**, and modifications to the methods section have been incorporated to align with the updated figures.

Thank you again for your dedication to improving the scientific rigor of our manuscript.

Reviewer #1 - metal-nucleic acid frameworks, chemistry (Remarks to the Author):

This work has significant implications on the use of MNFs in cancer therapy. The authors assert that they have achieved a route to MNFs via milder (compared to other approaches for the preparation of MNFs or MOFs) synthesis conditions by incorporating multi-fragmented long DNA, calcium ions, and the therapeutic protein IRF-1. Due to the mild conditions used, the MNFs can effectively encapsulate therapeutic proteins that would otherwise be susceptible to heat-induced degradation.

Moreover, it is worth noting that previous literature primarily relied on single-functional DNA for MNF or MOF preparation, necessitating additional chemical steps to attach targeting moieties to their surfaces. The MNFs discussed in this manuscript feature long DNA strands with a dual function: 1) induce ROS-mediated DNA damage; 2) bind to the HER-2 receptor on gastric cancer cells. As a result, there is no need for an additional step where the surface is modified for cancer cell targeting.

In the manuscript, the authors have developed MNFs comprised of calcium ions, fragmented DNA (including HER-2 aptamer and GLUT-1 DNAzyme), and IRF-1. These MNFs were designed to induce the production of reactive oxygen species (ROS) in gastric cancer cells that overexpress HER-2, leading to an anti-cancer effect. The authors have confirmed that HER-2 promotes MNF internalization in gastric cancer cells and demonstrated that GLUT-1 DNAzyme and IRF-1 trigger ROS production at the cellular level. In addition, the authors have presented evidence of reduced tumor growth in mice following MNF treatment.

Overall, the study presents a multifunctional metal-nucleic acid nanosystem.

However, more information about the experimental conditions/methodologies and data analysis are needed to facilitate assessment and interpretation. Furthermore, it is difficult to draw definitive conclusions from the results due to the absence of interpretations grounded in established literature within the Results and Discussion sections. It is worth noting that the manuscript references a total of 38 sources, a number that is significantly lower than what is typically found in articles published in Nature Communications, which often include more than 50 references. It would be beneficial

to expand the analysis to include mechanistic insights into the intracellular uptake pathways of MNFs, elucidating how individual components influence the endocytic process. Also, it would be beneficial to add tissue/cellular-level analysis to substantiate the observed tumor reduction in the murine model.

Therefore, this manuscript does not meet the standard for publication in Nature Communications at this time.

Dear reviewer, thank you for your insightful comments on our manuscript. We appreciate your acknowledgment of the implications of our work on the use of MNFs in cancer therapy. We also extend our gratitude for your insightful questions, as these inquiries have significantly contributed to enhancing the quality of our manuscript.

In light of your comments, we will address more comprehensive experimental details and a more extensive literature discussion in the revised version. Specifically, we will provide additional experimental information, incorporate more in-depth data analysis grounded in established literature, increase the number of references to include a more comprehensive literature review, elucidate the intracellular uptake pathways of MNFs, and provide additional tissue/cellular-level analysis to support the observed tumor reduction in the further Patient-Derived Xenograft Mice model.

We hope the responses detailed below will adequately address the reviewer's points and will satisfy the concerns that were raised.

Introduction

1. The authors should clarify the advantages of MNFs over MOFs and explain the rationale behind selecting calcium ions as a component of the MNFs.

We thank you for your valuable suggestions, and we understand that further clarification is needed for these points.

First, the Metal-Nucleic Acid Frameworks (MNFs) are self-assembled materials composed of DNA sequences and metal ions, while the Metal-Organic Frameworks (MOFs) are composed of metal ions and organic ligands. The greatest advantage of MNFs over MOFs lies in the multifunctionality of the DNA sequences contained within MNFs compared to the organic ligands. Specifically, (1) DNA sequences can serve as nucleic acid drugs with therapeutic functions; (2) DNA can construct adapter sequences for specific cell targeting; (3) DNA can be used to construct molecular fluorescence beacons for disease gene diagnosis. Therefore, utilizing MNF materials as carriers for drug delivery not only harnesses the functionality of the drug itself but also leverages the functions of DNA sequences simultaneously.

However, MNFs suffer from the disadvantages of insufficient stability compared to MOFs and the inability to achieve rapid synthesis at room temperature. **Therefore, our main highlight in this work was to discover that multi-fragmented long-chain DNA can readily bind with metal ions and facilitate the rapid stable synthesis of MNFs at room temperature.** By regulating the

functionality of the DNA sequences, we were able to construct DNAzyme MNF materials with targeting capabilities that can identify and regulate tumor genes, as well as loading protein drugs.

Regarding the selection of calcium ions during the preparation of MNF materials, we had several reasons. (1) The DNAzyme we utilized was responding to calcium ions as co-factor; thus, calcium ions can provide the required metal cofactor for the DNAzyme after the MNF is dissociated. (2) Calcium ions possess various anti-cancer-related functions, and in this study, we primarily focused on exploring their role in mitochondrial calcification. (3) From a material construction perspective, based on both previous research and our own explorations, it has been observed that calcium ions do not readily facilitate robust self-assembly with DNA to form MNF materials. Thus, the utilization of calcium ions in the preparation of MNF materials serves to highlight the significant role played by elongated DNA sequences in improving the binding affinity between DNA sequences and metal ions.

We hope these explanations satisfy your requirements.

Revision made:

Introduction

The integration of metals with DNA structures can lead to the creation of exceedingly powerful materials^{1, 2,.....}

In recent years, MNFs have gradually gained momentum in the field of cancer treatment by combining the catalytic functions of metal ions with the therapeutic functions of nucleic acid drugs⁷⁻¹⁰. MNFs are composed of metal ions and DNA sequences, which similar to another famous material “Metal-Organic Frameworks (MOFs)” that consist of metal ions and organic ligands¹¹. One of the primary advantages of MNFs over MOFs is the multifunctionality of DNA sequences¹² compared to organic ligands. Specifically, DNA sequences in MNFs can serve as therapeutic nucleic acid drugs¹³, facilitate specific cell targeting through adapter sequences¹⁴, and enable disease gene diagnosis via the construction of molecular fluorescence beacons^{15, 16}. Consequently, the utilization of MNF materials for drug delivery not only exploits the inherent properties of the drug but also harnesses the diverse functionalities of the DNA sequences.

..... In this study, we incorporated a HER-2 targeted aptamer into the Ca²⁺-dependent GLUT-1 DNAzyme^{4, 23}, thereby elongating the oligonucleotide length to achieve Ca²⁺-assisted self-mineralization under room temperature, rendering visualized and silenced treatment of GLUT-1 mRNA (**Scheme 1A**). The HER-2 targeting aptamers were selected through SELEX (Systematic Evolution of Ligands by Exponential enrichment) studies by previous researchers.²⁴⁻²⁶ This approach not only increases the MNFs synthetic capacity but achieves targeted delivery of MNFs to HER-2 overexpressing cells. The choice of calcium ions during MNF material preparation was based on several factors. Firstly, the responsiveness of the DNAzyme to calcium ions enabled the provision of essential metal cofactors for the DNAzyme following material release. Secondly, the diverse anti-cancer functions of calcium ions were a focal point of our study, particularly their role in mitochondrial calcification. Thirdly, considering material construction, previous research and our own investigations have demonstrated the challenging nature of achieving robust self-assembly between calcium ions and DNA to form MNF materials. Therefore, our use of calcium ions highlights how elongated DNA sequences can enhance the binding capacity of DNA sequences with metal ions.

2. Please explain how the HER-2 aptamers were selected to target HER-2 overexpressing gastric cancer cells.

We thank you for raising this crucial point regarding the description of the method used for selecting HER-2 targeting aptamers. In this study, it is important to clarify that we referenced and utilized sequences from reported research, and therefore, we did not conduct any screening process ourselves.

Based on the literature we found (<https://doi.org/10.1073/pnas.1302594110>), for the process to select the HER-2 aptamers targeting HER-2 overexpressing gastric cancer cells, a rigorous SELEX (Systematic Evolution of Ligands by EXponential enrichment) process was conducted. The SELEX process emphasized the identification of specific binders, which were subjected to random mutations using Polymerase Chain Reaction (PCR). This rigorous selection process was employed to ensure the specificity and affinity of the HER-2 aptamers for targeting HER-2 overexpressing gastric cancer cells.

Furthermore, the targeting specificity of this screened aptamer toward HER-2 from various cell lines has been widely validated, including the N-87 cell line used in this manuscript, as well as several breast cancer cell lines (Mahlknecht, Maron et al. 2013, Lee, Dam et al. 2015, Ma, Zhan et al. 2019).

Therefore, in accordance with the reviewer's suggestion, we have added a description of how the aptamers were screened in the "introduction" part. We hope that the revisions will meet the reviewer's requirements.

Revision made:

..... In this study, we incorporated a HER-2 targeted aptamer into the Ca²⁺-dependent GLUT-1 DNzyme^{4, 23}, thereby elongating the oligonucleotide length to achieve Ca²⁺-assisted self-mineralization under room temperature, rendering visualized and silenced treatment of GLUT-1 mRNA (Scheme 1A). The HER-2 targeting aptamers were selected through SELEX (Systematic Evolution of Ligands by Exponential enrichment) studies by previous researchers²⁴⁻²⁶. This approach not only increases the MNFs synthetic capacity but achieves targeted delivery of MNFs to HER-2 overexpressing cells.....

3. Ensure consistent use of "HER-2" instead of "Her-2."

We appreciate your careful attention to detail, and we apologize for the inconsistency in the use of the term "HER-2" and "Her-2" in our manuscript. We ensured the consistent use of "HER-2" throughout the entire manuscript in the revised version.

4. Please include literature to support the use of IRF-1 and GLUT-1 DNzyme in cancer therapy.

Dear Reviewer, we appreciate your careful evaluation of our manuscript and the insightful comments provided. We have taken into account your suggestion regarding the need for a more comprehensive analysis of references. In response, we have significantly expanded the citation of relevant literature in the "introduction" part and also throughout the entire manuscript.

In introduction

1) For HER-2 aptamer selection, we cite literature including ref(Mahlknecht, Maron et al. 2013, Lee, Dam et al. 2015, Ma, Zhan et al. 2019).

1. Lee, H., Dam, D.H., Ha, J.W., Yue, J. & Odom, T.W. Enhanced Human Epidermal Growth Factor Receptor 2 Degradation in Breast Cancer Cells by Lysosome-Targeting Gold Nanoconstructs. *ACS Nano* 9, 9859-9867 (2015).
2. Ma, W. et al. An Intelligent DNA Nanorobot with in Vitro Enhanced Protein Lysosomal Degradation of HER2. *Nano Lett* 19, 4505-4517 (2019).
3. Mahlkecht, G. et al. Aptamer to ErbB-2/HER2 enhances degradation of the target and inhibits tumorigenic growth. *Proc Natl Acad Sci U S A* 110, 8170-8175 (2013).

2) For support the use of GLUT-1 DNzyme in cancer therapy, we cite new literature including ref(Alakus, Berlth et al. 2010, Wu, Zhang et al. 2022, Wang, He et al. 2023, Yan, Ma et al. 2023)

4. Wu, S. et al. Nano-enabled Tumor Systematic Energy Exhaustion via Zinc (II) Interference Mediated Glycolysis Inhibition and Specific GLUT1 Depletion. *Advanced Science* 9, 2103534 (2022).
5. Wang, R. et al. Site-Specific Bioorthogonal Activation of DNzymes for On-Demand Gene Therapy. *Journal of the American Chemical Society* 145, 17926-17935 (2023).
6. Yan, J. et al. An autocatalytic multicomponent DNzyme nanomachine for tumor-specific photothermal therapy sensitization in pancreatic cancer. *Nature Communications* 14, 6905 (2023).
7. Alakus, H. et al. S1941 GLUT-1 Expression in Gastric Cancer: A New Independent Prognostic Marker. *Gastroenterology* 138, S-285 (2010).

3) For support the use of IRF-1 and in cancer therapy, we cite literature including ref(Tanaka, Ishihara et al. 1996, Tan, Yuan et al. 2020)

8. Tan, L. et al. Interferon regulatory factor-1 suppresses DNA damage response and reverses chemotherapy resistance by downregulating the expression of RAD51 in gastric cancer. *Am J Cancer Res* 10, 1255-1270 (2020).
9. Tanaka, N. et al. Cooperation of the tumour suppressors IRF-1 and p53 in response to DNA damage. *Nature* 382, 816-818 (1996).

4) For cancer synergistic therapy by using DNzyme and metal co-factor, we cite(Zhao, Li et al. 2022)

10. Zhao, Y. et al. Multifunctional DNzyme-Anchored Metal-Organic Framework for Efficient Suppression of Tumor Metastasis. *ACS Nano* 16, 5404-5417 (2022).

5) For DNAzyme and Aptamer mediated ROS busting in cancer cells, we cite ref(Zhi, Zhang et al. 2023)

11. Zhi, S., Zhang, X., Zhang, J., Wang, X.-y. & Bi, S. Functional Nucleic Acids-Engineered Bio-Barcode Nanoplatforams for Targeted Synergistic Therapy of Multidrug-Resistant Cancer. *ACS Nano* 17, 13533-13544 (2023).

6) For DNAzyme related mitochondria therapy, we cite(Wang, Yi et al. 2023)

12. Wang, D. et al. Photoactivated DNA Nanodrugs Damage Mitochondria to Improve Gene Therapy for Reversing Chemoresistance. *ACS Nano* 17, 16923-16934 (2023).

7) For DNAzyme based combination therapy, we cite(Qian, Zhou et al. 2022)

13. Qian, R.-C. et al. Combination Cancer Treatment: Using Engineered DNAzyme Molecular Machines for Dynamic Inter- and Intracellular Regulation. *Angewandte Chemie International Edition* 61, e202210935 (2022).

8) For MOF introduction we cite(Tyagi, Wijesundara et al. 2023)

14. Tyagi, N., Wijesundara, Y.H., Gassensmith, J.J. & Papat, A. Clinical translation of metal-organic frameworks. *Nat Rev Mater* (2023).

9) For comparing the advantage for MNF over MOF materials, we cite(Yang, Zhang et al. 2020) to emphasize the advantage of DNA sequence over organic ligands.

15. Yang, Y., Zhang, R. & Fan, C.H. Shaping Functional Materials with DNA Frameworks. *Trends Chem* 2, 137-147 (2020).

10) For aptamer introduction, we cite(DeRosa, Lin et al. 2023)

16. DeRosa, M.C. et al. In vitro selection of aptamers and their applications. *Nat Rev Method Prime* 3 (2023).

11) We also cite(Liu and Lu 2007, Selnihhin, Sparvath et al. 2018) to describe how to utilize DNA and DNAzyme material to build molecular beacon.

17. Selnihhin, D., Sparvath, S.M., Preus, S., Birkedal, V. & Andersen, E.S. Multifluorophore DNA Origami Beacon as a Biosensing Platform. *ACS Nano* 12, 5699-5708 (2018).

18. Liu, J. & Lu, Y. A DNAzyme Catalytic Beacon Sensor for Paramagnetic Cu²⁺ Ions in Aqueous Solution with High Sensitivity and Selectivity. *Journal of the American Chemical Society* 129, 9838-9839 (2007).

In results

12) We cite more references for discussing the persistence length of longer strand DNA during the Molecular Dynamic analyzing.

19. Maassen, S.J., de Ruiter, M.V., Lindhoud, S. & Cornelissen, J. Oligonucleotide Length-Dependent Formation of Virus-Like Particles. *Chemistry* 24, 7456-7463 (2018).

20. Roth, E., Glick Azaria, A., Girshevitz, O., Bitler, A. & Garini, Y. Measuring the Conformation and Persistence Length of Single-Stranded DNA Using a DNA Origami Structure. *Nano Letters* 18, 6703-6709 (2018).

21. Chi, Q., Wang, G. & Jiang, J. The persistence length and length per base of single-stranded DNA obtained from fluorescence correlation spectroscopy measurements using mean field theory. *Physica A: Statistical Mechanics and its Applications* 392, 1072-1079 (2013).

13) When discussing the detecting limitation of the gene of interest, we cite these citations.

24. Femino, A.M., Fay, F.S., Fogarty, K. & Singer, R.H. Visualization of Single RNA Transcripts in Situ. *Science* 280, 585-590 (1998).

25. Treck, T., Lionnet, T., Shroff, H. & Lehmann, R. mRNA quantification using single-molecule FISH in *Drosophila* embryos. *Nature Protocols* 12, 1326-1348 (2017).

26. Pan, J. et al. Visible/near-infrared subdiffraction imaging reveals the stochastic nature of DNA walkers. *Science Advances* 3, e1601600 (2017).

27. Sheng, C. et al. Spatially resolved in vivo imaging of inflammation-associated mRNA via enzymatic fluorescence amplification in a molecular beacon. *Nature Biomedical Engineering* 6, 1074-1084 (2022).

14) We refer to these citations when referring to the distribution of GLUT-1 in the body and the circulation of MNF in the body.

28. Lu, Y.X. et al. Pharmacological Ascorbate Suppresses Growth of Gastric Cancer Cells with GLUT1 Overexpression and Enhances the Efficacy of Oxaliplatin Through Redox Modulation. *Theranostics* 8, 1312-1326 (2018).

29. Joost, H.G. & Thorens, B. The extended GLUT-family of sugar/polyol transport facilitators: nomenclature, sequence characteristics, and potential function of its novel members. *Mol Membr Biol* 18, 247-256 (2001).

30. Airley, R., Evans, A., Mobasher, A. & Hewitt, S.M. Glucose transporter Glut-1 is detectable in peri-necrotic regions in many human tumor types but not normal tissues: Study using tissue microarrays. *Annals of Anatomy - Anatomischer Anzeiger* 192, 133-138 (2010).

31. Qiu, C. et al. Regulating intracellular fate of siRNA by endoplasmic reticulum membrane-decorated hybrid nanoplexes. *Nature Communications* 10, 2702 (2019).

15) When discussing lysosomal escape experimental methods, we cite

32. Xu, J. et al. Precise targeting of POLR2A as a therapeutic strategy for human triple negative breast cancer. *Nature Nanotechnology* 14, 388-397 (2019).

16) When discussing the impact of our IRF/H-GDz/Ca material on the drug resistance of the newly constructed PDX model, we cited the following literature

37. Deeley, R.G. & Cole, S.P. Function, evolution and structure of multidrug resistance protein (MRP). *Semin Cancer Biol* 8, 193-204 (1997).

38. Liu, J. et al. Nanoenabled Intracellular Calcium Bursting for Safe and Efficient Reversal of Drug Resistance in Tumor Cells. *Nano Letters* 20, 8102-8111 (2020).

39. Yuan, J.S. et al. Interferon regulatory factor-1 reverses chemoresistance by downregulating the expression of P-glycoprotein in gastric cancer. *Cancer Lett* 457, 28-39 (2019).

16) When clarifying how the observation time point for the 48-hour in vivo experiments was determined and when comparing findings with other reported studies on oligonucleotide drugs, we cited

40. Hallare, J. & Gerriets, V. in *StatPearls* (StatPearls Publishing Copyright © 2023, StatPearls Publishing LLC., Treasure Island (FL); 2023).

41. Nnane, I.P. in *Encyclopedia of Analytical Science (Second Edition)*. (eds. P. Worsfold, A. Townshend & C. Poole) 126-133 (Elsevier, Oxford; 2005).

42. Jiang, T. et al. Cation-Free siRNA Micelles as Effective Drug Delivery Platform and Potent RNAi Nanomedicines for Glioblastoma Therapy. *Advanced Materials* 33, 2104779 (2021).

43. Xu, X. et al. Enhancing tumor cell response to chemotherapy through nanoparticle-mediated codelivery of siRNA and cisplatin prodrug. *Proc Natl Acad Sci U S A* 110, 18638-18643 (2013).

17) When studying the metabolites in tumor cells after GLUT-1 is silenced, we cited the following literature

44. Du, B., Yu, M. & Zheng, J. Transport and interactions of nanoparticles in the kidneys. *Nature Reviews Materials* 3, 358-374 (2018).

45. Xu, X. et al. Multifunctional Envelope-Type siRNA Delivery Nanoparticle Platform for Prostate Cancer Therapy. *ACS Nano* 11, 2618-2627 (2017).

18) When studying renal metabolism

46. Choi, H.S. et al. Design considerations for tumour-targeted nanoparticles. *Nature Nanotechnology* 5, 42-47 (2010).

We hope that these revisions will enhance the depth, accuracy, and credibility of the manuscript and meet the reviewer's expectations.

5. A(b) – Correct the typo "Low yeild" to "Low yield."

Thank you for bringing the typographical error to our attention. We apologize for the oversight and have corrected the error from "Low yeild" to "Low yield" in the revised version of the manuscript.

6. Clarify which figures are referenced in the paragraph on Pg. 6, Line 116 - Pg.7, Line 133.

We thank you for your query regarding the figures referenced in the specified paragraph (Pg. 6, Line 116 - Pg.7, Line 133). We apologize for any confusion, as this particular section of our work does not have accompanying figures.

We highly appreciate and support your suggestion for visual representation of this data through the use of tables. Accordingly, we have included a corresponding table in the supplementary material (Supplementary Table S2), providing a detailed description of the results for better visualization.

Revision made:

In main text:

..... (Fig. 1a2, Supplementary Fig.1). These observations provided macroscopic evidence supporting the superior MNF synthesis capability of longer DNA sequences compared to shorter ones.

Subsequently, to further investigate the structural composition of the MNF, we analyzed the DNA and calcium elements content in the generated nanomaterials (Table S2). We observed that for the H-GDz sequence, when the input amount of DNA was approximately 52.39 μg, the actual amount of DNA in the obtained product was around 50.08 μg, indicating a high utilization efficiency of the DNA sequence (95.6%).....

In supporting information:

Supplementary Tables S2. Structural composition of the MNF.

	Input DNA weight (μg)	Incorporated DNA (μg)			DNA utilization efficiency			Incorporated Calcium (μg)			Calcium mass ratio			yield μg
H-GDz (83 bases)	52.39	50.08	51.09	50.81	95.6%	97.5%	96.9%	10.3	10.4	11.1	17.0%	16.9%	17.9%	61.26 \pm 0.79
GDz (41 bases)	51.5	48.4	47.9	46.2	94.0%	93.0%	89.7%	6	5.5	5.8	11.0%	10.3%	11.1%	53.26 \pm 1.20
Substrate (20 bases)	50.8	1.8	1.1	1.2	3.5%	2.1%	2.3%	0.1	0.05	0.1	5.3%	4.3%	7.6%	1.44 \pm 0.39

Supplementary Fig. Calculated results of calcium and DNA content in Ca-based MNF materials.

7. Pg. 7, Lines 134-142 – Please provide a citation to support this claim.

We appreciate your suggestion to include citations in the paragraph (Pg. 7, Lines 134-142). Indeed, the foundation laid by previous research is instrumental in helping readers better ascertain the validity of our conclusions.

For the claim that “oligonucleotides are known to lack strong base stacking forces and exhibit flexible spatial conformations”, we cite ref(Maassen, de Ruiter et al. 2018).

19. Maassen, S.J., de Ruiter, M.V., Lindhoud, S. & Cornelissen, J. Oligonucleotide Length-Dependent Formation of Virus-Like Particles. *Chemistry* 24, 7456-7463 (2018).

For saying that “increasing the length of oligonucleotide sequences are different from augmenting the nucleotide concentration, holds the potential to enhance the folding and entanglement probability of the sequences during the reaction process”, we cite the (Chi, Wang et al. 2013, Roth, Glick Azaria et al. 2018) as the foundational references to proof the longer polymers are more flexible than the shorter ones, which easily to twist together.

20. Roth, E., Glick Azaria, A., Girshevitz, O., Bitler, A. & Garini, Y. Measuring the Conformation and Persistence Length of Single-Stranded DNA Using a DNA Origami Structure. *Nano Letters* 18, 6703-6709 (2018).

21. Chi, Q., Wang, G. & Jiang, J. The persistence length and length per base of single-stranded DNA obtained from fluorescence correlation spectroscopy measurements using mean field theory. *Physica A: Statistical Mechanics and its Applications* 392, 1072-1079 (2013).

For the information “the entanglement of longer oligonucleotide chains thus, obtaining higher MNF production”, this is the main hypothesis we proofed in our work. So, we further mentioned it's a hypothesis.

Revision made:

The observed results can be attributed to the inherent characteristics of oligonucleotides, which are known to lack strong base stacking forces and exhibit flexible spatial conformations¹⁸. Consequently, increasing the length of oligonucleotide sequences (which is different from augmenting the nucleotide concentration), holds the potential to enhance the folding and entanglement probability of the sequences during the reaction process^{39, 40}. Hence, we hypothesis the entanglement of longer oligonucleotide chains endowing the intertwined DNA clusters with a higher potential, which in turn enhances the affinity of DNA for calcium ions, thus, obtaining higher MNF production.

8. Pg.11 – Please clarify the meaning of GLUT-1 imaging performance.

We appreciate your suggestions on clarifying the implications of exploring the imaging properties of GLUT-1. Our preparation of DNAzyme molecular beacons was primarily motivated by the following three reasons:

- 1) To observe the binding process of H-GDz upon encountering the target gene, confirming the DNAzyme's ability to recognize the target substrate, specifically the GLUT-1 sequence.
- 2) To showcase the sustained activity of functionalized DNAzymes in MNF materials post internalization by cells. Given that the H-GDz developed in this study serves as a proof of concept, our emphasis was on maximizing the material's functionality and validating its function.
- 3) This study aims to lower the application threshold of MNF materials and leverage the capabilities offered by DNA nanotechnology to realize multifaceted functions. Disease gene detection through DNA molecular beacons has always been a research focal point. Although the GLUT-1 gene selected in this study is abundantly present in all cell types, its detection is not challenging. However, this experiment provides a practical foundation for detecting low-abundance genes in future extensive MNF applications.

Revision made:

Subsequently, we validated the detection capability of the H-GDz molecular beacon for GLUT-1 mRNA (**Fig. 2a**). This molecular beacon was composed of a dual-labeled fluorescent probe, FAM/BHQ-1, where the fluorescence of FAM was effectively quenched by BHQ-1 in the absence of GLUT-1 mRNA through forster resonance energy transfer (FRET) effect. When the target gene was encountered, the molecular beacon underwent a conformational change, resulting in the release of FAM fluorescence and the visualization of the target gene, marked as red color.**The preparation of DNAzyme molecular beacons served to demonstrate the persistent activity of functionalized DNAzymes within MNF materials after cell internalization, validate their target substrate (GLUT-1 mRNA) recognition capability, and harness DNA nanotechnology for diverse MNF applications.** From the results, we observed that the fluorescence intensity of the system increased with increasing concentrations of GLUT-1 mRNA substrate (**Fig. 2a**), which indicated that the H-GDz molecular beacon could specifically recognize and bind to GLUT-1 mRNA, leading to an enhanced fluorescence signal in the presence of the target mRNA.....

Discussion

.....of aptamers in constructing MNF materials, laying the foundation for flexible design of DNA sequences. Thus, our research heralds a new era for MNF materials, brimming with extraordinary possibilities and unparalleled impact in various research and application domains.

The preparation of DNAzyme molecular beacons served three main purposes: confirming their target substrate recognition ability, demonstrating the sustained activity of functionalized DNAzymes in MNF materials post-cellular internalization, and leveraging DNA nanotechnology for multifaceted MNF applications. While GLUT-1 is present in many cell types, our study provides a practical framework for detecting low-abundance genes in future MNF applications.

9. In Figure 2A, the cleaved-GLUT1 mRNA band is difficult to visualize at 2 mM Ca²⁺. Please provide a densitometric analysis of the bands and clarify the identity of the band between H-GDz-GLUT1 and H-GDz. Please also cite references for the detectable range of GLUT-1 mRNA and its physiological relevance.

Note: we switched the order of Figs. 2a and 2b, as mentioned in comments 10.

(1) We appreciate your observation regarding the unclear bands. We conducted quantitative analysis of the substrate and the newly generated bands due to cleavage (the experiment was conducted three times, yielding similar results). Quantification of the bands was performed using ImageJ, based on the results of the three independent experiments. The total content of cleaved and uncleaved substrate chains is not fixed, and a possible reason for this is that after substrate cleavage, some cleaved substrate chains may still bind to the DNzyme. This could result in the observation of a small amount of cleaved GLUT-1 mRNA even when most of the substrate is cleaved under the 1mM calcium ion condition. Data represent the mean ± SD. Statistical significance was calculated through one-way ANOVA using a Tukey post hoc test. P < 0.05 indicated statistically significant. Source data also provided in the Source data file.

Supplementary Figure 6. Densitometric analysis of the bands through ImageJ. (n = 3 independent experiments and the data are presented as mean values ± SD) All statistics were calculated using one-way ANOVA using a Tukey post hoc test. Source data are provided as a Source Data file.

Revision made:

.....However, due to the absence of calcium ions, the mRNA was not cleaved. As the calcium ion concentration was gradually increased, a noticeable decrease in the intensity of the GLUT-1 mRNA band was observed at 0.5 mM (Supplementary Fig.6). At a Ca²⁺ concentration of 2 mM, the appearance of a cleaved GLUT-1 mRNA band was observed, indicating the successful cleavage of the target gene. Furthermore, at a Ca²⁺ concentration of 4 mM, the target GLUT-1 gene was completely silenced (Fig. 2b and Supplementary Fig.6).....

(2) Regarding the identity of the band between H-GDz-GLUT1 and H-GDz, the band may have resulted from the mRNA being cleaved into two segments, with one segment still attached to the DNAzyme, thus positioning its molecular weight between that of H-GDz-GLUT1 and H-GDz. We have created an illustrative diagram of this phenomenon and provided supplementary information on the original image to address this issue.

Revision made:

.....Furthermore, at a Ca^{2+} concentration of 4 mM, the target GLUT-1 gene was completely silenced (**Fig. 2b and Supplementary Fig.3**). We observed a new band between H-GDz-GLUT1 and H-GDz at calcium ion concentrations exceeding 2mM (indicating extensive cleavage of the GLUT-1 substrate). Its molecular weight falls between that of H-GDz-GLUT1 and H-GDz, possibly suggesting the presence of a fragment of the cleaved substrate still attached to the DNAzyme. These findings elucidated that the depleting activity of H-GDz on GLUT-1 mRNA was influenced by the concentration of calcium ions, with higher concentrations leading to more pronounced gene regulation effects.....

(3) For the citation of the detection range of GLUT-1 mRNA and the impact of GLUT-1 mRNA on physiological significance, we have read and referenced the following literature.

Firstly, for the detection of mRNA, it can be divided into extracellular and intracellular (fixed cells/live cells) mRNA detection.

For extracellular mRNA sequence detection, the current lowest detection limit is around 100nM(Wang, Luo et al. 2018)-1uM(Sato, Watanabe et al. 2015), depending on the detection system, and the system concentration. As for intracellular mRNA detection, nowadays, single-molecule fluorescence in situ hybridization (FISH) technology can be used to detect individual mRNA in fixed cells(Femino, Fay et al. 1998) and drosophila embryos(Trcek, Lionnet et al. 2017). For live-cell live imaging, various techniques can amplify mRNA detection signals and achieve low abundance mRNA detection(Pan, Cha et al. 2017, Sheng, Zhao et al. 2022), but there are few descriptions of the detection limit of intracellular mRNA concentration. More analysis of its physiological relevance is carried out through PCR for relative quantification, presenting the results as relative values (such as relative internal reference genes) (Lu, Wu et al. 2018).

Regarding the impact of GLUT-1 mRNA concentration on physiological significance, Glut-1 is commonly described as ubiquitous presence, but its expression, like other glucose transporter

subtypes, has tissue specificity (Joost and Thorens 2001). Apart from specific high glucose uptake tissues such as astrocytes, high GLUT-1 expression has always been an indicator of carcinogenesis (Glut-1 can be detected in the necrotic area of many human tumor types, but not in normal tissues) (Airley, Evans et al. 2010, Lu, Wu et al. 2018).

We have incorporated this information into the manuscript, hope to address all the concerns raised by the reviewer.

Revision made:

First, glut-1 is generally described as ubiquitous present, but like other glucose transporter subtypes, its expression has tissue specificity (Joost and Thorens 2001). In addition to specific high glucose uptake tissues, such as astrocytes, high levels of GLUT-1 expression have been an indicator of carcinogenesis (Glut-1 can be detected in the necrotic areas of many human tumor types, but not in normal tissues) (Airley, Evans et al. 2010, Lu, Wu et al. 2018). Hence, here we validated the detection capability of the H-GDz molecular beacon for GLUT-1 mRNA (**Fig. 2a**). This molecular beacon was composed of a dual-labeled fluorescent probe, FAM/BHQ-1, where the fluorescence of FAM was effectively quenched by BHQ-1 in the absence of GLUT-1 mRNA through forster resonance energy transfer (FRET) effect. When the target gene was encountered, the molecular beacon underwent a conformational change, resulting in the release of FAM fluorescence and the visualization of the target gene, marked as red color.

For mRNA detection, the current detection limit is approximately 100nM (Wang, Luo et al. 2018)-1 μ M (Sato, Watanabe et al. 2015), depending on the detection system and system concentration. For intracellular mRNA detection, single-molecule fluorescence in situ hybridization (FISH) technology is now capable of detecting individual mRNA in fixed cells (Femino, Fay et al. 1998, Trcek, Lionnet et al. 2017). Various techniques for live-cell real-time imaging can amplify mRNA detection signals and achieve low-abundance mRNA detection (Pan, Cha et al. 2017, Sheng, Zhao et al. 2022). In this work, the preparation of DNAzyme molecular beacons served to demonstrate the persistent activity of functionalized DNAzymes within MNF materials after cell internalization, validate their target substrate (GLUT-1 mRNA) recognition capability, and harness DNA nanotechnology for diverse MNF applications. From the results, we observed that the fluorescence intensity of the system increased with increasing concentrations of GLUT-1 mRNA substrate (**Fig. 2a**), which indicated that the H-GDz molecular beacon could specifically recognize and bind to GLUT-1 mRNA, leading to an enhanced fluorescence signal in the presence of the target mRNA.

For methods part:

Polyacrylamide gel electrophoresis (PAGE) assays

.....For different groups, the incubation is performed at 37°C, for 3 hours with 1 μ M of H-GDz and 2 μ M of substrate under different metal ion concentrations (0, 0.1, 0.5, 1, 2, 4 mM).

10. The authors should consider switching the order of Figs. 2A and 2B. In Figure 2E, please indicate what the red color represents in the figure legend.

We appreciate your suggestion and agree that this order better aligns with the logical. Consequently, we have adjusted Figures 2a and 2b and made a series of modifications in the manuscript. We also indicate what the red color represents in the figure legend, as reviewer asked.

Revision made:

For Figure 2a and 2b:

For Figure 2e:

Figure 2.....(e) Fluorescence confocal microscopy was employed to evaluate the detection and regulation capabilities of various materials on the GLUT-1 gene in the cellular context. The intracellular GLUT-1 protein was labeled with antibody and represented as yellow fluorescence, the DNzyme was labeled as red, while the cell nuclei were stained with DAPI and shown as blue color.....

11. The data regarding the lysosomal escape of the MNFs (Figure S4) is difficult to interpret. Please include data to show the intracellular trafficking of the MNFs to confirm whether they will be localized in the lysosomes.

We appreciate your concern regarding our lysosome escape data. We acknowledge the reviewer's observations, including the issue of excessively high fluorescence intensity of H-GDz and the over-saturation of images, which may impact the accuracy of Pearson's correlation for colocalization. Inspired by the reviewer's comments and in response to your request, we re-conducted the lysosome escape experiment. We ensured that the DNA fluorescence intensity was appropriate and zoomed in on the cells to facilitate observation of lysosomal status. We used both Rcoloc and line-scan analysis to calculate the positioning between DNA and lysosomes. We hope these data could satisfy the reviewers.

During the experiment, we initially co-incubated the MNF material with cells for one hour and replaced it with blank culture medium at the 1-hour timepoint. Subsequently, at later 0 hours, 1.5 hours, and 7.5 hours timepoint, we stained the lysosomes of the cells, with a staining duration of 30 minutes. We performed simultaneous imaging of the drug and lysosomes, with imaging time points recorded as 30 minutes, 2 hours, and 8 hours. Lysosomes were labeled with LysoTracker™ Green DND-26, (Sun, Chang et al. 2020) special packaging, Catalog number: L7526, showing green fluorescent signals. DNA sequences were labeled with Cy5.5, displaying red signals. Cell nuclei were stained with DAPI, shown as blue. We calculated the Rcoloc in the results, indicating that within 30 minutes and 2 hours, DNA fluorescence and lysosome localization were good, with Rcoloc values of 0.7027 and 0.5570, respectively. However, at 8 hours, the Rcoloc value between Cy5.5 and LysoTracker was -0.5592, suggesting DNA was released from lysosomes at 8 hours. Furthermore, we zoomed in on individual cells and performed topological analysis of fluorescence intensity along the marked white lines, revealing better overlap between DNA and lysosomes at 2 hours and reduced overlap at 8 hours. The above experiments were independently repeated three times, yielding similar results. Topological data is provided in the source data file.

The new result figures are presented below as Supplementary Figure 8. We have also provided a description in the main text and updated the experimental details in the methods section.

Revision made:

In main text:

..... The endocytic ability of each material group can be reflected by the detected GLUT-1 mRNA (represented by the red signal of FAM), as higher endocytic intensity corresponds to a stronger GLUT-1 detection signal. Moreover, from the results of lysosomal escape (**Supplementary Fig.8**), it can be observed that there is a better overlap between MNF and lysosomes at 2 hours and reduced overlap at 8 hours, reflecting that the MNF nano-system were able to effectively escape from the lysosomes at 8 hours. Therefore, fluorescent confocal microscopy (**Fig. 2e**) and flow cytometry (**Fig. 2f**) were used to evaluate the nanomaterial groups (H-GDz/Ca and MH-GDz/Ca) as well as the pure DNA sequence groups (H-GDz + Ca²⁺ and MH-GDz + Ca²⁺) gene regulation ability at 10 h timepoint. The concentration of free calcium ions was set to be consistent with the nanomaterials.....

In Methods:

Lysosome escape

For the assessment of lysosomal escape, NIC-N87 cells were cultured overnight on confocal dishes. H-GDz/Ca NPs labeled with Cy5.5 were utilized in the experiment. The MNF material was initially co-incubated with cells for one hour, and it was replaced with blank culture medium at the 1-hour mark. Subsequently, at 0 hours, 1.5 hours, and 7.5 hours, the lysosomes of the cells were stained for a duration of 30 minutes. Simultaneous imaging of the drug and lysosomes was performed, with imaging time points recorded as 30 minutes, 2 hours, and 8 hours. Lysosomes were labeled with LysoTracker™ Green DND-26, special packaging, Catalog number: L7526, exhibiting green fluorescent signals. DNA sequences within MNFs were labeled with Cy5.5, presenting red signals. Cell nuclei were stained with DAPI, appearing blue. The cells were examined using a fluorescence confocal microscope. ImageJ software was employed to carry out the topological analysis of fluorescence intensity along the marked white lines, which provided insights into the degree of overlap between the NPs and lysosomes in each experimental group.

In Supporting Information:

Supplementary Figure 8. Lysosomal escape experiment of MNF material. The MNF material was synthesized using Cy5.5-labeled H-GDz, allowing the observation of the intracellular localization of MNF material through the Cy5.5 channel (red signal). Lysosomes within the cells were detected using the green channel of LysoTracker™ Green DND-26. The co-localization analysis of the two channels within the cells was performed using the Coloc 2 plugin and line-scan analysis in ImageJ. (n = 3 independent experiments, with similar results. Source data are provided as a Source Data file.)

12. Please clarify why the F-intensity of H-GDz is increasing at the 8-h time point and provide details on how this assay was performed. Were the MNFs removed from the cells at a fixed time before analysis at the 2-h and 8-h time points? Since the F-intensity highly influences the Pearson's correlation for colocalization, it is unclear whether the colocalization analysis is valid and that the lysosomal escape of the MNFs can be concluded. Please provide the sample size used to draw this claim and add scale bars to the images.

We appreciate the reviewer's highly meaningful questions. Indeed, as pointed out by the reviewer, the F-intensity significantly influences the Pearson's correlation for colocalization. As mentioned in our previous response to the question 11, in our earlier experiments, we did not remove MNFs at fixed time points but co-incubated MNFs with cells for 2 hours and 8 hours, respectively. Although this approach is feasible (Xu, Liu et al. 2019), it led to oversaturation of drug internalization at 8 hours, impacting the accuracy of Pearson's correlation for colocalization. We acknowledge this issue, and in order to address it, we have redesigned and re-conducted the experiments using a more appropriate methodology to ensure the reliability of the results. The detailed experimental results and process is as follows.

Supplementary Figure 8. Lysosomal escape experiment of MNF material. The MNF material was synthesized using Cy5.5-labeled H-GDz, allowing the observation of the intracellular localization of MNF material through the Cy5.5 channel (red signal). Lysosomes within the cells were detected using the green channel of LysoTracker™ Green DND-26. The co-localization analysis of the two channels within the cells was performed using the Coloc 2 plugin and line-scan analysis in ImageJ. (n = 3 independent experiments, with similar results. Source data are provided as a Source Data file.)

Revision made:

In main text:

..... The endocytic ability of each material group can be reflected by the detected GLUT-1 mRNA (represented by the red signal of FAM), as higher endocytic intensity corresponds to a stronger GLUT-1 detection signal. Moreover, from the results of lysosomal escape (**Supplementary Fig.8**), it can be observed that there is a better overlap between MNF and lysosomes at 2 hours and reduced overlap at 8 hours, reflecting that the MNF nano-system were able to effectively escape from the lysosomes at 8 hours. Therefore, fluorescent confocal microscopy (**Fig. 2e**) and flow cytometry (**Fig. 2f**) were used to evaluate the nanomaterial groups (H-GDz/Ca and MH-GDz/Ca) as well as the pure DNA sequence groups (H-GDz + Ca²⁺ and MH-GDz + Ca²⁺) gene regulation ability at 10 h timepoint. The concentration of free calcium ions was set to be consistent with the nanomaterials.....

In Methods:

Lysosome escape

For the assessment of lysosomal escape, NIC-N87 cells were cultured overnight on confocal dishes. H-GDz/Ca NPs labeled with Cy5.5 were utilized in the experiment. The MNF material was initially co-incubated with cells for one hour, and it was replaced with blank culture medium at the 1-hour mark. Subsequently, at 0 hours, 1.5 hours, and 7.5 hours, the lysosomes of the cells were stained for a duration of 30 minutes. Simultaneous imaging of the drug and lysosomes was performed, with imaging time points recorded as 30 minutes, 2 hours, and 8 hours. Lysosomes were labeled with LysoTracker™ Green DND-26, special packaging, Catalog number: L7526, exhibiting green fluorescent signals. DNA sequences were labeled with Cy5.5, presenting red signals. Cell nuclei were stained with DAPI, appearing blue. The cells were examined using a fluorescence confocal microscope. ImageJ software was employed to carry out the topological analysis of fluorescence intensity along the marked white lines, which provided insights into the degree of overlap between the NPs and lysosomes in each experimental group.

13. Please provide higher magnification images to support the claims associated with Figure 2F; specify the sample size and the number of biological replicates in the data associated with Figure 3.

We thank the reviewer for pointing out that high-magnification images are needed to support the conclusions of Figure 2F. As suggested by the reviewer, we have conducted experiments again for these five groups, employing a higher magnification to facilitate a clear observation of intracellular protein expression and fluorescence signals for readers. The new images were shown below as Supplementary Figure 8.

Moreover, we specified the sample size and the number of biological replicates in the legend of Figure 3.

Revision made:

In main text:

.....From the results in **Fig. 2e and** higher magnification images in **Supplementary Fig.9**, compared with MH-GDz/Ca NPs group, the H-GDz/Ca NPs group exhibited 2.95 times (**Supplementary Fig.5**) higher efficient GLUT-1 detection, suggesting great cell internalization of H-GDz/Ca NPs. In contrast, the cellular uptake of free DNA sequences was significantly diminished due to their.....

Fig. 3 (j) Western blot analysis confirming the modulation of intracellular RAD51 by different nanomaterials. (k) Assessment of nuclear DNA damage. *The Mean Gray Value (average gray value within the selection) is calculated using ImageJ to determine the average intensity of pixels within the selected area, and it is independent of cell density. The data presented in the figures represent the mean values with standard deviation (mean \pm SD). Statistical comparisons were performed using one-way ANOVA, followed by the Tukey post-hoc test for multiple comparisons. The experiments in (d, e, f, g, i, j, k) were repeated three times independently with similar results. The source data underlying figures a, b, d, e, f, h, i, k is available in the provided Source Data file.*

Supplementary Figure 9. Higher magnification to facilitate a clear observation of intracellular protein expression and fluorescence signals. (n = 3 independent experiments, with similar results.)

14. Please include particle characterization after IRF-1 loading (e.g., size and morphology).

We thank you for bringing up this critical point. Indeed, we did not provide a detailed characterization of the protein-loaded MNF materials, and we sincerely apologize for this oversight. To address this important point in our study, we conducted both Dynamic Light Scattering (DLS) analysis and Transmission Electron Microscopy (TEM) imaging of the protein-loaded MNF particle. The DLS analysis revealed that the average particle size of the nanomaterials increased from around 130 nm (non-loading) to about 200 nm after protein loading. Additionally, TEM images further confirmed these observations, illustrating the morphology and distribution of the IRF-1 loaded particle. These results provide detailed insights into the impact of protein loading on the structure of the MNF, offering crucial information for our understanding of their properties. We thank reviewer for mention this great concern, and we incorporated a further description of these experimental results in the revised manuscript.

Supplementary Figure 14. Characterization of H-GDz/Ca MNF materials after loading IRF-1 protein. (n = 3 independent experiments with similar results, Source data are provided as a Source Data file.)

Revision made:

In main text:

.....After confirming the superior gene regulation capability of HER-2 aptamer-contained MNF nano-system (H-GDz/Ca), we embarked on an investigation into its loading content of the DNA homologous recombination repair protein RAD51 inhibitor, IRF-1. We initially characterized the size and morphology of MNF materials after loading with protein drugs (**Supplementary Fig. 14**). DLS analysis showed that the average particle size of the IRF/H-GDz/Ca increased from around 130 nm (H-GDz/Ca) to about 200 nm. Additionally, TEM images further confirmed these observations, illustrating the uniform spherical structure and distribution of the IRF-1 loaded particles. Subsequently, employing flow cytometry techniques, the inhibitory efficacy of IRF/H-GDz/Ca system on RAD51 were examined within NIC-N87 cells (**Fig. 3a**). The findings unveiled a substantial reduction in RAD51 expression upon achieving an IRF-1 concentration of 100 ng/mL. Consequently, guided by the therapeutic dose of IRF-1 and the requisite DNAzyme dosage (500 nM).....

15. Pertaining to Fig. 3D, explain why the Ca²⁺ levels are similar between MNF with and without the HER-2 aptamer.

We appreciate the reviewer for raising this concern. It might have arisen due to our unclear explanation, leading to a misunderstanding. In fact, in the five groups described in Fig. 3D (Control, H-MGDz/Ca, H-GDz/Ca, IRF/H-MGDz/Ca, and IRF/H-GDz/Ca), apart from the control group, all the other groups possess the HER-2 aptamer. The distinction lies in the presence or absence of GDz and IRF.

In Figure 3, we designed these five groups to investigate the therapeutic effects of each element in the material, as Figure 2 has already confirmed the targeting function of the HER-2 aptamer. We emphasized this point in the manuscript, hoping to address the reviewer's concerns.

Revision made:

.....Five groups including Control, H-MGDz/Ca, H-GDz/Ca, IRF/H-MGDz/Ca, and IRF/H-GDz/Ca were designed, and MGDz sequence is a mutated sequence of GLUT-1 DNazymes with same nuclear bases. All the other groups possess the HER-2 aptamer, while the distinction lies in the presence or absence of GDz and IRF. First, the intracellular calcium levels were characterized (Fig. 3d). The results showed that the four nano-groups exhibited similar intracellular calcium levels compared to the control group.....

16. Please clarify how the mean gray values of the images were determined. Some of the images (IRF/H-GDz/Ca (Fig. 3E) and controls (Fig. 3D, 3I and 3K) appear to have different cellular densities. Please clarify what threshold and sample size were used to collect the images.

We appreciate the reviewer for bringing up the issue regarding the specific definition of ImageJ software parameters. As mentioned by the reviewer, different images contain varying cell densities, making it meaningful to choose the average mean gray value of the cell area rather than the Integrated density.

From the ImageJ user guide:

Mean gray value: Average gray value within the selection. This is the sum of the gray values of all the pixels in the selection divided by the number of pixels.

Integrated density: The sum of the values of the pixels in the image or selection. This is equivalent to the product of Area and Mean Gray Value.

In our case, based on our procedure, taking each group from Fig 3e as an example, we initially set the threshold and demarcated the regions containing cells in each image. Subsequently, we measured the average grayscale value within these demarcated areas. Consequently, this result can reflect the grayscale values within each cell, unaffected by the cell density.

Editorial Note: Program screenshot
has been redacted

Revision made:

Fig. 3 Investigation of the synergistic therapeutic effects of IRF-1 in conjunction with GLUT-1 inhibition materials. (K) Assessment of nuclear DNA damage. The Mean Gray Value (average gray value within the selection) is calculated using ImageJ to determine the average intensity of pixels within the selected area, and it is independent of cell density. The data presented in the figures represent the mean values with standard deviation (mean \pm SD). Statistical comparisons were performed using one-way ANOVA, followed by the Tukey post-hoc test for multiple comparisons. The experiments in (d, e, f, g, i, j, k) were repeated three times independently with similar results. The source data underlying figures a, b, d, e, f, h, i, k is available in the provided Source Data file.

17. Please consider investigating the uptake of the MNFs in non-HER-2 expressing cell lines in order to make the claim that the specific targeting of gastric cancer cells and elucidate the cellular uptake pathway analysis to increase the quality of the manuscript.

We appreciate the excellent suggestion from the reviewer on how to clearly demonstrate that HER-2 targeted MNF materials exhibit higher selectivity for HER-2-positive cell lines. We have accepted the reviewer's advice and conducted a comparative study on the internalization of H-GDz/Ca MNF materials in Normal Human Dermal Fibroblasts (NHDF) cell, chosen as a HER-2 low-expressing cell line, and N87 cells as HER-2 positive cell line. We employed two methods, including confocal observation of the fluorescence intensity in internalized cells and measurement of internalization using flow cytometry. The experimental results are presented in the supplementary material.

Supplementary Figure 13. The internalization experiments of MNF materials on different HER-2 expressing cell lines involved confocal microscopy and flow cytometry. Confocal experiments and flow cytometry techniques were employed to validate the uptake of H-GDz/Ca MNF NPs by both human dermal fibroblast cells (NHDF) and gastric cancer N87 cells. The H-GDz sequence was labeled with Cy5.5 and shown as red signal, while the cell nuclear was labeled by DAPI (Blue color). (n = 3 independent experiments with similar results.)

Revision made:

Flow cytometry analysis further confirmed the cell-targeting ability of the HER-2 integrated materials (**Fig. 2f**). Compared with control group, GLUT-1 expression was reduced by approximately 37.5% in the H-GDz + Ca²⁺ group, while significantly reduced by up to 70% in the H-GDz/Ca NPs group. Simultaneously, a noticeable shift of cells from the Q3 region (71.0 %) characterized by high GLUT-1 expression and low GLUT-1 detection towards the Q1 region (41.3 %) was observed. Therefore, these findings provided compelling evidence that the transformation of H-GDz into nanostructured MNF materials through self-mineralization techniques can significantly enhance its intracellular gene monitoring and regulating capabilities. In addition to validating the impact of the aptamer sequence's accuracy on endocytosis, we also examined the internalization of MNF materials by N87 cells versus HER-2-negative normal human dermal fibroblasts (NHDF) (**Supplementary Fig. 13**). The results similarly demonstrated the selectivity of MNF materials for HER-2-positive cell lines.

18. Please specify the sample size and number of biological replicates to understand the reproducibility of the results.

We appreciate the reviewer's attention to sample size and biological replicates. We have provided a detailed description of the number of repetitions for each experiment, the statistical analysis methods, and whether the data are available in the source data for each figure. We thank the reviewer for their efforts to enhance the quality of our manuscript.

Revision made:

Fig. 1. Mechanism of MNF synthesis based on long-chain DNA binding with calcium ions and regulation of MNF properties. Statistics in (j) were calculated using two-tailed paired t tests, and the experiments in (a, h, i, j, k) were repeated three times independently with similar results. The source data from (i, j, k) are provided as a Source Data file.

Fig. 2. Validation of the targeting ability of each material towards NIC-N87 cells and characterization of their detection and regulation capabilities towards GLUT-1. Western blot analysis to demonstrate the silencing efficacy of different groups on GLUT-1 expression. Statistics in (f) were calculated using two-tailed paired t tests, and the experiments in (a, b, c, e, f, g) were repeated three times independently with similar results. The source data from (f, g) are provided as a Source Data file.

Fig. 3 Investigation of the synergistic therapeutic effects of IRF-1 in conjunction with GLUT-1 inhibition materials. (K) Assessment of nuclear DNA damage. The Mean Gray Value (average gray value within the selection) is calculated using ImageJ to determine the average intensity of pixels within the selected area, and it is independent of cell density. The data presented in the figures represent the mean values with standard deviation (mean \pm SD). Statistical comparisons were performed using one-way ANOVA, followed by the Tukey post-hoc test for multiple comparisons. The experiments in (d, e, f, g, i, j, k) were repeated three times independently with similar results. The source data underlying figures a, b, d, e, f, h, i, k is available in the provided Source Data file.

Fig. 4. Distribution and therapeutic efficacy of nanomaterials in mice. Body weight changes during the treatment period. All statistics (b, f, g) were calculated using one-way ANOVA using a Tukey post hoc test, and the experiments in (a, b, c) were repeated three times independently with similar results. The source data from (b, e-h) are provided as a Source Data file.

Figure 5. The protein regulation in the in-situ tumor model and the characterization of MNF materials in the PDX model, along with their therapeutic efficacy. (a) Immunohistochemical characterization of the expression regulation of GLUT-1 and RAD51 in different treatment groups. (b) Expression of GLUT-1 in liver and kidney tissues. (c) Therapeutic efficacy of different treatment groups in the PDX model. (d) Representation of ROS in tumor slice tissues stained with Dihydroethidium (DHE), marked in red. (e) Characterization of overall three-dimensional nuclear damage in tumor slices. (f) Labeling of CAF cells in tumor slices. All results were repeated three times with similar results.

Supplementary Figure 2. The persistence length of short strand GDz, and the persistence length of long strand H-GDz. The source data is available in the provided Source Data file.

Supplementary Figure 6. Densitometric analysis of the bands through ImageJ. (n = 3 independent experiments and the data are presented as mean values \pm SD) All statistics were calculated using one-way ANOVA using a Tukey post hoc test. Source data are provided as a Source Data file.

Supplementary Figure 8. Lysosomal escape experiment of MNF material. The MNF material was synthesized using Cy5.5-labeled H-GDz, allowing the observation of the intracellular localization of MNF material through the Cy5.5 channel (red signal). Lysosomes within the cells were detected using the green channel of LysoTracker™ Green DND-26. The co-localization analysis of the two channels within the cells was performed using the Coloc 2 plugin and line-

scan analysis in ImageJ. (n = 3 independent experiments, with similar results. Source data are provided as a Source Data file.)

Supplementary Figure 9. Higher magnification to facilitate a clear observation of intracellular protein expression and fluorescence signals. (n = 3 independent experiments, with similar results.)

Supplementary Figure 10. Quantification of fluorescence signal intensity in confocal microscopy experiments using ImageJ. (A) Quantification of fluorescence intensity emitted in the FAM channel (470/525 nm) after binding of GDz to mRNA. (B) Immunofluorescence experiment targeting GLUT-1, labeled in the TRICT channel (557/576 nm). (n = 3 independent experiments and the data are presented as mean values \pm SD) Statistics were calculated using two-tailed paired t-test for the two interested group. Source data are provided as a Source Data file.

Supplementary Figure 11. The experiments with MNF materials on SNU216 cells. The H-GDz sequence was labeled with Cy5.5 and shown as red signal, while the cell nuclear was labeled by DAPI (Blue color) (a) Confocal experiments to confirm the cellular uptake of each material. (b) Flow cytometry techniques to verify the internalization of nanomaterials by SNU216 cells. (c) Assessment of the regulation of GLUT-1 within cells for each material. (n = 3 independent experiments with similar results)

Supplementary Figure 13. The internalization experiments of MNF materials on different HER-2 expressing cell lines involved confocal microscopy and flow cytometry. Confocal experiments and flow cytometry techniques were employed to validate the uptake of H-GDz/Ca MNF NPs by both human dermal fibroblast cells (NHDF) and gastric cancer N87 cells. The H-GDz sequence was labeled with Cy5.5 and shown as red signal, while the cell nuclear was labeled by DAPI (Blue color). (n = 3 independent experiments with similar results.)

Supplementary Figure 14. Characterization of H-GDz/Ca MNF materials after loading IRF-1 protein. (n = 3 independent experiments with similar results, Source data are provided as a Source Data file.)

Supplementary Figure 15. Regulation of metabolic pathways in N87 cells by different MNF materials. Characterization of GLUT-1 protein expression in N87 cells by different material groups, along with downstream characterization of PPP pathway and hexose phosphorylation pathways. (n = 3 independent experiments and the data are presented as mean values \pm SD) All statistics were calculated using one-way ANOVA using a Tukey post hoc test. Source data are provided as a Source Data file.

Supplementary Figure 16. Regulation of metabolic pathways in SNU216 cells by different MNF materials. (a) Characterization of intracellular PPP metabolism, NADH, and Cysteine consumption processes, as well as GSH synthesis in SNU216 cells. (b) Analysis of the nuclear damage marker γ -H2AX using flow cytometry. (c) Cytotoxicity experiments of different material groups on SNU216 cells. (n = 3 independent experiments and the data are presented as mean values \pm SD) All statistics were calculated using one-way ANOVA using a Tukey post hoc test. Source data are provided as a Source Data file.

Supplementary Figure 17. Observation of nuclear damage in cells under different treatment conditions using TEM. In the control and H-MGDz/Ca groups, the nuclei appear intact with evenly distributed internal DNA. In the H-GDz/Ca group, there are few black aggregates within the DNA, indicating DNA damage and condensation. In the IRF/H-MGDz/Ca group, nuclear morphology is distorted, and numerous black aggregates are present, indicating nuclear damage. In the IRF/H-GDz/Ca group, severe nuclear condensation is observed, with extensive DNA damage within the nuclei. (n = 3 independent experiments, with similar results.)

Supplementary Figure 18. Cytotoxicity experiments of each MNF material group on N87 cells. (n = 3 independent experiments and the data are presented as mean values \pm SD) Source data are provided as a Source Data file.

Supplementary Figure 19. Pharmacokinetics of IRF/H-GDz/Ca and IRF/MH-GDz/Ca NPs through blood circulation. (n = 3 independent experiments and the data are presented as mean values \pm SD) Source data are provided as a Source Data file.

Supplementary Figure 20. Intravenous administration of MNF materials and their distribution in mice at 96h timepoint. (n = 3 independent experiments with similar results).

Supplementary Figure 21. Imaging of the materials in the tumor site of mice, DAPI staining was employed to label the cell nuclei. (n = 3 independent experiments with similar results).

Supplementary Figure 23. H&E stained tissue sections of heart, liver, spleen, lung and kidney. (n = 3 independent experiments with similar results).

Supplementary Figure 24. The tissue sections from the constructed PDX model were used for HER-2 staining. (n = 3 independent experiments with similar results).

Supplementary Figure 25. Tumor growth and weight for different groups for the PDX model. (n = 3 independent experiments and the data are presented as mean values \pm SD) Source data are provided as a Source Data file.

Supplementary Figure 26. Immunofluorescence detection of P-glycoprotein (P-gp) for the PDX model. (n = 3 independent experiments with similar results.)

Supplementary Figure 27. The blood biochemistry and Hemolysis test. (a) After intravenous injection of different NPs, the blood biochemistry of mice was measured, with untreated mice as the control. (b) Hemolysis test conducted for the IRF/H-GDz/Ca group. ALT: Alanine Aminotransferase; AST: Aspartate Aminotransferase; Cr: Creatinine, TBIL: Total Bilirubin. (n = 3 independent experiments and the data are presented as mean values \pm SD) All statistics were calculated using one-way ANOVA using a Tukey post hoc test. Source data are provided as a Source Data file.

Figure 4

19. There is a lack of information in the Materials and Methods for the in vivo studies, which makes it difficult to evaluate the results. What is the strain/age/sex? Inoculation of human cells in naïve mice will cause host responses and rejection thus the results may not be valid. Is a mice xenograft model used? How long was the tumor incubated before treatment?

We thank you for pointing out the shortcomings in our description of animal experiments. In our *in vivo* studies, we utilized BALBc-nu nude mice (Female, 6–8 weeks old), a well-established immunocompromised strain commonly used for xenograft models. The mice were female to maintain consistency, however, the results weren't affected by gender. The inoculation involved subcutaneous injection of 5 million N87 cells to establish a subcutaneous tumor model. The use of nude mice helps mitigate host responses and rejection issues. When the tumor volume reached approximately 100 mm³, we removed the tumor and cut it into uniformly sized 1 mm³ tumor blocks for the construction of orthotopic tumor models.

We have incorporated this information into the Materials and Methods section to enhance clarity and facilitate a comprehensive evaluation of our study.

Revision made:

Ethical statement

Our study strictly followed ethical guidelines, with all animal procedures conducted in accordance with the Guidelines for the Care and Use of Laboratory Animals. Approval was obtained from the Institutional Animal Care and Use Committee (IACUC) of () under the reference number (). Approval for human experiment was granted by the Ethics Committee of () (Approval No.:). Written and informed consent was procured from patient before the collection of tissue samples. All experimental procedures adhered to ethical standards and international guidelines, including the "International Ethical Guidelines for Biomedical Research Involving Human subjects" "Declaration of Helsinki".

Gastric Cancer Orthotopic model

First, a subcutaneous tumor model was established by injecting 1×10^6 N87 gastric cancer cells labeled with Luciferase/GFP, into the BALBc-nu strain (Female, 6–8 weeks old). Once the tumor volume reached approximately 100 mm³, the tumor was excised and cut into 1 mm³ tumor tissue blocks. Nude mice of the BALBc-nu strain (6–8 weeks old) were anesthetized and positioned in the left lateral decubitus position. After standard skin disinfection, a 1 cm oblique incision was made below the right costal margin to access the abdominal cavity. Subsequently, the abdominal cavity was opened to expose the gastric organs. Tumor tissue blocks were sutured to the gastric surface to minimize tissue damage. If necessary, hemostasis was performed, and the incision was sutured to close the abdominal cavity. Finally, the surgical site was disinfected again, and the mice were placed in a warm incubator or recovery area for postoperative care, adhering to ethical and animal care guidelines. The mice were maintained under specific conditions: a 12/12 day/night cycle, humidity at 50-60%, and a temperature of 22-26°C. The tumor-bearing mice were randomly allocated into five groups once the tumor volume reached approximately 100 mm³. While female animals were selected for the experiments, the results are not gender-restricted. Humane endpoints include conditions such as tumor burden exceeding 10% of normal body weight, weight loss in

animals exceeding 20% of their normal weight, and persistent self-harm by the animals. When humane endpoints are reached, euthanasia is performed through cervical dislocation under deep anesthesia.

Patient-derived xenografts

We obtained fresh gastric cancer tumor tissue from patients undergoing clinical surgery. The tumor tissue was placed in a tissue storage solution and transported at 4°C. Tumor modeling was initiated within 2 hours. During modeling, necrotic tumor tissue was first removed, and the tumor was cut into small pieces of 3 mm x 3 mm x 3 mm. The tumor tissue blocks were implanted subcutaneously in the right anterior shoulder of the mice. All PDX experiments were conducted in 6-8-week-old female NOG (NOD.Cg-Prkdc^{scid} IL2rg^{tm1Sug}/JicCr1) mice provided by Beijing Vital River Laboratory Animal Technology Co., Ltd. Tumor growth was observed daily to obtain the first-generation gastric cancer-bearing mice. When the subcutaneous tumors in NOG mice grew to approximately 300 mm³, the skin was incised, and the subcutaneous tumors were dissected. Part of the tumor tissue was used as a source for passaging. This process was repeated until stable third-generation PDX-bearing mice were obtained for subsequent experiments. The gastric cancer tumor tissue was validated for HER-2 positive characteristics through immunohistochemistry. All mice were housed and bred in a specific pathogen-free (SPF) facility with controlled temperature (22°C) and lighting (12:12 hours light-dark cycle). The tumor-bearing mice were randomly assigned to five groups when the tumor volume reached approximately 100 mm³. The experiment was approved by the Ethics Committee of () (Approval No.:), and informed, written consent was obtained from patients before collecting tissue. Ethical considerations include defining humane endpoints, which encompass scenarios where the tumor burden exceeds 10% of the normal body weight, weight loss surpasses 20% of the normal weight, or persistent self-harm by the animals is observed. In such cases, euthanasia is carried out through cervical dislocation under deep anesthesia.

20. Fig.4 D, E, F and G represent the same results (Tumor reduction). The authors should include data on the effect of this treatment on the tumor microenvironment.

We appreciate the reviewer for pointing out the singularity of our results presentation. Indeed, the tumor microenvironment (comprising tumor cells, immune cells, and Cancer-Associated Fibroblasts CAF cells) plays a crucial role in understanding the material's inhibitory effects on tumors. Due to our utilization of a PDX model based on severely immunodeficient NOG (NOD.Cg-Prkdcscid IL2rgtm1Sug/JicCr1) mice, exploring CAFs is feasible. We employed α -smooth muscle actin (α -sma) (green signal) and fibroblast activation protein (FAP) (yellow signal) staining to characterize CAF cells. The results indicate that the inhibition of GLUT-1 does decrease the differentiation of CAF cells, while IRF-1 does not inhibit CAF differentiation. We add this information into the main text as Figure 5f.

Revision made:

.....in **Fig. 5e**. The results indicate that IRF/H-GDz/Ca MNF NPs can mediate comprehensive nuclear damage throughout the three-dimensional tumor, providing strong evidence of the therapeutic effectiveness of the material.

On the other hand, the regulation of the tumor microenvironment is crucial for the treatment of PDX tumor models. Cancer-associated fibroblast (CAF) cells in the tumor were labeled using α -smooth muscle actin (α -SMA) (green signal) and fibroblast activation protein (FAP) (yellow signal). We observed that the inhibition of GLUT-1 downregulates the expression of CAF cells; however, IRF-1 does not inhibit CAF (**Fig. 5f**). This indicates that our designed MNF materials can achieve the inhibition of CAFs by suppressing GLUT-1, thereby enhancing therapeutic efficacy.

Moreover, it has been reported that the presence of calcium.....

21. The data associated with Figure S7 is key to assessing the functionality of MNFs (GLUT-1 and RAD51 inhibition) in vivo. Figure S7 should be placed in the main manuscript. The image of IRF/H-GDz/CA should also represent the area of tissue as other groups. Please indicate what the dotted circle represents.

We appreciate the reviewer's recognition of the importance of the Figure S7 in assessing the functionality of MNFs in vivo. We have moved Figure S7 to the main manuscript as Figure 5a. Additionally, we have adjusted the image of IRF/H-GDz/Ca to represent the area of tissue similarly to other groups. The dotted circle now indicates the region of tumor, and this clarification has been added to the figure legend for better understanding.

Revision made:

Figure 5. Protein expression modulation of MNF materials across various tissues and characterization of tumor biomarkers in the further PDX model. (a) H&E-stained tumor tissue sections, as well as immunohistochemistry experiments for GLUT-1 and RAD51. (n=3 independent experiments, with similar results. The dashed line indicates the marked tumor tissue.) (b).....

22. Including ROS measurement/detection in vivo would increase the quality of the manuscript. This could help with interpretation of the in vivo results.

We appreciate your suggestion to assess the levels of Reactive Oxygen Species (ROS) within the tumor tissue. Indeed, evaluating ROS levels in vivo provides a more reflective measure of the impact of our designed MNF materials on the tumor tissue. During the study, 5 μm thickness slides were incubated with Dihydroethidium (DHE) as described by the manufacturer. Fluorescent images were analyzed using Zeiss 880 laser scanning confocal microscope.

Revision made:

In Figure 5d

Figure 5. The protein regulation in the in-situ tumor model and the characterization of MNF materials in the PDX model, along with their therapeutic efficacy..... (d) Representation of ROS in tumor slice tissues stained with Dihydroethidium (DHE), marked in red All results were repeated three times with similar results.

In main text

.....IRF/H-GDz/Ca, could significantly inhibit tumor growth, and the body weight of mice in all groups did not decrease.

Subsequently, we investigated the intratumoral levels of ROS (**Fig. 5d**). We observed that the IRF/H-GDz/Ca group exhibited the highest ROS content within the tumor. This can be attributed to the inhibition of GLUT-1 and the ROS generation and GSH depletion mediated by IRF-1 (Zhang, Cheng et al. 2022).

Tumor cell nucleus damage is the most indicative parameter of the efficacy of our designed MNF material. We performed continuous slicing on tumors from each group and utilized immunofluorescence staining to label the nuclear damage marker γ -H2AX.....

23. Please specify the sample size and number of biological replicates for reproducibility of results.

We acknowledge the reviewer's diligence in scrutinizing the sample size and biological replicates in our study. To address this concern, we have meticulously outlined the number of experimental repetitions for each figure, elucidated the statistical analysis methods employed, and explicitly stated the availability of data in the source data for every figure. We sincerely appreciate the reviewer's contributions to refining the quality of our work.

Revision made:

Fig. 4. Distribution and therapeutic efficacy of nanomaterials in mice.Body weight changes during the treatment period. All statistics (b, f, g) were calculated using one-way ANOVA using a Tukey post hoc test, and the experiments in (a, b, c) were repeated three times independently with similar results. The source data from (b, e-h) are provided as a Source Data file.

Conclusion

24. Since many effective methods have been reported to stimulate ROS production and an anti-cancer effect, additional discussion is required to clarify the advantages of the MNFs as compared to other nanosystems.

We appreciate the reviewer's suggestion and included an additional discussion section to highlight the advantages of MNF over other nano-systems in stimulating ROS production and exhibiting anti-cancer effects. This indeed will provide a comprehensive understanding of the unique features and benefits offered by MNF in the context of cancer therapy. We once again express our gratitude to the reviewer for providing valuable feedback to enhance the quality of our manuscript.

Revision made:

Discussion

The key advantage of MNFs over MOFs lies in the multifunctionality of the DNA sequences compared to the organic ligands. DNA sequences within MNFs can act as nucleic acid drugs with therapeutic functions, construct adapter sequences for targeted cell binding, and form molecular fluorescence beacons for disease gene diagnosis. Therefore, utilizing MNF materials as carriers for drug delivery not only leverages the functionality of the drug itself but also harnesses the multifunctionality of the DNA sequences simultaneously. However, it is imperative to note that MNFs may suffer from certain limitations, including stability concerns compared to MOFs and challenges associated with achieving rapid synthesis at room temperature.

Our work.....laying the foundation for flexible design of DNA sequences.

The preparation of DNAzyme molecular beacons served three main purposes: confirming their target substrate recognition ability, demonstrating the sustained activity of functionalized DNAzymes in MNF materials post-cellular internalization, and leveraging DNA nanotechnology for multifaceted MNF applications. While GLUT-1 is present in many cell types, our study provides a practical framework for detecting low-abundance genes in future MNF applications.

From a therapeutic perspective, the current scientific landscape is rich with endeavors to enhance ROS production within the tumor microenvironment for robust anticancer effects(Xu, Saw et al. 2017, Zheng, Ding et al. 2021). In alignment with this research enthusiasm, our simplified MNF material presents distinctive advantages for ROS-induced tumor therapy. First, the MNF boasts a

carrier-free therapeutic architecture, integrating therapeutic DNA functional sequences with mitochondrial-disrupting metal ions, thus avoiding the introduction of non-therapeutic components. Second, leveraging the innate targeting proficiency of DNA nanotechnology, MNF requires no exogenous cellular targeting modifications, streamlining the therapeutic approach. Third, protein-loading capability of MNF facilitates the IRF-1 loading and inhibition of nuclear repair proteins RAD51, leading to a significant enhancement of ROS-mediated nuclear DNA fragmentation and thereby amplifying the ROS-driven therapeutic cascade. More importantly, our research heralds a new era for MNF materials, brimming with extraordinary possibilities and unparalleled impact in various research and application domains.

Reviewer #2 - DNazymes, cancer therapy (Remarks to the Author):

This is an interesting manuscript that describes the potential of metal-nucleic acid frameworks (MNFs) to serve as functional nanomaterials. However, MNF production have in the past generally needed harsh generative conditions (heat, time, small oligos). DNazymes are DNA-based cleaving agents that degrade mRNA. Here, the authors use a multi-segmented oligonucleotide (MSO) approach to provide proof-of-principle evidence for the efficacy of an interferon regulatory factor-1 (IRF-1) loaded Ca²⁺/aptamer-DNAzyme MNF to target regulate GLUT-1 mRNA expression in human EGFR-2 (Her-2)-positive gastric cancer cells. The authors provide evidence of GSH depletion and ROS homeostasis, among other measures of efficacy. Tumor growth was inhibited by 90% in the RF/H-GDz/Ca group.

The work may of significance to the field and related fields particularly if there is added evidence for the generality of the observations, but currently this is just focused on a gastric cancer model. The work generally supports the conclusions and claims, but would be strengthened by incorporation of the following additional items:

Dear reviewer, we thank you for your support and for taking the time to provide detailed constructive feedback on our manuscript. We hope the responses detailed below will adequately address the reviewer's points and will satisfy the concerns that were raised.

1. The Abstract should mention efficacy data in tumor bearing mice.

Thank you for the reviewer's valuable suggestion. We included efficacy data in tumor-bearing mice in the Abstract to provide a more comprehensive overview of the experimental outcomes.

Revision made:

Abstract

.....As a proof-of-concept, we constructed an interferon regulatory factor-1 (IRF-1) loaded Ca²⁺/(aptamer-deoxyribozyme) MNF to target regulate glucose transporter (GLUT-1) expression in human epidermal growth factor receptor-2 (HER-2) positive gastric cancer cells. This MNF nanodevice disrupts GSH/ROS homeostasis, suppresses DNA repair, and augments ROS-mediated DNA damage therapy, with tumor inhibition rate up to 90%. Our work signifies a significant advancement towards a new era of universal MNF application.

2. The legends in the Suppl Materials should indicate how many ‘n’ (biologically independent determinations) the plots and images are meant to represent.

We appreciate your attention to this important point. In the legends of the Supplementary Materials, we include information about the number of biologically independent determinations ('n') represented by the plots and images. More importantly, we further add the sample size and number of biological replicates in the main text. This addition aims to provide transparency and clarity regarding the experimental design and data representation.

Revision made:

For the Supplementary Materials

Supplementary Figure 2. The persistence length of short strand GDz, and the persistence length of long strand H-GDz. The source data is available in the provided Source Data file.

Supplementary Figure 6. Densitometric analysis of the bands through ImageJ. (n = 3 independent experiments and the data are presented as mean values \pm SD) All statistics were calculated using one-way ANOVA using a Tukey post hoc test. Source data are provided as a Source Data file.

Supplementary Figure 8. Lysosomal escape experiment of MNF material. The MNF material was synthesized using Cy5.5-labeled H-GDz, allowing the observation of the intracellular localization of MNF material through the Cy5.5 channel (red signal). Lysosomes within the cells were detected using the green channel of LysoTracker™ Green DND-26. The co-localization analysis of the two channels within the cells was performed using the Coloc 2 plugin and line-scan analysis in ImageJ. (n = 3 independent experiments, with similar results. Source data are provided as a Source Data file.)

Supplementary Figure 9. Higher magnification to facilitate a clear observation of intracellular protein expression and fluorescence signals. (n = 3 independent experiments, with similar results.)

Supplementary Figure 10. Quantification of fluorescence signal intensity in confocal microscopy experiments using ImageJ. (A) Quantification of fluorescence intensity emitted in the FAM channel (470/525 nm) after binding of GDz to mRNA. (B) Immunofluorescence experiment targeting GLUT-1, labeled in the TRICT channel (557/576 nm). (n = 3 independent experiments and the data are presented as mean values \pm SD) Statistics were calculated using two-tailed paired t-test for the two interested group. Source data are provided as a Source Data file.

Supplementary Figure 11. The experiments with MNF materials on SNU216 cells. The H-GDz sequence was labeled with Cy5.5 and shown as red signal, while the cell nuclear was labeled by DAPI (Blue color) (a) Confocal experiments to confirm the cellular uptake of each material. (b) Flow cytometry techniques to verify the internalization of nanomaterials by SNU216 cells. (c) Assessment of the regulation of GLUT-1 within cells for each material. (n = 3 independent experiments with similar results)

Supplementary Figure 13. The internalization experiments of MNF materials on different HER-2 expressing cell lines involved confocal microscopy and flow cytometry. Confocal experiments and flow cytometry techniques were employed to validate the uptake of H-GDz/Ca MNF NPs by both human dermal fibroblast cells (NHDF) and gastric cancer N87 cells. The H-GDz sequence was labeled with Cy5.5 and shown as red signal, while the cell nuclear was labeled by DAPI (Blue color). (n = 3 independent experiments with similar results.)

Supplementary Figure 14. Characterization of H-GDz/Ca MNF materials after loading IRF-1 protein. (n = 3 independent experiments with similar results, Source data are provided as a Source Data file.)

Supplementary Figure 15. Regulation of metabolic pathways in N87 cells by different MNF materials. Characterization of GLUT-1 protein expression in N87 cells by different material groups, along with downstream

characterization of PPP pathway and hexose phosphorylation pathways. (n = 3 independent experiments and the data are presented as mean values \pm SD) All statistics were calculated using one-way ANOVA using a Tukey post hoc test. Source data are provided as a Source Data file.

Supplementary Figure 16. Regulation of metabolic pathways in SNU216 cells by different MNF materials. (a) Characterization of intracellular PPP metabolism, NADH, and Cysteine consumption processes, as well as GSH synthesis in SNU216 cells. (b) Analysis of the nuclear damage marker γ -H2AX using flow cytometry. (c) Cytotoxicity experiments of different material groups on SNU216 cells. (n = 3 independent experiments and the data are presented as mean values \pm SD) All statistics were calculated using one-way ANOVA using a Tukey post hoc test. Source data are provided as a Source Data file.

Supplementary Figure 17. Observation of nuclear damage in cells under different treatment conditions using TEM. In the control and H-MGDz/Ca groups, the nuclei appear intact with evenly distributed internal DNA. In the H-GDz/Ca group, there are few black aggregates within the DNA, indicating DNA damage and condensation. In the IRF/H-MGDz/Ca group, nuclear morphology is distorted, and numerous black aggregates are present, indicating nuclear damage. In the IRF/H-GDz/Ca group, severe nuclear condensation is observed, with extensive DNA damage within the nuclei. (n = 3 independent experiments, with similar results.)

Supplementary Figure 18. Cytotoxicity experiments of each MNF material group on N87 cells. (n = 3 independent experiments and the data are presented as mean values \pm SD) Source data are provided as a Source Data file.

Supplementary Figure 19. Pharmacokinetics of IRF/H-GDz/Ca and IRF/MH-GDz/Ca NPs through blood circulation. (n = 3 independent experiments and the data are presented as mean values \pm SD) Source data are provided as a Source Data file.

Supplementary Figure 20. Intravenous administration of MNF materials and their distribution in mice at 96h timepoint. (n = 3 independent experiments with similar results).

Supplementary Figure 21. Imaging of the materials in the tumor site of mice, DAPI staining was employed to label the cell nuclei. (n = 3 independent experiments with similar results).

Supplementary Figure 23. H&E stained tissue sections of heart, liver, spleen, lung and kidney. (n = 3 independent experiments with similar results).

Supplementary Figure 24. The tissue sections from the constructed PDX model were used for HER-2 staining. (n = 3 independent experiments with similar results).

Supplementary Figure 25. Tumor growth and weight for different groups for the PDX model. (n = 3 independent experiments and the data are presented as mean values \pm SD) Source data are provided as a Source Data file.

Supplementary Figure 26. Immunofluorescence detection of P-glycoprotein (P-gp) for the PDX model. (n = 3 independent experiments with similar results.)

Supplementary Figure 27. The blood biochemistry and Hemolysis test. (a) After intravenous injection of different NPs, the blood biochemistry of mice was measured, with untreated mice as the control. (b) Hemolysis test conducted for the IRF/H-GDz/Ca group. ALT: Alanine Aminotransferase; AST: Aspartate Aminotransferase; Cr: Creatinine, TBIL: Total Bilirubin. (n = 3 independent experiments and the data are presented as mean values \pm SD) All statistics were calculated using one-way ANOVA using a Tukey post hoc test. Source data are provided as a Source Data file.

For the main text

Fig. 1. Mechanism of MNF synthesis based on long-chain DNA binding with calcium ions and regulation of MNF properties. Statistics in (j) were calculated using two-tailed paired t tests, and the experiments in (a, h, i, j, k) were repeated three times independently with similar results. The source data from (i, j, k) are provided as a Source Data file.

Fig. 2. Validation of the targeting ability of each material towards NIC-N87 cells and characterization of their detection and regulation capabilities towards GLUT-1. Western blot analysis to demonstrate the silencing efficacy of different groups on GLUT-1 expression. Statistics in (f) were calculated using two-tailed paired t tests, and the experiments in (a, b, c, e, f, g) were repeated three times independently with similar results. The source data from (f, g) are provided as a Source Data file.

Fig. 3 Investigation of the synergistic therapeutic effects of IRF-1 in conjunction with GLUT-1 inhibition materials. (K) Assessment of nuclear DNA damage. The Mean Gray Value (average gray value within the selection) is calculated using ImageJ to determine the average intensity of pixels within the selected area, and it is independent of cell density. The data presented in the figures represent the mean values with standard deviation (mean \pm SD). Statistical comparisons were performed using one-way ANOVA, followed by the Tukey post-hoc test for multiple comparisons. The experiments in (d, e, f, g, i, j, k) were repeated three times independently with similar results. The source data underlying figures a, b, d, e, f, h, i, k is available in the provided Source Data file.

Fig. 4. Distribution and therapeutic efficacy of nanomaterials in mice.Body weight changes during the treatment period. All statistics (b, f, g) were calculated using one-way ANOVA using a Tukey post hoc test, and the experiments in (a, b, c) were repeated three times independently with similar results. The source data from (b, e-h) are provided as a Source Data file.

Figure 5. The protein regulation in the in-situ tumor model and the characterization of MNF materials in the PDX model, along with their therapeutic efficacy. (a) Immunohistochemical characterization of the expression regulation of GLUT-1 and RAD51 in different treatment groups. (b) Expression of GLUT-1 in liver and kidney tissues. (c) Therapeutic efficacy of different treatment groups in the PDX model. (d) Representation of ROS in tumor slice tissues stained with Dihydroethidium (DHE), marked in red. (e) Characterization of overall three-dimensional nuclear damage in tumor slices. (f) Labeling of CAF cells in tumor slices. All results were repeated three times with similar results.

3. What was the scientific rationale for the choice of mutant bases in MH-GDz sequences containing Mutated-Her-2 aptamer?

We thank you for raising this crucial question regarding the mechanism of HER-2 aptamer mutation design. The selection of mutant bases in the Mutated-Her-2 aptamer was based on two key considerations:

1. **Significance of GC Pairing in Aptamer Structure:** From the perspective of DNA base pairing, the binding strength of GC is a critical force maintaining the secondary structure of DNA aptamers. Therefore, altering the order of GC pairs can significantly impact the secondary structure of the aptamer (**Supplementary Figure 6**). We simulated the secondary structures of HER-2 aptamer before and after mutation using the NUPACK software, revealing substantial differences.

Supplementary Figure 7. Simulated the secondary structures of HER-2 aptamer before and after mutation using the NUPACK software.

2. **Consideration of Experimental Design:** Our experiment aimed to investigate the influence of DNA sequence length on the construction ability of MNF. Hence, we aimed to mutate the HER-2 aptamer while keeping the sequence length and base type constant between the mutated and wild-type versions. This approach ensures that, during MNF construction, factors other than DNA sequence length are excluded, allowing us to focus on the specific impact of length on construction ability. Therefore, we chose to modify only the arrangement of GC pairs.

By considering these aspects, we selectively introduced order changes for the GC bases, to better understand the effect of mutation on the performance of the MNF. We appreciate the reviewer's attention and guidance, and we have provided detailed explanations in the manuscript.

Revision made:

Furthermore, we investigated the targeting ability of H-GDz towards HER-2 positive NIC-N87 gastric cancer cells. To comparative evaluate the functionality of the HER-2 aptamer, we designed H-GDz sequences containing the HER-2 aptamer, MH-GDz sequences containing a Mutated-HER-2 aptamer, and GDz sequences without the HER-2 aptamer, for the preparation of MNF materials (**Fig. 2C**). The choice of mutant bases in the Mutated-Her-2 aptamer was driven by two key considerations. Firstly, altering the order

of GC pairs has a profound impact on the secondary structure of the aptamer, and this was confirmed through simulations using NUPACK software (Supplementary Fig.7). Secondly, our experimental goal was to investigate the influence of DNA sequence length on the construction ability of MNF. Hence, we only modified the arrangement of GC pairs, without introducing any new base or new sequence, to ensure a focused exploration of length-related impacts during MNF construction.

4. While the targeting effect of H-GDz was tested in Her-2 positive NIC-N87 gastric cancer cells, similar experiments should be performed in non-gastric cancer cells or Her-2 negative/low gastric cancer cells to demonstrate specificity beyond mutant MNF.

We appreciate the insightful suggestion from the reviewer, and we acknowledge the importance of extending similar experiments to non-gastric cancer cells or HER-2 negative/low gastric cancer cells. In response to your guidance, we conducted a comparative study on the internalization of H-GDz/Ca MNF materials in Normal Human Dermal Fibroblasts (NHDF) cells, chosen as a HER-2 low-expressing cell line, and N87 cells as a HER-2 positive cell line. We utilized two methods, including confocal observation of fluorescence intensity in internalized cells and measurement of internalization via flow cytometry. The detailed experimental results are presented in the supplementary material.

Supplementary Figure 13. The internalization experiments of MNF materials on different HER-2 expressing cell lines involved confocal microscopy and flow cytometry. Confocal experiments and flow cytometry techniques were employed to validate the uptake of H-GDz/Ca MNF NPs by both human dermal

fibroblast cells (NHDF) and gastric cancer N87 cells. The H-GDz sequence was labeled with Cy5.5 and shown as red signal, while the cell nuclear was labeled by DAPI (Blue color). (n = 3 independent experiments with similar results.)

Revision made:

Flow cytometry analysis further confirmed the cell-targeting ability of the HER-2 integrated materials (**Fig. 2f**). Compared with control group, GLUT-1 expression was reduced by approximately 37.5% in the H-GDz + Ca²⁺ group, while significantly reduced by up to 70% in the H-GDz/Ca NPs group. Simultaneously, a noticeable shift of cells from the Q3 region (71.0 %) characterized by high GLUT-1 expression and low GLUT-1 detection towards the Q1 region (41.3 %) was observed. Therefore, these findings provided compelling evidence that the transformation of H-GDz into nanostructured MNF materials through self-mineralization techniques can significantly enhance its intracellular gene monitoring and regulating capabilities. In addition to validating the impact of the aptamer sequence's accuracy on endocytosis, we also examined the internalization of MNF materials by N87 cells versus HER-2-negative normal human dermal fibroblasts (NHDF) (**Supplementary Fig. 13**). The results similarly demonstrated the selectivity of MNF materials for HER-2-positive cells.

5. The MNFs accumulated in the kidney and liver. The mice did not demonstrate overt adverse effects. Did the authors investigate whether there were changes in liver and kidney biomarkers?

We appreciate your keen observation, and indeed, the MNFs material exhibited some accumulation in the kidney and liver during the 48-hour ex vivo organ imaging, raising concerns. The accumulation of nanomaterials in vivo can change over time. To provide a clearer reflection of the in vivo metabolism of our material, we conducted longer-term observations. At 96 hours, the signal in the kidney region had essentially disappeared under the same fluorescence scale. This indicates that the concentration of MNF in the body is extremely low. Therefore, the accumulation of nanomaterials in the liver and kidneys does not persist for an extended period and can be completely metabolized. At the 96-hour time point, there was still some MNF material aggregation in the tumor, showing the tumor targeting accumulation effect of the MNF NPs.

Supplementary Figure 20. Intravenous administration of MNF materials and their distribution in mice at 96h timepoint. (n = 3 independent experiments with similar results).

Furthermore, we assessed the levels of GLUT-1 in the liver and kidneys and found no differences among the different treatment groups. Thus, MNFs materials do not impact the protein expression in the liver and kidney organs.

Figure 5. The protein regulation in the in-situ tumor model and the characterization of MNF materials in the PDX model, along with their therapeutic efficacy.....(b) Expression of GLUT-1 in liver and kidney tissues. (c) Therapeutic efficacy of different treatment groups in the PDX model..... three times with similar results.

Revision made:

.....thoroughly assessed. Firstly, the body weight changes of the mice during the treatment period were closely monitored, and no significant differences were observed between the treatment groups and the control group, indicating the absence of adverse effects on the overall health of the animals (**Fig. 4h**). Additionally, histological analysis of vital organs using H&E staining, and GLUT-1 staining of liver and kidney was conducted to evaluate potential tissue damage or abnormalities (**Supplementary Fig. 23 and Fig. 5b**). Remarkably, the results of H&E staining demonstrated the normal morphology and histology of the organs, and no difference of GLUT-1 expression in both the treatment and control groups, further supporting the safety of the formulations.....

6. Tumor growth was inhibited by 90% in the IRF/H-GDz/Ca group. Were there statistically significant effects observed in the H-GDz/Ca and IRF/H-MGDz/Ca groups?

We thank you for raising this interesting point regarding the significance of differences between the H-GDz/Ca and IRF/H-MGDz/Ca groups. Each group has its unique advantages. The H-GDz/Ca group incorporates functional GDz and calcium ions, enabling ROS amplification and inducing mitochondrial damage. On the other hand, IRF/H-MGDz/Ca contains calcium ions, promoting mitochondrial calcification to elevate ROS levels. However, lacking GDz functionality, it cannot achieve ROS amplification by inhibiting glucose uptake and suppressing GSH levels. Nonetheless, loaded with IRF-1 protein, it inhibits RAD51 expression, enhancing ROS's DNA-cleaving ability. In summary, both groups exhibit merits in their treatment strategies.

To statistically evaluate the differences, we employed one-way ANOVA to calculate P values among the three groups: H-GDz/Ca, IRF/H-MGDz/Ca, and IRF/H-GDz/Ca. The analysis revealed no significant difference between the H-GDz/Ca and IRF/H-MGDz/Ca groups. However, the IRF/H-GDz/Ca group, benefiting from the advantages of both groups, demonstrated a significantly different therapeutic effect compared to the other two groups.

Fig. 4. (f) Violin plots illustrating the average changes in Average Radiance in the tumor site for each group of mice.

7. Student's t test was used for comparisons. The authors should indicate if the data was normal. Why was ANOVA (if data was normal) not used for the multi-group studies?

We appreciate the reviewer's attention to statistical methods. In our study, Student's t-test was employed for comparisons between two groups. However, we acknowledge the importance of addressing the normality of the data and using ANOVA for the multi-group analysis. We conducted normality tests for the datasets, and the results show that our data conform to a normal distribution. Prism GraphPad can only compute the Shapiro-Wilk test when the sample size is less than 5000.

Normality and Lognormality Tests		A	B	C	D	E
		1	2	3	4	5
1	Test for normal distribution	Y	Y	Y	Y	Y
2	Shapiro-Wilk test					
3	W	0.9891	0.8922	0.8210	0.8124	0.8120
4	P value	0.9529	0.3933	0.1456	0.1264	0.1256
5	Passed normality test (alpha=0.05)?	Yes	Yes	Yes	Yes	Yes

For multi-group studies, ANOVA is indeed a suitable method when the data meet the assumption of normality and homogeneity of variances. We include information about the normality tests conducted and whether ANOVA was considered based on the normality of the data in the Materials and Methods section for clarity. We value the reviewer's input in enhancing the statistical reporting in our manuscript.

Revision made:

Statistics analysis

Statistical analyses were conducted using GraphPad Prism 9.0, and the results are presented as mean values \pm standard deviation. The normality of the data was first confirmed by Shapiro-Wilk test through using GraphPad Prism 9.0, when the sample size is less than 5000. Student's unpaired t-test (two-tailed) was employed for comparison of two groups, while one-way ANOVA using a Tukey post hoc test was employed for comparison of \geq three groups. A significance level of $p < 0.05$ was considered statistically significant. Measurements were obtained from distinct samples, and no statistical method was used to predetermine the sample size. Data collection and analysis were not performed blind to the experimental conditions, and no data were excluded from the analyses. It is noteworthy that all results can be replicated from the available source data files.

We also recalculated for the data in Figure 4b, 4f and 4g, as below.

8. There is no Discussion, only a Conclusions section. The Discussion also should refer to limitations using DNAzymes and limitations in this study.

We appreciate your valuable guidance and agree that a dedicated “Discussion” section is essential to provide a more comprehensive interpretation of the results and address the limitations of the study. In the revised manuscript, we incorporate a Discussion section that not only delves into the findings but also explicitly discusses the limitations, particularly those associated with the use of DNAzymes. This addition will enhance the overall content and ensure a more thorough presentation of the study. Thank you again for your valuable feedback.

Revision made:

Discussion

The key advantage of MNFs over MOFs lies in the multifunctionality of the DNA sequences compared to the organic ligands. DNA sequences within MNFs can act as nucleic acid drugs with therapeutic functions, construct adapter sequences for targeted cell binding, and form molecular fluorescence beacons for disease gene diagnosis. Therefore, utilizing MNF materials as carriers for drug delivery not only leverages the functionality of the drug itself but also harnesses the multifunctionality of the DNA sequences simultaneously. However, it is imperative to note that MNFs may suffer from certain limitations, including stability concerns compared to MOFs and challenges associated with achieving rapid synthesis at room temperature.

Our work has revealed a pivotal breakthrough in the synthesis of MNF systems byFurthermore, we have validated for the first time the crucial functional activity of aptamers in constructing MNF materials, laying the foundation for flexible design of DNA sequences.

From the perspective of constructing GLUT-1 molecular beacons, the preparation of DNAzyme molecular beacons served three main purposes: confirming their target substrate recognition ability, demonstrating the sustained activity of functionalized DNAzymes in MNF materials post-cellular internalization, and leveraging DNA nanotechnology for multifaceted MNF applications. While GLUT-1 is present in many cell types, our study provides a practical framework for detecting low-abundance genes in future MNF applications.

From a therapeutic perspective, the current scientific landscape is rich with endeavors to enhance ROS production within the tumor microenvironment for robust anticancer effects(Xu, Saw et al. 2017, Zheng, Ding et al. 2021). In alignment with this research enthusiasm, our simplified MNF material presents distinctive advantages for ROS-induced tumor therapy. First, the MNF boasts a carrier-free therapeutic architecture, integrating therapeutic DNA functional sequences with mitochondrial-disrupting metal ions, thus avoiding the introduction of non-therapeutic components. Second, leveraging the innate targeting proficiency of DNA nanotechnology, MNF requires no exogenous cellular targeting modifications, streamlining the therapeutic approach. Third, protein-loading capability of MNF facilitates the IRF-1 loading and inhibition of nuclear repair proteins RAD51, leading to a significant enhancement of ROS-mediated nuclear DNA fragmentation and thereby amplifying the ROS-driven therapeutic cascade. More importantly, our research heralds a new era for MNF materials, brimming with extraordinary possibilities and unparalleled impact in various research and application domains.

Of note, there still exists significant potential for further exploration in the application of DNAzymes within the scope of this experiment. Several aspects remain unresolved. 1)While DNAzymes are currently predominantly utilized for regulating intracellular mRNA and miRNA, their potential to modulate double-stranded DNA in the nucleus warrants deeper investigation. 2) This study focused on a comprehensive examination of one parameter, DNA length, and its impact on MNF construction. Yet, the influence of other factors, such as the type and proportion of composite metal ions, on the stability of MNF materials remains unknown. Therefore, the future biomedical field offers ample opportunities for continued exploration and refinement of DNAzyme-based MNF materials.

Reviewer #3 DNazymes, orthotopic models (Remarks to the Author):

I find this paper well-researched, simple to follow, and to the point.

There are a few issues to ponder, which I trust will enhance the quality of the paper.

Dear reviewer, we sincerely appreciate your positive feedback on our manuscript. Your encouraging words have significantly boosted our confidence and enthusiasm for this research, and we extend our heartfelt gratitude to you. Your comments have been carefully considered, and we are fully committed to addressing the identified issues to further enhance the quality of our manuscript.

Specifics:

1. Line 45: correct spelling is 'guarantee'

We appreciate the reviewer for carefully reading our paper and identifying the typos. We have thoroughly reviewed the manuscript and made the necessary corrections.

Revision made:

.....Hence, it is crucial to improve the preparation into mild conditions and **guarantee** efficient intracellular delivery of macromolecular drugs using multifunctional DNA sequences to broaden the applications of MNFs.....

2. Line 89: define what 'IRF' is at first mention.

Thank you for pointing out this issue. Indeed, IRF-1 is the abbreviation for Interferon Regulatory Factor-1, while IRF is a further abbreviation indicating the loading of IRF-1 into H-GDz/Ca MNF materials. We have emphasized this clarification in the manuscript, hoping to meet the reviewer's requirements.

Revision made:

To overcome this challenge, we employed a mild condition MNF preparation system to load **interferon regulatory factor-1 (IRF-1)³⁶⁻³⁸ (abbreviated as IRF/H-GDz/Ca)**, to suppress RAD51 expression³² (Scheme 1C). In combination with silencing GLUT-1 to achieve ATP depletion (Scheme 1D).....

.....Meanwhile, from the therapeutic viewpoint, we hypothesize that by effectively cellular delivery of our **IRF-1 loaded H-GDz/Ca MNF nanoplatfrom (IRF/H-GDz/Ca)**, the therapeutic efficacy of mitochondrial calcification strategy could be greatly enhanced through GSH/ROS homeostasis disruption, intracellular ATP depletion, and RAD51 suppression.

3. Scheme 1, part D: correct spelling is 'Mitochondria'. Also, Title should have 'Inhibition' with lower case 'I' ... ie. 'inhibition'.

We thank the reviewer for reading our manuscript so carefully. Correcting these errors can significantly enhance the quality of the article.

Revision made:

4. Line 112: change 'to our astonishment' to 'unexpectedly' as the former is journalistic.

We appreciate the reviewer's solid literary background. Indeed, the expression "to our astonishment" appears somewhat informal. We change this to unexpectedly.

Revision made:

.....However, **unexpectedly**, the 20-base sequence was incapable of binding with calcium ions and forming nanoparticles (NPs) (**Fig. 1a2, Supplementary Fig.1**). These observations provided macroscopic evidence supporting the superior MNF synthesis capability of longer DNA sequences compared to shorter ones.

5. Line 185: correct spelling is 'synthesis'.

We would like to express our gratitude to the reviewer for taking the time to read our manuscript and for addressing our writing issues.

Revision made:

.....(**Fig. 1K**). The results confirmed the presence of the target elements in the nanoparticles, including calcium (Ca), phosphorus (P), nitrogen (N), and oxygen (O). This analysis provided direct evidence for the successful **synthesis** of MNFs.

6. Line 228: correct spelling is 'comparatively'.

We once again express our gratitude to the reviewer for taking the time to delve into our manuscript and for pointing out our writing issues.

Revision made:

.....Furthermore, we investigated the targeting ability of H-GDz towards HER-2 positive NIC-N87 gastric cancer cells. To **comparatively** evaluate the functionality of the HER-2 aptamer, we

designed H-GDz sequences containing the HER-2 aptamer, MH-GDz sequences containing a Mutated-HER-2 aptamer, and GDz sequences without the HER-2 aptamer, for the preparation of MNF materials (**Fig. 2C**).....

7. Line 236: introduce what type of cell line NIC-N87 is, and why it was chosen for this study.

Thank you for providing such valuable feedback, especially regarding the clarification of why we chose to study N87 cells. Clearly articulating this viewpoint contributes to a deeper understanding of the design and functional characteristics of our materials. We have supplemented the explanation in the main text to meet the reviewer's request.

Revision made:

..... The NCI-N87 cell line was derived from the stomach of a male patient with gastric carcinoma in 1976. Due to its high expression of the HER-2 receptor, it has been utilized for subsequent experiments involving HER-2 targeted delivery of MNF. The targeting effect of H-GDz towards NIC-N87 cells was manifested in two aspects:

8. Line 268: correct word is 'validate'.

Thank you once again to the reviewer for pointing out the errors in our manuscript. The corrections made to these issues have significantly improved the quality of the manuscript.

Revision made:

Western blotting was also utilized to further **validate** the expression of GLUT-1 within the cells (**Fig. 2G**). The results showed that the HER-2 modified H-GDz + Ca²⁺ group exhibited a slight decrease in GLUT-1 protein expression.....

9. Fig 3A: Where are the MFI error bars?

We thank you for pointing out the issues relate to the statistical analysis. In accordance with the journal's requirements, we have reanalyzed the data and created revised plots that now include specific data points, exact p-values, and error bars.

Revision made:

Revisions are show in Figure 3a, d, e, f, h, i, k

10. Fig 4b: remove dotted/dashed line around figure 4B. It is distracting, and does not really help with segregating B from A, C or D.

We appreciate and agree with your guidance. We have removed the dashed circle in Figure 4B as suggested.

Revision made:

11. Line 665: correct term is "TRITC".

Revision made:

.....After 10 hours of incubation, the cells were observed using confocal laser scanning microscopy (CLSM). NPs were under FAM channel (470/525 nm) and GLUT-1 were under **TRITC channel** (557/576 nm). Additionally, the cells were detached using trypsin, washed with PBS, and subjected to flow cytometry analysis using a BD LSRFortessa flow cytometer (BD Biosciences).....

12. Line 679: explain why samples were not denatured when doing the GLUT-1 protein electrophoresis.

We appreciate your inquiry regarding whether the sample protein should be boiled. The decision to avoid boiling the sample aligns with Abcam's recommendation, as boiling could lead to irreversible aggregation of the GLUT-1 protein. Additionally, boiling might prevent certain proteins from properly entering the gel or result in the formation of polymers that could be misconstrued as background during analysis. This precaution is taken to ensure the integrity of the protein and accurate interpretation of the experimental results.

Revision made:

.....Protein loading buffer was added to the samples, and they were thoroughly mixed and denatured at 98°C for 10 minutes. GLUT-1 protein was not denatured since Abcam's instructions mention that GLUT-1 protein may aggregate irreversibly after boiling. Avoiding boiling helps prevent GLUT-1 proteins from entering the gel or forming polymers that are often mistaken for background. Our experimental design takes this factor into consideration to ensure the accuracy and reliability of.....

13. Line 690: mention whether in the release study, the sample tubes were mixed constantly or stagnant.

You have raised an important question about the release conditions. Indeed, this is crucial for other researchers when replicating the experiment to ensure the smooth release of the drug. We have added the relevant information.

Revision made:

....., 1 mL of IRF/H-GDz/Ca NPs (containing 2 μ M H-GDz and 2 μ g IRF-1) were placed in 1 mL tubes. The tubes were continuously shaken during the release process at 37 degrees Celsius. Two different pH conditions, pH 7 and pH 6, were tested for drug release. At each time point, the samples were.....

14. Line 787: how was treatment administered in vivo, route?

The reviewer raised a critical question about the administration method. We apologize for our oversight and appreciate the reviewer for bringing it to our attention.

Revision made:

***In Vivo* Anticancer Efficacy**

The mice were categorized into five groups, namely control, H-MGDz/Ca, H-GDz/Ca, IRF/H-MGDz/Ca, and IRF/H-GDz/Ca groups. The therapeutics were administered at a concentration equivalent to GDz: 500 nmol/kg through tail vein injection. After a 21-day treatment period, the tumors connected to the stomach were collected for comparative analysis.

Reviewer #4 - Gastric cancer, targeting GLUT (Remarks to the Author):

In this manuscript, the authors reported an interferon regulatory factor-1(IRF-1) loaded Ca²⁺/(aptamer-deoxyribozyme) MNF to target regulate glucose transporter (GLUT-1) expression in human epidermal growth factor receptor-2 (Her-2) positive gastric cancer cells, which could disrupt GSH/ROS homeostasis, suppress DNA repair, and augment ROS-mediated DNA damage therapy. Although there are interesting parts in the contents of the study, in its current form the data are preliminary, and the conclusions require further validations. The direction of the study has a certain suspicion of artificial selection and lacks objective logic, some details and relevant discussions are missed. Specifically, the authors should address the following points to improve the quality of the paper

Dear reviewer, thank you for your valuable feedback on our manuscript. Your insights are instrumental in guiding us towards improving the manuscript, and we sincerely appreciate your time and effort in reviewing our work. We also value your recognition of the interesting aspects of our study. The series of suggestions and concerns raised by the reviewers are very valuable, and we take them very seriously. We are committed to addressing these points to enhance the quality and integrity of our research. We hope our efforts can meet the expectations of the reviewer.

1. The author verified the binding capacity of DNA sequences of different lengths with calcium ions,

1) since the synthesis conditions of metal-DNA nanocomposites are complex and easily degradable, how to ensure the synthesis efficiency and stability in vitro and in vivo experiments.

Indeed, as pointed out by the reviewer, the traditional fabrication of MNF involves stringent conditions, intricate synthesis, and relatively poor stability. **However, it is precisely this challenge that our study aims to overcome.** In this study, we innovatively explored the impact of DNA sequence length on MNF construction for the first time, breaking away from the conventional methods that focus solely on adjusting the concentration ratio of DNA and metal ions.

By applying Molecular Dynamics simulation techniques, we discovered that long-chain DNA sequences exhibit a higher binding affinity with calcium ions. **This revelation has revolutionary implications for the practical synthesis of MNF.** 1) We can now achieve rapid (5 minutes) synthesis of MNF materials in room temperature aqueous conditions, making the synthesis process remarkably facile. 2) The MNF materials synthesized in our study are more stable and yield higher quantities (Supplementary Table S2) compared to those produced using short-chain DNA. **Therefore, the synthesis of metal-DNA nanocomposites is no longer complex or unstable.**

Supplementary Tables S2. Structural composition of the MNF.

	Input DNA weight (μg)	Incorporated DNA (μg)			DNA utilization efficiency			Incorporated Calcium (μg)			Calcium mass ratio			yield μg
H-GDz (83 bases)	52.39	50.08	51.09	50.81	95.6%	97.5%	96.9%	10.3	10.4	11.1	17.0%	16.9%	17.9%	61.26±0.79
GDz (41 bases)	51.5	48.4	47.9	46.2	94.0%	93.0%	89.7%	6	5.5	5.8	11.0%	10.3%	11.1%	53.26±1.20
Substrate (20 bases)	50.8	1.8	1.1	1.2	3.5%	2.1%	2.3%	0.1	0.05	0.1	5.3%	4.3%	7.6%	1.44±0.39

Furthermore, to ensure the stability of MNF both in vitro and in vivo, we conducted stability tests of MNF materials in Dulbecco's Modified Eagle Medium (DMEM) cell culture medium. Leveraging the efficient binding capability of long-chain DNA with calcium ions, the morphology of H-GDz/Ca MNF materials remained unchanged in DMEM for one week. Additionally, in response to the reviewer's request (Question 6 and Question 8), we supplemented the animal experiments including the characterization of MNF materials blood circulation time (Question 6) and the representation of H-GDz in tumor tissues (Question 8). We confirmed the prolonged circulation capability of MNF materials in the blood, reaching approximately 43 hours (According to the principle that it takes about 5 half-lives for a drug to be eliminated from the body, and the half-life in our study is "8.6 hours", Question 6). Also, we ensure the effective delivery of the H-GDz sequence to the interior of tumor cells (Question 8). Hence, we significantly enhancing MNF's clinical applicability compared to previous studies which completely degraded within 36h, even after surface protection(Liu, Hu et al. 2019).

In summary, with the improvements made to the synthesis conditions of MNF materials, we can now ensure efficient synthesis and stability for both in vitro and in vivo experiments.

Supplementary Figure 4. The TEM image of MNF material after storage in DMEM for one week.

Revision made:

.....indicating a higher proportion of negatively charged DNA in larger particles (**Fig. 1j**). We further verified the stability of the nanomaterials in DMEM culture medium and found that their morphology remained unchanged within DMEM for one week, ensuring the stability of MNF (**Supplementary Fig.4**). In the final step, we conducted Energy-dispersive X-ray spectroscopy (EDS) analysis specifically on the MNFs with a DNA/Ca²⁺ ratio of 1:2 (**Fig. 1k**). The results confirmed the presence of the target elements in the nanoparticles, including calcium (Ca), phosphorus (P), nitrogen (N), and oxygen (O). This analysis provided direct evidence for the successful **synthesis** of MNFs.....

2) how to determine the concentration of the transferred oligonucleotide sequence to be 83 mM when constructing?

We thank reviewer for addressing this important question. The DNA sequences were directly ordered from the company, and their freeze-dried DNA powder can be prepared to any concentration by adding calculated volume of nuclease-free water.

Specifically, functional DNA fragments of three different lengths (20 nucleotides, 41 nucleotides, and 83 nucleotides) were designed. For the DNA strand containing 83 nucleotides, a 1mM DNA solution was prepared, resulting in a “nucleotide concentration” of 83mM. For a DNA sequence containing 20 nucleotides, a 4.15mM solution was prepared to achieve a “nucleotide concentration” of 83mM. For a DNA sequence containing 41 nucleotides, a 2.03mM solution was prepared to obtain a “nucleotide concentration” of 83mM. The “nucleotide concentration” is equal to the DNA sequence concentration multiplied by the number of nucleotides in the DNA sequence.

Revision made:

MNF synthesis with different lengths and mass analysis

DNA sequences are ordered directly from the company, and the freeze-dried powder can be prepared at any concentration by adding the required water. Specifically, functional DNA fragments of varying lengths (20 nucleotides, 41 nucleotides, and 83 nucleotides) were designed. For the 83-nucleotide DNA, a 1mM solution was prepared, resulting in a nucleotide concentration of 83mM. For the 20-nucleotide DNA, a 4.15mM solution was prepared to achieve a nucleotide concentration of 83mM. Similarly, for the 41-nucleotide DNA, a 2.03mM solution was prepared to obtain a nucleotide concentration of 83mM. The nucleotide concentration is calculated by multiplying the DNA sequence concentration with the number of nucleotides in the sequence. This approach ensures the precise determination of the nucleotide's concentration during MNF construction. The DNA fragments (2 μ L, 1 mM) were used for MNF synthesis by incubating them with 1M CaCl₂ in a reaction volume of 200 μ L at room temperature. The total weight of the MNF material was measured after freeze-drying. The presence of calcium (Ca) in the MNF material was detected using a Calcium Assay Kit (Colorimetric) from Abcam (ab102505). The DNA content was determined by subtracting the DNA content in the supernatant from the initial amount of DNA before the reaction. The DNA weight was measured using NanoDrop™ 2000/2000c Spectrophotometers (Thermo Scientific™).

3) During the synthesis process, does the folding of longer oligonucleotide chains and spatial conformation changes affect the affinity between the DNA sequence and calcium ions, Will MNF production also be affected as a result? The authors need to provide additional explanations on these aspects.

We thank you for your attention to whether changes in folding and spatial conformation of longer oligonucleotide chains will affect the affinity between DNA sequences and Ca²⁺, as well as whether changes in binding force will affect the production (yield) of MNF. Your viewpoint is absolutely correct, and it is also the core content of this work. We sincerely apologize for any confusion caused by the lack of emphasis on certain points in our writing.

From the simulation results in in Fig. 1b, the H-GDz strand (long strand) indeed show higher folding and entanglement effect. Also, from Fig.1g, we comparably calculated the binding density value (Radial Distribution Function) of both H-GDz and GDz sequences with Ca²⁺ in a Molecular Simulation environment. The results indeed show that long-chain H-GDz attracts a higher density of calcium ions than short-chain GDz. **Therefore, the reviewer's viewpoint is potentially validated. The folding or conformational changes in long DNA chains indeed impact the increased affinity between DNA and calcium ions.** We hypothesize that the entanglement force

offsets the mutual negative electric repulsion between each DNA strand, accumulating higher negative electric potential energy and thus attracting more calcium ions.

Figure 1. (g) The radial distribution function (RDF) analysis of GDz and H-GDz sequences. The horizontal axis represented the distance between calcium ions and oxygen atoms in the phosphate backbone, while the vertical axis represented the density of calcium ions.

On the other hand, we conducted yield calculations for DNA strands of different lengths. It was observed that long-chain DNA can indeed lead to higher MNF production. We analyzed the components in the MNF product and discovered that long-chain DNA indeed contains a higher calcium ion content, aligning with the results of molecular simulations. **Therefore, the reviewer's point is entirely correct, that long-chain DNA can attract a higher content of calcium ions through folding and winding, leading to a higher yield.** Certainly, this conclusion is also the core innovation of our work. We have not only achieved a significant simplification of MNF material fabrication but also delved into its mechanism using molecular simulation methods.

Supplementary Tables S2. Structural composition of the MNF.

	Input DNA weight (μg)	Incorporated DNA (μg)			DNA utilization efficiency			Incorporated Calcium (μg)			Calcium mass ratio			yield μg
H-GDz (83 bases)	52.39	50.08	51.09	50.81	95.6%	97.5%	96.9%	10.3	10.4	11.1	17.0%	16.9%	17.9%	61.26 \pm 0.79
GDz (41 bases)	51.5	48.4	47.9	46.2	94.0%	93.0%	89.7%	6	5.5	5.8	11.0%	10.3%	11.1%	53.26 \pm 1.20
Substrate (20 bases)	50.8	1.8	1.1	1.2	3.5%	2.1%	2.3%	0.1	0.05	0.1	5.3%	4.3%	7.6%	1.44 \pm 0.39

In order to further proof that longer DNA strand is softer and has strong entangling ability, we introduced another parameter Persistence length (L_p). L_p is calculated from the data between the start time and the end time of the end-to-end distance vs time plot, while end-to-end distance is the distance that points from one end of a DNA to the other end. The smaller the L_p value, the higher the flexibility of the material, and therefore the higher its degree of folding. The results show that the L_p value of H-GDz of long-chain DNA is 5.476, while the L_p value of short-chain DNA is 9.527. Therefore, H-GDz is indeed softer than GDz, and its folding probability is higher.

Revision made:

In main text:

The observed results can be attributed to the inherent characteristics of oligonucleotides, which are known to lack strong base stacking forces and exhibit flexible spatial conformations¹⁸. **Consequently, increasing the length of oligonucleotide sequences (which is different from augmenting the nucleotide concentration),**

holds the potential to enhance the folding and entanglement probability of the sequences during the reaction process^{39, 40}. Hence, we hypothesize the entanglement of longer oligonucleotide chains endowing the intertwined DNA clusters with a higher potential, which in turn enhances the affinity of DNA for calcium ions, thus, obtaining higher MNF production.

To obtain molecular level insight into the aforementioned hypothesis, molecular dynamics (MD) simulations were employed, where two DNA systems of identical nucleotide counts were constructed (Fig. 1b and c). System 1 consisted of ten GDz sequences, while system 2 comprised.....

.....Conversely, in the H-GDz system, the DNA fragments displayed a pronounced intertwining phenomenon, forming a densely packed aggregation by 100 ns (Fig. 1D). This uncovering suggests that longer DNA sequences are more inclined to undergo entanglement and intertwining. To further demonstrate this hypothesis, we introduced another parameter, persistence length (Lp). A smaller Lp value indicates higher flexibility and, consequently, a higher degree of folding. The results indicate that the long-chain DNA, H-GDz, has an Lp value of 5.476, while the short-chain DNA, GDz, has an Lp value of 9.527 (Supplementary Fig.2). Therefore, H-GDz is indeed softer than GDz, with a higher probability of folding.....

In supplementary material:

Molecular dynamics simulations

The persistence length (Lp) is determined using the following formula: $\langle h^2 \rangle = 2L_p L_0 [1 - (\frac{L_p}{L_0}) (1 - \exp(-\frac{L_0}{L_p}))]$. Here, $\langle h^2 \rangle$ represents the mean squared end-to-end distance, L₀ is the extended chain length, and L_p is the persistence length. The end-to-end distance is computed for each selected chain in every trajectory frame, and the values are then averaged over the entire trajectory.

Short strand GDz

Persistence length: 9.527 Å Extended chain length: 273.150 Å
 Plot internal distances for: 205 -bond segments
 End-to-end distance: 67.579 Å Time series standard deviation: 14.628 Å Molecule distribution standard deviation: 21.356 Å

long strand H-GDz

Persistence length: 5.476 Å Extended chain length: 645.210 Å
 Plot internal distances for : 459 -bond segments
 End-to-end distance: 80.332 Å Time series standard deviation: 8.440 Å Molecule distribution standard deviation: 23.533 Å

Supplementary Figure 2. The persistence length of short strand GDz, and the persistence length of long strand H-GDz. The source data is available in the provided Source Data file.

2. When constructing MOFs, why did the authors choose calcium ions, which do not easily form MNF structures with nuclear acids, instead of other metal materials with milder synthesis conditions? The authors need to supplement the binding affinity of other metal ions with longer DNA sequences.

We thank you for raising questions regarding metal ion selection. We would be happy to explain why we chose calcium ions.

From the perspective of therapeutic function:

First, this work studies metal nucleic acid frameworks (MNFs) rather than metal organic frameworks (MOFs). MNFs are self-assembled materials composed of DNA sequences and metal ions that offer unique advantages over MOFs due to the inherent versatility of DNA sequences. DNA sequences within MNFs can serve as nucleic acid drugs, cellular targeting ligand as well as molecular fluorescent beacons.

Secondly, we focus on improving the catalytic activity of DNAzymes *in vivo*. The main reason that currently limits the catalytic efficiency of DNAzymes in clinical practice is that it does not have the required metal cofactor in the body. MNF materials include both DNA sequences and metal ions, so using MNF to construct DNAzyme delivery system has unparalleled advantages. Since there are DNAzymes highly responsive to calcium ions, here calcium ions and Ca^{2+} responsive DNAzymes were selected in this work.

However, for iron ions and copper ions, which are relatively readily for synthesizing MNF materials (reaction still need 90°C for 3h), there are currently no therapeutic DNAzymes that utilize these metal ions as cofactors. Besides, from another therapeutic standpoint, our focus lies on the mitochondrial calcification function mediated by calcium ions and the synergistic treatment potential when combined with GLUT-1 silencing.

From the material design perspective:

This study primarily investigates the impact of DNA sequence length on the construction of MNF materials. Therefore, among numerous metals, calcium ions, due to their relatively weaker binding effect, were selected and can be seen as a more typical representation.

Of course, when considering aspects of material synthesis beyond therapeutic implications, the reviewer's suggestion (to explore other metals) holds great significance. Therefore, we have decided to adopt the reviewer's recommendation, choosing Fe^{2+} for the preparation of GDz and H-GDz (**room temperature 5-minute**), with the following results. Of note, this H-GDz/Fe holds no therapeutic meaning.

Here, **1** represents pure Fe^{2+} at 100 mM, **2** represents Fe^{2+} ions combined with GDz, and **3** represents Fe^{2+} ions combined with H-GDz. The results indicate that H-GDz holds higher yields, with an increased size observed in its TEM morphology. On the other hand, the short-chain GDz exhibits a lower yield, with a smaller and smoother morphology. This result reaffirms our hypothesis that long-chain DNA is more prone to forming MNF materials with metal ions. However, since the preparation of Fe-MNF is not the main focus of this study, we did not further optimize the Fe^{2+} /DNA concentration ratio to control the preparation of better Fe-MNF materials.

Further perspective:

As the reviewer said, other metal ions are important and therefore worthy of further study, including the types of metal ions, the concentration ratios of different metal ions to nucleic acids, and even the ratio between mixed metal ions and other parameters. At the same time, after selecting different metal ions, it is necessary to further design DNA sequences that conform to the functions of the corresponding metal ions. Therefore, the MNF field has broad research directions. We thank the reviewers for raising this idea, and we will continue to delve deeper into these areas in our future work. We have incorporated these considerations into the Introduction and Discussion sections of the manuscript.

Revision made:

Introduction

The integration of metals with DNA structures can lead to the creation of exceedingly powerful materials^{1,2,.....}

In recent years, MNFs have gradually gained momentum in the field of cancer treatment by combining the catalytic functions of metal ions with the therapeutic functions of nucleic acid drugs³⁻⁶. MNFs are composed of metal ions and DNA sequences, which similar to another famous material “Metal-Organic Frameworks (MOFs)” that consist of metal ions and organic ligands. One of the primary advantages of MNFs over MOFs is the multifunctionality of DNA sequences compared to organic ligands. Specifically, DNA sequences in MNFs can serve as therapeutic nucleic acid drugs, facilitate specific cell targeting through adapter sequences, and enable disease gene diagnosis via the construction of molecular fluorescence beacons. Consequently, the utilization of MNF materials for drug delivery not only exploits the inherent properties of the drug but also harnesses the diverse functionalities of the DNA sequences.

..... In this study, we incorporated a Her-2 targeted aptamer⁷⁻⁹ into the Ca^{2+} -dependent GLUT-1 DNAzyme¹⁰, thereby elongating the oligonucleotide length to achieve Ca^{2+} -assisted self-mineralization under room temperature, rendering visualized and silenced treatment of GLUT-1 mRNA (**Scheme 1A**). This approach not only increases the MNFs synthetic capacity but achieves targeted delivery of MNFs to Her-2 overexpressing cells. The choice of calcium ions during MNF material preparation was based on several factors. Firstly, the responsiveness of the DNAzyme to calcium ions enabled the provision of essential metal cofactors for the DNAzyme following material degradation. Secondly, the diverse anti-cancer functions of calcium ions were a focal point of our study, particularly their role in mitochondrial calcification. Thirdly, considering material construction, previous research and our own investigations have demonstrated the challenging nature of achieving robust self-assembly between calcium ions and DNA to form MNF materials. Therefore, our use of calcium ions highlights how elongated DNA sequences can enhance the binding capacity of DNA sequences with metal ions.

Discussion

.....Of note, there still exists significant potential for further exploration in the application of DNAzymes outside the scope of this experiment. Several aspects remain unresolved. 1) While DNAzymes are currently predominantly utilized for regulating intracellular mRNA and miRNA, their potential to modulate double-stranded DNA in the nucleus warrants deeper investigation. 2) This study focused on a comprehensive examination of one parameter, DNA length, and its impact on MNF construction. Yet, the influence of other factors, such as the type and proportion of composite metal ions, on the stability of MNF materials remains unknown. Therefore, the future biomedical field offers ample opportunities for continued exploration and refinement of DNAzyme-based MNF materials.

3.The toxicity of nanocarriers is essential for delivery. The cell viability of IRF/H-GDz/Ca MNF with different dosages should be measured.

We appreciate the reviewer's emphasis on the importance of MNF toxicity. We support the reviewer's perspective and have expanded our study to include cell viability experiments not only with N87 cells but also with SNU216 cells for a more comprehensive analysis of the cytotoxicity of the IRF/H-GDz/Ca MNF nanomaterial. The results are presented in the Supporting Materials as (Supplementary Fig. 18 and Supplementary Fig. 16c).

From the results, after 24 hours of treatment, IRF/H-GDz/Ca MNF material demonstrates stronger cytotoxicity in both N87 and SNU216 cell lines, compared to other materials, including H-MGDz/Ca, H-GDz/Ca, and IRF/H-MGDz/Ca. Hence, owing to the effective GLUT-1 silencing facilitated by GLUT-1 DNAzyme and the enhanced nuclear damage effect mediated by IRF, IRF/H-GDz/Ca MNF exhibits stronger therapeutic potential. This combination of mechanisms suggests promising prospects for IRF/H-GDz/Ca MNF in treatment.

Supplementary Figure 18. Cytotoxicity experiments of each MNF material group on N87 cells. (n = 3 independent experiments and the data are presented as mean values \pm SD) Source data are provided as a Source Data file.

Supplementary Figure 16. (c) Cytotoxicity experiments of different material groups on SNU216 cells. (n = 3 independent experiments and the data are presented as mean values \pm SD) Source data are provided as a Source Data file.

Revision made:

In main text:

.....expression and IRF-1 introduction. The damage to the cell nucleus for different groups was also observed in the bio-TEM images (**Supplementary Fig.6**). Therefore, our rationally designed IRF/H-GDz/Ca MNF demonstrated significant power for ROS augmentation triggered DNA damage.

Finally, we conducted cell toxicity experiments, and the results (**Supplementary Fig.18 and Supplementary Fig. 16c**) indicate that after 24 hours of treatment, IRF/H-GDz/Ca MNF material exhibited stronger cytotoxicity compared to other materials, including H-MGDz/Ca, H-GDz/Ca, and the IRF/H-MGDz/Ca group. Therefore, owing to the effective GLUT-1 silencing facilitated by GLUT-1 DNzyme and the enhanced nuclear damage effect mediated by IRF, IRF/H-GDz/Ca MNF demonstrates heightened potential for therapeutic applications.....

In methods:

Cell Toxicity Assay

The efficacy of MNF materials on gastric cancer cells (N87 and SNU216 cell lines) was determined through WST-1 cell viability assay. N87 cancer cells and SNU216 cells were seeded in a 96-well plate at a density of 3000 cells per well and incubated in cell growth medium overnight at 37°C with 5% CO₂. On the following day, the cell growth medium was replaced with fresh medium containing different concentrations of H-MGDz/Ca, H-GDz/Ca, IRF/H-MGDz/Ca, and IRF/H-GDz/Ca MNF, and cells were cultured for 24 hours. All dilutions for cell viability assessment were prepared in cell growth medium. After incubation, 10µL of WST-1 reagent was added to each well, and the cells were incubated for an additional 2 hours at 37°C with 5% CO₂. Absorbance at 440 nm was measured using a multimode reader (Thermo Scientific Inc., Waltham, MA, USA).

4. GLUT-1, disrupt pentose phosphate pathway (PPP) and the synthesis of NADPH, leading to a decrease in cysteine synthesis and disruption of GSH/ROS homeostasis. The authors need to supplement the changes in metabolic key products and enzymes in five groups including Control, H-MGDz/Ca, H-GDz/Ca, IRF/H-MGDz/Ca, and IRF/H-GDz/Ca in Figure 3.

We thank you for pointing out the deficiencies in our research regarding the metabolic changes mediated by GLUT-1 inhibition and its downstream products. We appreciate the reviewer's suggestion, and we supplemented the information on changes in metabolic key products and enzymes in the five groups, including Control, H-MGDz/Ca, H-GDz/Ca, IRF/H-MGDz/Ca, and IRF/H-GDz/Ca.

First, we reconducted the western blot experiments to confirm our MNF nanoparticles can downregulated the GLUT-1 expression. Then, it was reported that the inhibition of GLUT-1, will decrease levels of the pentose phosphate pathway (PPP) intermediate 6-Phosphogluconate (6PG) and specifically increased NADP⁺/NADPH ratio (Liu, Olszewski et al. 2020). To investigate these alterations, we employed specifically the 6-Phosphogluconate Assay Kit (Colorimetric) (ab211071) and the NADP/NADPH Quantification Kit (Sigma, Catalog Number MAK038), to assess the levels of 6PG and the NADP/NADPH ratio. Subsequently, we conducted additional assessments to examine the impact of GLUT-1 inhibition on downstream Cysteine synthesis, by using the Cysteine Assay Kit (Sigma, Catalog Number MAK255). The results of the above three experiments are described in the main text and are presented in the Supplementary Materials as Supplementary Figure 15, serving as supplemental information for Figure 3.

For the GSH content, we have already measured by using GSH/GSSG Ratio Detection Assay Kit (Fluorometric - Green) (ab138881), and the results showed in Figure 3h. Also, the ROS results was show as Figure 3i.

Revision made:

.....With Bio-TEM, the morphology of mitochondria under different therapeutic groups were further examined (**Fig. 3G**). The results revealed that the mitochondria in the H-MGDz/Ca group appeared like those in the control group, with well-defined cristae and intact structures. However, in the H-GDz/Ca group, the mitochondria showed evident damage, characterized by disrupted membranes and the presence of cavity-like structures. The IRF/H-MGDz/Ca group also exhibited noticeable mitochondrial impairment. Notably, the mitochondria in the IRF/H-GDz/Ca group displayed a significant number of damaged mitochondria with hollow-like cavities, indicating that the combination effects of GLUT-1 inhibition and IRF-1 introduction.

It has been reported that the inhibition of GLUT-1 results in a decrease in the levels of the pentose phosphate pathway (PPP) intermediate, 6-Phosphogluconate (6PG), and a specific increase in the NADP⁺/NADPH ratio (Liu, Olszewski et al. 2020). Furthermore, the NADPH downregulation will also block the cysteine synthesis. To investigate these alterations, we utilized the 6-Phosphogluconate assay kit, NADP/NADPH quantification kit, and the cysteine assay kit to assess the levels of 6PG, NADP/NADPH ratio and cysteine concentration (**Supplementary Fig.15**). From the results, we found significantly decreased of the 6PG concentration, while NADP⁺/NADPH ratio were increased for H-GDz/Ca and IRF/H-GDz/Ca group. Meanwhile, cysteine concentration significantly decreased for H-GDz/Ca and IRF/H-GDz/Ca group, which proof that the GIUT-1 inhibition will block the PPP metabolism and further affect

the cysteine synthesis. This series of metabolic outcomes was also validated in another SNU216 gastric cancer cell line (**Supplementary Fig. 16a**).

The GSH content served as a critical indicator regulated by GLUT-1 inhibition. Hence, the GSH levels (μmol per million cells) were determined (**Fig. 3h and Supplementary Fig. 16a**). The results demonstrated.....

In methods

Metabolic key products characterization

The 6-Phosphogluconate (6-PGA) Assay Kit from Abcam (ab211071) was utilized in the experiment. Following the washing of 3×10^5 cells with PBS, they were resuspended in 100 μL of extraction buffer for on-ice lysis and subsequent centrifugal extraction. After adding the reaction mixture, cells were incubated at 37°C for 1 hour. The optical density (OD) at 450nm was measured using an enzyme reader, and the 6-PGA content was determined through a standard curve. For the Cysteine Assay, the kit from Sigma (Catalog Number MAK255) was employed. After washing 3×10^5 cells with PBS, they were resuspended in 100 μL Pierce™ RIPA Buffer for on-ice lysis and centrifugal extraction. The OD was measured at room temperature with excitation at 365nm and emission at 450nm. The Cysteine content was determined using a standard curve. The NADP/NADPH Quantification Kit from Sigma (Catalog Number MAK038) was used for the NADP/NADPH assay. After washing 3×10^5 cells with PBS, they were resuspended in 800 μL extraction buffer, lysed on ice, and centrifugally extracted. The OD at 450nm for both total NADP and NADPH was measured. Subsequently, the obtained values were applied to their respective standard curves to determine the content of total NADP and NADPH. Finally, the NADP/NADPH ratio was calculated.

In supporting information

Supplementary Figure 15. Regulation of metabolic pathways in N87 cells by different MNF materials. Characterization of GLUT-1 protein expression in N87 cells by different material groups, along with downstream characterization of PPP pathway and hexose phosphorylation pathways. (n = 3 independent experiments and the data are presented as mean values \pm SD) All statistics were calculated using one-way ANOVA using a Tukey post hoc test. Source data are provided as a Source Data file.

5. Multiple clinical drug studies targeting HER2 positive advanced gastric cancer patients have shown certain therapeutic efficacy, while the drug resistance could not be ignored. Whether IRF/H-GDz/Ca MNF targeting HER2 positive gastric cancer cells described in the article can effectively improve drug resistance can be explained by constructing preclinical models such as PDX and organoid models.

We admire the reviewer's in-depth understanding of the drug resistance issues encountered by nanomaterials in clinical research. We strongly support the reviewer's viewpoint; hence, we further constructed a PDX model derived from HER-2 positive gastric cancer patients to investigate the inhibitory effect of our IRF/H-GDz/Ca material on drug resistance in this model.

We carefully reviewed literature related to drug resistance and found that Multidrug resistance proteins (MRPs) belong to the C family of the ATP-binding cassette (ABC) transporter group. Among them, the major ABC transporters involved in the development of multidrug resistance are ABC subfamily B member 1 [(ABCB1/P-glycoprotein (P-gp)).(Deeley and Cole 1997)

Based on our prepared IRF/H-GDz/Ca MNF material, both the calcium ions and IRF have significant inhibitory effects on P-glycoprotein. Specifically, the accumulation of calcium ions to a certain extent leads to mitochondrial calcium damage, inhibiting ATP synthesis and subsequently suppressing P-gp (Liu, Zhu et al. 2020). Additionally, IRF-1 itself possesses an inhibitory effect on P-gp (Yuan, Yin et al. 2019). Therefore, our IRF/H-GDz/Ca MNF material theoretically has a role in inhibiting drug resistance. We appreciate the reviewer for introducing this novel perspective, enriching, and enhancing the functionality of our material.

In our study, Firstly, we confirmed that the tissues from the constructed PDX model exhibited high expression of HER-2 receptors, as shown in the results below.

Supplementary Figure 24. The tissue sections from the constructed PDX model were used for HER-2 staining. (n = 3 independent experiments with similar results).

Then, we divided the samples into 5 groups, including Control, H-MGDz/Ca, H-GDz/Ca, IRF/H-MGDz/Ca, and IRF/H-GDz/Ca. From the results, the H-MGDz/Ca group did not exhibit a strong P-glycoprotein (P-gp) inhibitory effect, possibly due to insufficient calcium ion concentration to damage mitochondria and inhibit P-gp synthesis. This result is consistent with the JC-1 results in Figure 3e. On the other hand, the H-GDz/Ca group, with simultaneous inhibition of GLUT-1 protein and calcium ions, showed a significant enhancement in mitochondrial inhibition, leading to a substantial downregulation of P-gp protein. It is noteworthy that the IRF/H-MGDz/Ca group and H-GDz/Ca group exhibited similar abilities to downregulate P-gp protein, with the final

formulation IRF/H-GDz/Ca showing the strongest P-gp downregulation capacity among the nanomaterials. This suggests that IRF-1 indeed possesses a potent P-gp inhibitory capability and can synergize with GLUT-1 DNAzyme.

Supplementary Figure 26. Immunofluorescence detection of P-glycoprotein (P-gp) for the PDX model. (n = 3 independent experiments with similar results.)

Revision made:

In main text

To further enhance the translational potential of MNF nanomaterials, we established a subcutaneous Patient-Derived Xenograft (PDX) tumor model. The HER-2 positive characteristics of the tumor were first verified (**Supplementary Fig. 24**). Then, tumor inhibition experiments on the previous materials were conducted (**Fig. 5c and Supplementary Fig. 25**), and the results indicated that the final formulation, IRF/H-GDz/Ca, could significantly inhibit tumor growth, and the body weight of mice in all groups did not decrease.....

.....Moreover, it has been reported that the presence of calcium ions and IRF-1 in our IRF/H-GDz/Ca MNF material can inhibit P-glycoprotein (P-gp), thereby addressing drug resistance. Specifically, the accumulation of calcium ions induces mitochondrial damage, leading to reduced ATP synthesis and subsequent P-gp suppression. Additionally, IRF-1 itself has been reported to possess inhibitory effects on P-gp. Therefore, we conducted immunofluorescence staining experiments for P-gp (**Supplementary Fig. 26**) and observed that the H-MGDz/Ca group showed limited P-gp inhibition, possibly due to insufficient calcium ions to damage mitochondria. In contrast, the H-GDz/Ca group significantly enhanced mitochondrial inhibition, leading to substantial P-gp downregulation. Notably, the IRF/H-MGDz/Ca and H-GDz/Ca groups had similar P-gp downregulation abilities, with IRF/H-GDz/Ca showing the strongest effect among all formulations. This highlights IRF-1's potent P-gp inhibitory capability.

In methods

Ethical statement

Our study strictly followed ethical guidelines, with all animal procedures conducted in accordance with the Guidelines for the Care and Use of Laboratory Animals. Approval was obtained from the Institutional Animal Care and Use Committee (IACUC) of () under the reference number (). Approval for human experiment was granted by the Ethics Committee of () (Approval No.:). Written and informed consent was procured from patient before the collection of tissue samples. All experimental procedures adhered to ethical standards and international guidelines, including the "International Ethical Guidelines for Biomedical Research Involving Human subjects" "Declaration of Helsinki".

Patient-derived xenografts

We obtained fresh gastric cancer tumor tissue from patients undergoing clinical surgery. The tumor tissue was placed in a tissue storage solution and transported at 4°C. Tumor modeling was initiated within 2 hours. During modeling, necrotic tumor tissue was first removed, and the tumor was cut into small pieces of 3 mm x 3 mm x 3 mm. The tumor tissue blocks were implanted subcutaneously in the right anterior shoulder of the mice. All PDX experiments were conducted in 6-8-week-old female NOG (NOD.Cg-Prkdc^{scid} IL2rg^{tm1Sug}/JicCr1) mice provided by Beijing Vital River Laboratory Animal Technology Co., Ltd. Tumor growth was observed daily to obtain the first-generation gastric cancer-bearing mice. When the subcutaneous tumors in NOG mice grew to approximately 300 mm³, the skin was incised, and the subcutaneous tumors were dissected. Part of the tumor tissue was used as a source for passaging. This process was repeated until stable third-generation PDX-bearing mice were obtained for subsequent experiments. The gastric cancer tumor tissue was validated for HER-2 positive characteristics through immunohistochemistry. All mice were housed and bred in a specific pathogen-free (SPF) facility with controlled temperature (22°C) and lighting (12:12 hours light-dark cycle). The tumor-bearing mice were randomly assigned to five groups when the tumor volume reached approximately 100 mm³. The experiment was approved by the Ethics Committee of () (Approval No.), and informed, written consent was obtained from patients before collecting tissue. Ethical considerations include defining humane endpoints, which encompass scenarios where the tumor burden exceeds 10% of the normal body weight, weight loss surpasses 20% of the normal weight, or persistent self-harm by the animals is observed. In such cases, euthanasia is carried out through cervical dislocation under deep anesthesia.

6. Since nanoparticle behavior in mice bloodstream is more complex than in vitro, this study raises serious concerns: What is the blood circulation time of MNF in vivo? The authors need to clarify how to determine the observation time point of 48 hours in vivo experiments. Is there clear comparison of the theoretical and experimental data (e.g, curves)?

We appreciate the crucial point raised by the reviewer regarding the blood circulation time of MNF materials *in vivo* and why we chose 48 hours as the observation time point. We are grateful for the opportunity to elaborate on our considerations in this content.

Firstly, as requested by the reviewer, we calculated the blood circulation time of MNF materials *in vivo*. The blood circulation for MNF materials exhibited a typical two-compartment model, characterized by a rapid decline in the distribution phase and a longer elimination phase. The plasma half-life ($T_{1/2}$) represents the time it takes for the plasma concentration to decrease by 50%. Following intravenous administration, the initial rapid decline in serum concentration defining the “distribution half-life” ($T_{1/2, \alpha}$). Subsequently, after the completion of distribution, a relatively slow descent rate in serum concentration occurs, characterized by the “elimination half-life” ($T_{1/2, \beta}$). We calculated the elimination half-lives ($T_{1/2, \beta}$) by fitting the experimental data using GraphPad Prism 10, two phase decay model:

$$\begin{aligned} \text{SpanFast} &= (\text{Y0} - \text{Plateau}) * \text{PercentFast} * .01 \\ \text{SpanSlow} &= (\text{Y0} - \text{Plateau}) * (100 - \text{PercentFast}) * .01 \\ \text{Y} &= \text{Plateau} + \text{SpanFast} * \exp(-\text{KFast} * \text{X}) + \text{SpanSlow} * \exp(-\text{KSlow} * \text{X}) \end{aligned}$$

Plateau is the Y value at infinite times, expressed in the same units as Y.

Kfast and Kslow are the two rate constants, expressed in reciprocal of the X axis time units. If X is in minutes, then K is expressed in inverse minutes.

Half-life (fast) and Half-life (slow) ($T_{1/2, \beta}$) are in the time units of the X axis. They are computed as $\ln(2)/K$.

PercentFast is the fraction of the span (from Y0 to Plateau) accounted for by the faster of the two components.

The results indicate that the elimination half-life of IRF/H-GDz/Ca and IRF/MH-GDz/Ca *in vivo* is approximately 600 minutes. It is noteworthy that 94 to 97% of a drug will be below a clinically relevant plasma concentration after 4 to 5 elimination half-lives, and thus will be considered eliminated (Nnane 2005, Hallare and Gerriets 2023). **Therefore, at approximately 3000 minutes (around 50 hours), the MNF is estimated cleared from the bloodstream. Thus, selecting a time point around 50 hours will be suitable for the observation of stable drug accumulation in each organ without significant non-specific signals from transient drug residues in the body.** In our case, we considered that 48 hours is a commonly used time point in many oligonucleotide drug studies, facilitating comparison and validation with existing literature (Xu, Xie et al. 2013, Jiang, Qiao et al. 2021).

Nonlin fit		Table of results	
		IRF/H-GDz/Ca	IRF/MH-GDz/Ca
1	Two phase decay		
2	Best-fit values		
3	Y0	1.001	0.9779
4	Plateau	0.1002	0.1258
5	PercentFast	55.30	49.19
6	KFast	0.01540	0.01770
7	KSlow	0.001132	0.001162
8	Half Life (Slow)	612.4	596.3

Supplementary Figure 19. Pharmacokinetics of IRF/H-GDz/Ca and IRF/MH-GDz/Ca NPs through blood circulation. (n = 3 independent experiments and the data are presented as mean values \pm SD) Source data are provided as a Source Data file.

From the results within 48-hour observation period, we observed that the material predominantly accumulated in the renal organs. The Aptamer-DNAzyme sequence we designed can be considered as oligonucleotide drug. Literature reports indicate that negative nanomaterials and oligonucleotide drugs are mainly cleared through the kidneys (Xu, Wu et al. 2017, Du, Yu et al. 2018, Jiang, Qiao et al. 2021). Hence, this result reflects that 48-hour time point is a suitable time point, allowing for a better understanding of the predominant clearance of MNF by the kidneys, with minimal signal accumulation observed in other organs.

Figure 4a,b

Furthermore, in order to better address the reviewer's question, we conducted longer-term observations of the MNF material and found that at 96 hours, the signal in the kidney region had essentially disappeared under the same fluorescence scale. This indicates that the concentration of MNF in the body is extremely low. However, dissection of the excised organs revealed that, compared to IRF/MH-GDz/Ca, more of IRF/H-GDz/Ca still accumulated at the tumor region, further demonstrating the MNF's specific aggregation capability at the HER-2 positive tumor site.

Supplementary Figure 20. Intravenous administration of MNF materials and their distribution in mice at 96h timepoint. (n = 3 independent experiments with similar results).

In summary, the circulation time of MNF material *in vivo*, calculated through the two-phase decay formula, is approximately 43 hours, and it is not influenced by the correctness of the Her-2

sequence. Theoretically, adopting an observation time around 50 hours is beneficial for observing the accumulation of MNF material in tissues, without being affected by the transient residue in tissues caused by the high concentration of MNF material in the bloodstream. Since 48 hours is a common time point reported in the literature, we chose this time point. In theory, the longer the time exceeds 50 hours, the more concentrated retention of MNF material *in vivo* will be observed, highlighting its targeting ability in tumors. The results at 96 hours have proofed this theoretical expectation.

Revision made:

In main text:

Encouraged by the robust inhibitory effects of IRF/H-GDz/Ca MNF materials in gastric cancer cell experiments, their tumor-targeting specificity and anti-tumor efficacy were further evaluated *in vivo*. In the initial step, we first tested the blood circulation time of MNF materials *in vivo*. Following intravenous administration, the initial rapid decline in serum concentration is ascribed to drug distribution, defining the distribution half-life ($T_{1/2}$, α). Subsequently, after the completion of distribution, a relatively slow descent rate in serum concentration occurs, characterized by the elimination half-life ($T_{1/2}$, β). We calculated the elimination half-lives ($T_{1/2}$, β) by fitting the experimental data using GraphPad Prism 10, two phase decay model (**Supplementary Fig. 19**). The results indicate that the elimination half-life of IRF/H-GDz/Ca and IRF/MH-GDz/Ca *in vivo* is approximately 600 minutes. It is noteworthy that 94 to 97% of a drug will be eliminated after 4 to 5 elimination half-lives, which means the plasma concentrations of a given drug will be below a clinically relevant concentration and thus will be considered eliminated (Nnane 2005, Hallare and Gerriets 2023). Therefore, at approximately 3000 minutes, which is around 50 hours, the MNF is essentially cleared from the bloodstream. Thus, selecting time points around and above 50 hours will be suitable for the observation of stable drug accumulation in each organ without significant non-specific signals from transient drug residues in the body.

As illustrated in **Fig. 4a and Supplementary Fig. 20**, the *in vivo* distribution experiment of the MNF materials in mice was conducted for 96 hours, and the MNF were labeled with Cy5.5/BHQ-3 probes. Of note, the functionality of Cy5.5/BHQ-3 probes primarily serves to validate the binding between GDz and GLUT-1, rather than achieving specific imaging of the tumor.

The results revealed that MNF NPs constructed with the mutated HER-2 sequence (IRF/MH-GDz/Ca) exhibited substantial kidney accumulation at 48h-time point, which reflected from the lateral and prone positions results. This selective accumulation could be attributed to the inherent renal interception of negatively charged nanomaterials (Choi, Liu et al. 2010, Du, Yu et al. 2018), or negatively charged oligonucleotides (Xu, Wu et al. 2017, Du, Yu et al. 2018, Jiang, Qiao et al. 2021). Conversely, it was evident that IRF/H-GDz/Ca exhibited substantial accumulation in the tumor region within the stomach, and significant fluorescence signals were observed even 96h time point. This result highlighted the remarkable role of the HER-2 aptamer within the MNF material, facilitating targeted delivery of the MNF to the tumor site.

At the designated 48 and 96-hour time point, the mice were euthanized, and *ex-vivo* analysis was conducted to examine the fluorescence distribution of the nanodevice within various organs (**Fig. 4b and Supplementary Fig. 20**). At 48h-time point, consistent with the findings from *in vivo* imaging, the accumulation of MNFs was prominently observed in the kidney, particularly in the IRF/MH-GDz/Ca group, with the kidneys exhibiting the highest accumulation degree. Notably, the accumulation of IRF/H-GDz/Ca NPs in the kidneys was reduced by approximately 50%

compared to the IRF/MH-GDz/Ca group. Additionally, the fluorescence intensity in the tumor region of the IRF/H-GDz/Ca group was found to be 2.15 times higher than that of the IRF/MH-GDz/Ca group. At the 96-hour time point, we observed only fluorescence signals in the tumor, and no fluorescence signals were detectable in the kidneys. This indicates that the material can indeed accumulate at the tumor site and be completely cleared from the body without causing organ burden. Besides, these *ex-vivo* findings further emphasized the enhanced tumor-targeting ability of the nanomaterials achieved through the incorporation of the HER-2 sequence.

In methods part:

In the pharmacokinetics investigation, 6-8-week-old female BALBc-nu mice were randomly allocated into two groups (n=3) and subjected to intravenous administration of distinct formulations: (i) IRF/H-GDz/Ca, (ii) IRF/MH-GDz/Ca. All administered at a dosage of 500 nmol/kg. Blood samples (20 μ L) were retro-orbitally withdrawn at predetermined intervals, and hemostasis was promptly applied to mitigate bleeding. Blood samples, upon withdrawal, were immediately dissolved in 0.1 mL of lysis buffer (1% Triton X-100) through gentle sonification. Cy5.5-siRNAs were subsequently extracted by incubating the blood lysis samples in 0.5 mL of DMSO at room temperature overnight. The Cy5.5 level in the supernatant was quantified by measuring the fluorescence intensity using a microplate reader (Ex = 683 nm, Em = 703 nm).

7. There is no discussion of different cancer lines selection for in vitro and in vivo experiments. The authors need to supplement the relevant experiments of another cell line, such as SNU216, to increase the reliability of the experimental results.

We appreciate the reviewer's guidance and acknowledge the reviewer's viewpoints. To enhance the reliability of our experimental results,

For cell studies:

we have incorporated the supplementary experiments using SNU216 cell line as suggested by the reviewer. Our supplementary experiments involved utilizing confocal microscopy and flow cytometry to validate the effect of accuracy of the HER-2 aptamer sequence on MNF material uptake by the SNU216 cell line. Simultaneously, we assessed the influence of the endocytosed MNFs on GLUT-1 protein expression in cells. Furthermore, in response to the reviewer's suggestion, we characterized the metabolic regulation of 6-GP, NADH+/NADPH, Cysteine, and GSH in the SNU216 cell line following GLUT-1 inhibition. Additionally, using flow cytometry, we detected γ -H2AX in the nucleus, confirming the synergistic effect of GLUT-1 inhibition and IRF-1 delivery on nuclear damage. Finally, WST-1 experiments were conducted to evaluate cell viability under different concentrations of various materials. In the revised manuscript, we have provided a discussion of the SUN216 related results, and we hope these experiments meet the reviewer's expectations.

Supplementary Figure 11. The experiments with MNF materials on SNU216 cells. The H-GDz sequence was labeled with Cy5.5 and shown as red signal, while the cell nuclear was labeled by DAPI (Blue color) (a) Confocal experiments to confirm the cellular uptake of each material. (b) Flow cytometry techniques to verify the internalization of nanomaterials by SNU216 cells. (c) Assessment of the regulation of GLUT-1 within cells for each material. (n = 3 independent experiments with similar results)

Revision made:

In main text:

.....For the purpose of validating the robustness of the aforementioned findings, we conducted a replicate verification utilizing an additional HER-2 positive gastric cancer cell line (SNU216). The results similarly confirm that H-GDz/Ca materials, when possessing the correct HER-2 sequence, exhibit significant internalization in SNU216 cells (**Supplementary Fig. 11a,b**). Furthermore, the internalization of MNF materials is notably higher compared to the individual H-GDz sequence.

Flow cytometry analysis further confirmed the cell-targeting ability of the HER-2 integrated materials (**Fig. 2f and Supplementary Fig. 12**). Compared with control group, GLUT-1 expression was reduced by approximately 37.5% in the H-GDz + Ca²⁺ group, In addition to validating the impact of the aptamer sequence's accuracy on endocytosis, we also examined the internalization of MNF materials by N87 cells versus HER-2-negative normal human dermal fibroblasts (NHDF) (**Supplementary Fig. 13**). The results similarly demonstrated the selectivity of MNF materials for HER-2-positive cells.

Western blotting was also utilized to further validate the expression of GLUT-1 within the N87 and SNU216 cells (**Fig. 2g and Supplementary Fig. 11c**). The results showed that the HER-2 modified H-GDz + Ca²⁺ group exhibited a slight decrease in GLUT-1 protein expression, surpassing the effect observed in the HER-2 mutated MH-GDz + Ca²⁺ group. Of note, the H-GDz/Ca nanomaterial group exhibited the strongest inhibitory effect on GLUT-1, indicating again that the nanomaterials possess higher GLUT-1 expression regulatory capabilities compared to free DNA strands.

Supplementary Figure 16. Regulation of metabolic pathways in SNU216 cells by H-GDz/Ca MNF materials. (a) Characterization of intracellular PPP metabolism, NADH, and Cysteine consumption processes, as well as GSH synthesis in SNU216 cells. (b) Analysis of the nuclear damage marker γ -H2AX using flow cytometry. (c) Cytotoxicity experiments of different material groups on SNU216 cells. (n = 3 independent experiments and the data are presented as mean values \pm SD) All statistics were calculated using one-way ANOVA using a Tukey post hoc test. Source data are provided as a Source Data file.

Revision made:

In main text:

.....combination effects of GLUT-1 inhibition and IRF-1 introduction.

It has been reported that the inhibition of GLUT-1 results in a decrease in the levels of the pentose phosphate pathway (PPP) intermediate, 6-Phosphogluconate (6PG), and a specific increase in the NADP⁺/NADPH ratio (Liu, Olszewski et al. 2020). Furthermore, the NADPH downregulation will also block the cysteine synthesis. To investigate these alterations, we utilized the 6-Phosphogluconate assay kit, NADP/NADPH quantification kit, and the cysteine assay kit to assess the levels of 6PG, NADP/NADPH ratio and cysteine concentration (**Supplementary Fig.15**) in N87 cell line. From the results, we found significantly decreased of the 6PG concentration, while NADP⁺/NADPH ratio were increased for H-GDz/Ca and IRF/H-GDz/Ca group. Meanwhile, cysteine concentration significantly decreased for H-GDz/Ca and IRF/H-GDz/Ca group, which proof that the GLUT-1 inhibition will block the PPP metabolism and further affect the cysteine synthesis. This series of metabolic outcomes was also validated in another SNU216 gastric cancer cell line (**Supplementary Fig. 16a**).

The GSH content served as a critical indicator regulated by GLUT-1 inhibition. Hence, the GSH levels (μmol per million cells) were determined (**Fig. 3h and Supplementary Fig. 16a**). The results demonstrated that both H-GDz NPs and the IRF/H-MGDz/Ca group could effectively.....

.....increased by 110.5%, owing to the synergistic effects of downregulating GLUT-1 expression and IRF-1 introduction. The damage to the cell nucleus for different groups was also observed in the bio-TEM images (**Supplementary Fig.17**). Additionally, within the SNU216 cell line, we observed that IRF/H-GDz/Ca materials efficiently mediated the expression of γ -H2AX (**Supplementary Fig. 16b**). Therefore, our rationally designed IRF/H-GDz/Ca MNF demonstrated significant power for ROS augmentation triggered DNA damage.

Finally, we conducted cell toxicity experiments in both N87 and SNU216 cell line, and the results (**Supplementary Fig.18 and Supplementary Fig. 16c**) indicate that after 24 hours of treatment, IRF/H-GDz/Ca MNF material exhibited stronger cytotoxicity compared to other materials, including H-MGDz/Ca, H-GDz/Ca, and the IRF/H-MGDz/Ca group. Therefore, owing to the effective GLUT-1 silencing facilitated by GLUT-1 DNAzyme and the enhanced nuclear damage effect mediated by IRF, IRF/H-GDz/Ca MNF demonstrates heightened potential for therapeutic applications.

For in vivo studies:

In addition to further cell studies, we also implemented a new in vivo model (PDX model) to enhance the repeatability of our material. In the animal experiments, we explored various markers, including tumor inhibition experiments, observation of mouse body weight, ROS characterization within the tumor, P-gp characterization, and characterization of CAF cells in the tumor microenvironment. Furthermore, we have created a new Figure 5 in the manuscript to illustrate these additional data. We hope that these extended animal experiments will meet the requirements of the reviewers.

Supplementary Figure 24. The tissue sections from the constructed PDX model were used for HER-2 staining. (n = 3 independent experiments with similar results).

Figure 5. The protein regulation in the in-situ tumor model and the characterization of MNF materials in the PDX model, along with their therapeutic efficacy. (c) Therapeutic efficacy of different treatment groups in the PDX model. All results were repeated three times with similar results.

Supplementary Figure 25. Tumor growth and weight for different groups for the PDX model. (n = 3 independent experiments and the data are presented as mean values \pm SD) Source data are provided as a Source Data file.

Supplementary Figure 26. Immunofluorescence detection of P-glycoprotein (P-gp) for the PDX model. (n = 3 independent experiments with similar results.)

Figure 5. The protein regulation in the in-situ tumor model and the characterization of MNF materials in the PDX model, along with their therapeutic efficacy. (a) Immunohistochemical characterization of the expression regulation

of GLUT-1 and RAD51 in different treatment groups. (b) Expression of GLUT-1 in liver and kidney tissues. (c) Therapeutic efficacy of different treatment groups in the PDX model. (d) Representation of ROS in tumor slice tissues stained with Dihydroethidium (DHE), marked in red. (e) Characterization of overall three-dimensional nuclear damage in tumor slices. (f) Labeling of CAF cells in tumor slices. All results were repeated three times with similar results.

Revision made:

In main text:

To further enhance the translational potential of MNF nanomaterials, we established a subcutaneous Patient-Derived Xenograft (PDX) tumor model. The HER-2 positive characteristics of the tumor were first verified (**Supplementary Fig. 24**). Then, tumor inhibition experiments on the previous materials were conducted (**Fig. 5c and Supplementary Fig. 25**), and the results indicated that the final formulation, IRF/H-GDz/Ca, could significantly inhibit tumor growth, and the body weight of mice in all groups did not decrease.

Subsequently, we investigated the intratumoral levels of ROS (**Fig. 5d**). We observed that the IRF/H-GDz/Ca group exhibited the highest ROS content within the tumor. This can be attributed to the inhibition of GLUT-1 and the ROS generation and GSH depletion mediated by IRF-1 (Zhang, Cheng et al. 2022).

Tumor cell nucleus damage is the most indicative parameter of the efficacy of our designed MNF material. We performed continuous slicing on tumors from each group and utilized immunofluorescence staining to label the nuclear damage marker γ -H2AX, as depicted by the yellow signal in **Fig. 5e**. The results indicate that IRF/H-GDz/Ca MNF NPs can mediate comprehensive nuclear damage throughout the three-dimensional tumor, providing strong evidence of the therapeutic effectiveness of the material.

On the other hand, the regulation of the tumor microenvironment is crucial for the treatment of PDX tumor models. Cancer-associated fibroblast (CAF) cells in the tumor were labeled using α -smooth muscle actin (α -SMA) (green signal) and fibroblast activation protein (FAP) (yellow signal). We observed that the inhibition of GLUT-1 downregulates the expression of CAF cells; however, IRF-1 does not inhibit CAF (**Fig. 5f**). This indicates that our designed MNF materials can achieve the inhibition of CAFs by suppressing GLUT-1, thereby enhancing therapeutic efficacy.

Moreover, it has been reported that the presence of calcium ions and IRF-1 in our IRF/H-GDz/Ca MNF material can inhibit P-glycoprotein (P-gp), thereby addressing drug resistance. Specifically, the accumulation of calcium ions induces mitochondrial damage, leading to reduced ATP synthesis and subsequent P-gp suppression. Additionally, IRF-1 itself has been reported to possess inhibitory effects on P-gp. Therefore, we conducted immunofluorescence staining experiments for P-gp (**Supplementary Fig. 26**) and observed that the H-MGDz/Ca group showed limited P-gp inhibition, possibly due to insufficient calcium ions to damage mitochondria. In contrast, the H-GDz/Ca group significantly enhanced mitochondrial inhibition, leading to substantial P-gp downregulation. Notably, the IRF/H-MGDz/Ca and H-GDz/Ca groups had similar P-gp downregulation abilities, with IRF/H-GDz/Ca showing the strongest effect among all formulations. This highlights IRF-1's potent P-gp inhibitory capability.

Finally, we further investigated the safety of the MNF material in vivo. We assessed the blood biochemistry markers for the constructed PDX model, focusing on ALT (Alanine

Aminotransferase), AST (Aspartate Aminotransferase), TBIL (Total Bilirubin), and blood creatinine. Elevated ALT and AST levels indicate liver cell damage, while increased TBIL levels suggest issues with liver function and hemolysis. Blood creatinine elevation signifies impaired kidney function. Our nanomaterials (H-MGDz/Ca, H-GDz/Ca, IRF/H-MGDz/Ca, IRF/H-GDz/Ca) showed no significant differences in these markers compared to the healthy untreated group, indicating no adverse effects on physiological health (**Supplementary Fig. 27a**). In addition, in vitro hemolysis experiments confirmed that our nanomaterials do not cause blood cell rupture (**Supplementary Fig. 27b**).

In summary, the results from the PDX model demonstrate that our MNF system, loaded with IRF-1, significantly increases ROS levels in tumor tissues, enhances cellular nuclear damage, and exhibits no harm to normal tissues.

8. In Figure 4C, the authors detected the fluorescence intensity of nanomaterials in tumor tissue, and the authors need to provide continuous slices that reflect DNA damage indicators. Similarly, in the IRF/H-GDz/Ca group, the tumor was almost completely eradicated at day 21, exhibiting over 90% significant tumor reduction (Figure 4D). This significant treatment outcome can be attributed to the synergistic effect of GLUT-1 inhibition and IRF-1 increase (S7). The authors also need to provide the fluorescence intensity values of nanomaterials in these five groups.

The reviewer raised crucial questions that contribute to elucidating the mechanism of damage within tumors.

Regarding the characterization of tumor damage mediated by MNF materials in the PDX model, we adopted the reviewer's suggestion and conducted continuous slice experiments on tumors from each group. The results indicate that our final formulation group can achieve uniform three-dimensional damage to tumors. The yellow signals of cell nuclear damage (γ -H2AX) are evenly distributed on every tumor slice.

Figure 5. The protein regulation in the in-situ tumor model and the characterization of MNF materials in the PDX model, along with their therapeutic efficacy..... (e) Characterization of overall three-dimensional nuclear damage in tumor slices..... All results were repeated three times with similar results.

Regarding the distribution of nanomaterials in tumor tissue in Figure 4d:

We first demonstrated through two groups (in Figure 4c) that MNF materials containing the HER-2 aptamer sequence exhibit higher tumor targeting capabilities. After confirming the targeting ability of H-GDz, the 5 groups used in Figure 4D all employed HER-2-modified MNF materials, so their distribution capabilities should be consistent with those in Figure 4c. However, as pointed out by the reviewer, due to different MNFs containing distinct DNAzyme sequences and protein drugs, we cannot confirm the presence of abundant nanomaterials in each group. Therefore, we recharacterized the distribution of nanomaterials within the tumor. The characterization results are presented in the **Supplementary Figure 21.** as shown below.

Supplementary Figure 21. Imaging of the materials in the tumor site of mice, DAPI staining was employed to label the cell nuclei. (n = 3 independent experiments with similar results).

Revision made:

In main text:

.....significant tumor reduction, with approximately 60% and 75% decrease in tumor size, respectively (**Fig. 4e**). Furthermore, in the final treatment group, IRF/H-GDz/Ca, the tumor was nearly completely eradicated at day 21, exhibiting a remarkable tumor shrinkage of over 90% (**Fig. 4f**). The tumor tissue slides showed all therapeutic groups contained sufficient amount of MNFs inside the tumor (**Supplementary Fig.8**), and the remarkable therapeutic outcome can be attributed to the synergistic effects of GLUT-1 inhibition and the increase in IRF-1, which leads to downregulation of RAD51 expression (**Fig.5a**).....

.....Subsequently, we investigated the intratumoral levels of ROS (**Fig. 5d**). We observed that the IRF/H-GDz/Ca group exhibited the highest ROS content within the tumor. This can be

attributed to the inhibition of GLUT-1 and the ROS generation and GSH depletion mediated by IRF-1(Zhang, Cheng et al. 2022).

Tumor cell nucleus damage is the most indicative parameter of the efficacy of our designed MNF material. We performed continuous slicing on tumors from each group and utilized immunofluorescence staining to label the nuclear damage marker γ -H2AX, as depicted by the yellow signal in **Fig. 5e**. The results indicate that IRF/H-GDz/Ca MNF NPs can mediate comprehensive nuclear damage throughout the three-dimensional tumor, providing strong evidence of the therapeutic effectiveness of the material.

On the other hand, the regulation of the tumor microenvironment.....

9. In the section of in vivo biosafety study, although the authors investigated the changes of mouse body weight, more biosafety experiments need to be provided, for example, major organ pathological analysis, liver function evaluations, the biosafety of kidney, blood compatibility, etc.

We appreciate the reviewer's concerns regarding the safety of our nanomaterials. As the reviewer rightly pointed out, body weight alone may not provide comprehensive information about material safety. In response to this concern, we conducted tests on common biochemical markers in the blood for the constructed PDX model, aiming to address the reviewer's request.

Specifically, ALT and AST are commonly used biochemical markers to assess liver cell damage. They are primarily present in liver cells, and their release into the bloodstream indicates liver cell damage. ALT is mainly distributed in the cytoplasm of liver cells, while AST is mainly distributed in the mitochondria of liver cells. Therefore, elevated levels of ALT and AST may indicate abnormal liver function, suggesting damage to liver cells. TBIL, or Total Bilirubin, includes the total amount of bilirubin released from red blood cell breakdown through liver and bile metabolism, as well as the portion not processed by the liver. Total bilirubin levels are a common indicator for assessing liver function and conditions like hemolysis. Blood creatinine is an indicator for assessing kidney function. Creatinine is a byproduct of muscle metabolism, primarily excreted by the kidneys. Reduced excretion of creatinine due to impaired kidney function leads to elevated blood creatinine levels, indicating impaired kidney function. Therefore, an increase in blood creatinine levels is generally considered an indicator of kidney impairment.

Through the examination of these fundamental blood biochemistry markers, we evaluated the safety of our nanomaterials. The results indicate that our nanomaterials, including H-MGDz/Ca, H-GDz/Ca, IRF/H-MGDz/Ca, and IRF/H-GDz/Ca, showed no significant differences in blood parameters compared to the healthy untreated group. Thus, the MNF materials designed by us do not impose a burden on physiological health. Additionally, we conducted further hemolysis experiments in vitro, demonstrating that our nanomaterials do not cause rupture of blood cells.

Supplementary Figure 27. The blood biochemistry and Hemolysis test. (a) After intravenous injection of different NPs, the blood biochemistry of mice was measured, with untreated mice as the control. (b) Hemolysis test conducted for the IRF/H-GDz/Ca group. ALT: Alanine Aminotransferase; AST: Aspartate Aminotransferase; Cr: Creatinine, TBIL: Total Bilirubin. (n = 3 independent experiments and the data are presented as mean values \pm SD) All statistics were calculated using one-way ANOVA using a Tukey post hoc test. Source data are provided as a Source Data file.

Furthermore, we have performed pathological section tests on each organ tissue in the previous orthotopic gastric cancer model. We observed the slices both overall and at a local level, and no indicators of tissue damage in the response groups were identified. Therefore, based on the above indicators, we believe that our nanomaterials can meet the safety testing requirements within the body.

Supplementary Figure 21. H&E stained tissue sections of heart, liver, spleen, lung and kidney. (n = 3 independent experiments with similar results).

Revision made:

.....formulations. This highlights IRF-1's potent P-gp inhibitory capability.

Finally, we further investigated the safety of the MNF material in vivo. We assessed the blood biochemistry markers for the constructed PDX model, focusing on ALT (Alanine Aminotransferase), AST (Aspartate Aminotransferase), TBIL (Total Bilirubin), and blood

creatinine. Elevated ALT and AST levels indicate liver cell damage, while increased TBIL levels suggest issues with liver function and hemolysis. Blood creatinine elevation signifies impaired kidney function. Our nanomaterials (H-MGDz/Ca, H-GDz/Ca, IRF/H-MGDz/Ca, IRF/H-GDz/Ca) showed no significant differences in these markers compared to the healthy untreated group, indicating no adverse effects on physiological health (**Supplementary Fig. 27a**). In addition, in vitro hemolysis experiments confirmed that our nanomaterials do not cause blood cell rupture (**Supplementary Fig. 27b**).

In summary, the results from the PDX model demonstrate that our MNF system, loaded with IRF-1, significantly increases ROS levels in tumor tissues, enhances cellular nuclear damage, and exhibits no harm to normal tissues.

References

- Airley, R., A. Evans, A. Mobasher and S. M. Hewitt (2010). "Glucose transporter Glut-1 is detectable in peri-necrotic regions in many human tumor types but not normal tissues: Study using tissue microarrays." *Annals of Anatomy - Anatomischer Anzeiger* **192**(3): 133-138.
- Alakus, H., F. Berlth, E. Bollschweiler, U. Drebber, U. Warnecke-Eberz, R. Metzger, A. H. Hölscher and S. Monig (2010). "S1941 GLUT-1 Expression in Gastric Cancer: A New Independent Prognostic Marker." *Gastroenterology* **138**(5, Supplement 1): S-285.
- Chi, Q., G. Wang and J. Jiang (2013). "The persistence length and length per base of single-stranded DNA obtained from fluorescence correlation spectroscopy measurements using mean field theory." *Physica A: Statistical Mechanics and its Applications* **392**(5): 1072-1079.
- Choi, H. S., W. Liu, F. Liu, K. Nasr, P. Misra, M. G. Bawendi and J. V. Frangioni (2010). "Design considerations for tumour-targeted nanoparticles." *Nature Nanotechnology* **5**(1): 42-47.
- Deeley, R. G. and S. P. Cole (1997). "Function, evolution and structure of multidrug resistance protein (MRP)." *Semin Cancer Biol* **8**(3): 193-204.
- DeRosa, M. C., A. Lin, P. Mallikaratchy, E. M. McConnell, M. McKeague, R. Patel and S. Shigdar (2023). "In vitro selection of aptamers and their applications." *Nature Reviews Methods Primers* **3**(1).
- Du, B., M. Yu and J. Zheng (2018). "Transport and interactions of nanoparticles in the kidneys." *Nature Reviews Materials* **3**(10): 358-374.
- Femino, A. M., F. S. Fay, K. Fogarty and R. H. Singer (1998). "Visualization of Single RNA Transcripts in Situ." *Science* **280**(5363): 585-590.
- Hallare, J. and V. Gerriets (2023). Half Life. *StatPearls*. Treasure Island (FL), StatPearls Publishing Copyright © 2023, StatPearls Publishing LLC.
- Jiang, T., Y. Qiao, W. Ruan, D. Zhang, Q. Yang, G. Wang, Q. Chen, F. Zhu, J. Yin, Y. Zou, R. Qian, M. Zheng and B. Shi (2021). "Cation-Free siRNA Micelles as Effective Drug Delivery Platform and Potent RNAi Nanomedicines for Glioblastoma Therapy." *Advanced Materials* **33**(45): 2104779.
- Joost, H. G. and B. Thorens (2001). "The extended GLUT-family of sugar/polyol transport facilitators: nomenclature, sequence characteristics, and potential function of its novel members." *Molecular Membrane Biology* **18**(4): 247-256.
- Lee, H., D. H. Dam, J. W. Ha, J. Yue and T. W. Odom (2015). "Enhanced Human Epidermal Growth Factor Receptor 2 Degradation in Breast Cancer Cells by Lysosome-Targeting Gold Nanoconstructs." *ACS Nano* **9**(10): 9859-9867.
- Liu, B., F. Hu, J. Zhang, C. Wang and L. Li (2019). "A Biomimetic Coordination Nanoplatform for Controlled Encapsulation and Delivery of Drug–Gene Combinations." *Angewandte Chemie International Edition* **58**(26): 8804-8808.
- Liu, J. and Y. Lu (2007). "A DNAzyme Catalytic Beacon Sensor for Paramagnetic Cu²⁺ Ions in Aqueous Solution with High Sensitivity and Selectivity." *Journal of the American Chemical Society* **129**(32): 9838-9839.
- Liu, J., C. Zhu, L. Xu, D. Wang, W. Liu, K. Zhang, Z. Zhang and J. Shi (2020). "Nanoenabled Intracellular Calcium Bursting for Safe and Efficient Reversal of Drug Resistance in Tumor Cells." *Nano Letters* **20**(11): 8102-8111.
- Liu, X., K. Olszewski, Y. Zhang, E. W. Lim, J. Shi, X. Zhang, J. Zhang, H. Lee, P. Koppula, G. Lei, L. Zhuang, M. J. You, B. Fang, W. Li, C. M. Metallo, M. V. Poyurovsky and B. Gan (2020). "Cystine

transporter regulation of pentose phosphate pathway dependency and disulfide stress exposes a targetable metabolic vulnerability in cancer." Nature Cell Biology **22**(4): 476-486.

Lu, Y. X., Q. N. Wu, D. L. Chen, L. Z. Chen, Z. X. Wang, C. Ren, H. Y. Mo, Y. Chen, H. Sheng, Y. N. Wang, Y. Wang, J. H. Lu, D. S. Wang, Z. L. Zeng, F. Wang, F. H. Wang, Y. H. Li, H. Q. Ju and R. H. Xu (2018). "Pharmacological Ascorbate Suppresses Growth of Gastric Cancer Cells with GLUT1 Overexpression and Enhances the Efficacy of Oxaliplatin Through Redox Modulation." Theranostics **8**(5): 1312-1326.

Ma, W., Y. Zhan, Y. Zhang, X. Shao, X. Xie, C. Mao, W. Cui, Q. Li, J. Shi, J. Li, C. Fan and Y. Lin (2019). "An Intelligent DNA Nanorobot with in Vitro Enhanced Protein Lysosomal Degradation of HER2." Nano Lett **19**(7): 4505-4517.

Maassen, S. J., M. V. de Ruiter, S. Lindhoud and J. Cornelissen (2018). "Oligonucleotide Length-Dependent Formation of Virus-Like Particles." Chemistry **24**(29): 7456-7463.

Mahlknecht, G., R. Maron, M. Mancini, B. Schechter, M. Sela and Y. Yarden (2013). "Aptamer to ErbB-2/HER2 enhances degradation of the target and inhibits tumorigenic growth." Proc Natl Acad Sci U S A **110**(20): 8170-8175.

Nnane, I. P. (2005). PHARMACOKINETICS | Absorption, Distribution, and Elimination. Encyclopedia of Analytical Science (Second Edition). P. Worsfold, A. Townshend and C. Poole. Oxford, Elsevier: 126-133.

Pan, J., T.-G. Cha, F. Li, H. Chen, N. A. Bragg and J. H. Choi (2017). "Visible/near-infrared subdiffraction imaging reveals the stochastic nature of DNA walkers." Science Advances **3**(1): e1601600.

Qian, R.-C., Z.-R. Zhou, Y. Wu, Z. Yang, W. Guo, D.-W. Li and Y. Lu (2022). "Combination Cancer Treatment: Using Engineered DNAzyme Molecular Machines for Dynamic Inter- and Intracellular Regulation." Angewandte Chemie International Edition **61**(49): e202210935.

Roth, E., A. Glick Azaria, O. Girshevitz, A. Bitler and Y. Garini (2018). "Measuring the Conformation and Persistence Length of Single-Stranded DNA Using a DNA Origami Structure." Nano Letters **18**(11): 6703-6709.

Sato, S.-i., M. Watanabe, Y. Katsuda, A. Murata, D. O. Wang and M. Uesugi (2015). "Live-Cell Imaging of Endogenous mRNAs with a Small Molecule." Angewandte Chemie International Edition **54**(6): 1855-1858.

Selnhhin, D., S. M. Sparvath, S. Preus, V. Birkedal and E. S. Andersen (2018). "Multifluorophore DNA Origami Beacon as a Biosensing Platform." ACS Nano **12**(6): 5699-5708.

Sheng, C., J. Zhao, Z. Di, Y. Huang, Y. Zhao and L. Li (2022). "Spatially resolved in vivo imaging of inflammation-associated mRNA via enzymatic fluorescence amplification in a molecular beacon." Nature Biomedical Engineering **6**(9): 1074-1084.

Sun, B., R. Chang, S. Cao, C. Yuan, L. Zhao, H. Yang, J. Li, X. Yan and J. C. M. van Hest (2020). "Acid-Activatable Transmorphic Peptide-Based Nanomaterials for Photodynamic Therapy." Angewandte Chemie International Edition **59**(46): 20582-20588.

Tan, L., J. Yuan, W. Zhu, K. Tao, G. Wang and J. Gao (2020). "Interferon regulatory factor-1 suppresses DNA damage response and reverses chemotherapy resistance by downregulating the expression of RAD51 in gastric cancer." Am J Cancer Res **10**(4): 1255-1270.

Tanaka, N., M. Ishihara, M. S. Lamphier, H. Nozawa, T. Matsuyama, T. W. Mak, S. Aizawa, T. Tokino, M. Oren and T. Taniguchi (1996). "Cooperation of the tumour suppressors IRF-1 and p53 in response to DNA damage." Nature **382**(6594): 816-818.

Trcek, T., T. Lionnet, H. Shroff and R. Lehmann (2017). "mRNA quantification using single-molecule FISH in *Drosophila* embryos." Nature Protocols **12**(7): 1326-1348.

Tyagi, N., Y. H. Wijesundara, J. J. Gassensmith and A. Popat (2023). "Clinical translation of metal-organic frameworks." Nature Reviews Materials.

Wang, D., H. Yi, S. Geng, C. Jiang, J. Liu, J. Duan, Z. Zhang, J. Shi, H. Song, Z. Guo and K. Zhang (2023). "Photoactivated DNA Nanodrugs Damage Mitochondria to Improve Gene Therapy for Reversing Chemoresistance." ACS Nano **17**(17): 16923-16934.

Wang, R., W. He, X. Yi, Z. Wu, X. Chu and J.-H. Jiang (2023). "Site-Specific Bioorthogonal Activation of DNAzymes for On-Demand Gene Therapy." Journal of the American Chemical Society **145**(32): 17926-17935.

Wang, Z., Y. Luo, X. Xie, X. Hu, H. Song, Y. Zhao, J. Shi, L. Wang, G. Glinsky, N. Chen, R. Lal and C. Fan (2018). "In Situ Spatial Complementation of Aptamer-Mediated Recognition Enables Live-Cell Imaging of Native RNA Transcripts in Real Time." Angewandte Chemie International Edition **57**(4): 972-976.

Wu, S., K. Zhang, Y. Liang, Y. Wei, J. An, Y. Wang, J. Yang, H. Zhang, Z. Zhang, J. Liu and J. Shi (2022). "Nano-enabled Tumor Systematic Energy Exhaustion via Zinc (II) Interference Mediated Glycolysis Inhibition and Specific GLUT1 Depletion." Advanced Science **9**(7): 2103534.

Xu, J., Y. Liu, Y. Li, H. Wang, S. Stewart, K. Van der Jeught, P. Agarwal, Y. Zhang, S. Liu, G. Zhao, J. Wan, X. Lu and X. He (2019). "Precise targeting of POLR2A as a therapeutic strategy for human triple negative breast cancer." Nature Nanotechnology **14**(4): 388-397.

Xu, X., P. E. Saw, W. Tao, Y. Li, X. Ji, S. Bhasin, Y. Liu, D. Ayyash, J. Rasmussen, M. Huo, J. Shi and O. C. Farokhzad (2017). "ROS-Responsive Polyprodrug Nanoparticles for Triggered Drug Delivery and Effective Cancer Therapy." Advanced Materials **29**(33): 1700141.

Xu, X., J. Wu, Y. Liu, P. E. Saw, W. Tao, M. Yu, H. Zope, M. Si, A. Victorious, J. Rasmussen, D. Ayyash, O. C. Farokhzad and J. Shi (2017). "Multifunctional Envelope-Type siRNA Delivery Nanoparticle Platform for Prostate Cancer Therapy." ACS Nano **11**(3): 2618-2627.

Xu, X., K. Xie, X. Q. Zhang, E. M. Pridgen, G. Y. Park, D. S. Cui, J. Shi, J. Wu, P. W. Kantoff, S. J. Lippard, R. Langer, G. C. Walker and O. C. Farokhzad (2013). "Enhancing tumor cell response to chemotherapy through nanoparticle-mediated codelivery of siRNA and cisplatin prodrug." Proc Natl Acad Sci U S A **110**(46): 18638-18643.

Yan, J., X. Ma, D. Liang, M. Ran, D. Zheng, X. Chen, S. Zhou, W. Sun, X. Shen and H. Zhang (2023). "An autocatalytic multicomponent DNAzyme nanomachine for tumor-specific photothermal therapy sensitization in pancreatic cancer." Nature Communications **14**(1): 6905.

Yang, Y., R. Zhang and C. H. Fan (2020). "Shaping Functional Materials with DNA Frameworks." Trends in Chemistry **2**(2): 137-147.

Yuan, J. S., Z. J. Yin, L. L. Tan, W. Z. Zhu, K. X. Tao, G. B. Wang, W. J. Shi and J. B. Gao (2019). "Interferon regulatory factor-1 reverses chemoresistance by downregulating the expression of P-glycoprotein in gastric cancer." Cancer Letters **457**: 28-39.

Zhang, L., T. Cheng, H. Yang, J. Chen, X. Wen, Z. Jiang, H. Yi and Y. Luo (2022). "Interferon regulatory factor-1 regulates cisplatin-induced apoptosis and autophagy in A549 lung cancer cells." Medical Oncology **39**(4): 38.

Zhao, Y., R. Li, J. Sun, Z. Zou, F. Wang and X. Liu (2022). "Multifunctional DNAzyme-Anchored Metal-Organic Framework for Efficient Suppression of Tumor Metastasis." ACS Nano **16**(4): 5404-5417.

Zheng, P., B. Ding, R. Shi, Z. Jiang, W. Xu, G. Li, J. Ding and X. Chen (2021). "A Multichannel Ca²⁺ Nanomodulator for Multilevel Mitochondrial Destruction-Mediated Cancer Therapy." Advanced Materials **33**(15): 2007426.

Zhi, S., X. Zhang, J. Zhang, X.-y. Wang and S. Bi (2023). "Functional Nucleic Acids-Engineered Bio-Barcode Nanoplatfoms for Targeted Synergistic Therapy of Multidrug-Resistant Cancer." ACS Nano **17**(14): 13533-13544.

REVIEWERS' COMMENTS

Reviewer #1 (Remarks to the Author):

The authors have incorporated most of the reviewers' suggestions that has improved the quality of this paper, there are still some deficiencies that should be addressed before this manuscript is published. The authors have included a lot of information to support their hypothesis, but logic and flow is not properly structured. The paper needs additional help in scientific editing and proofreading. In addition, the paper would be strengthened by incorporating of following points;

The abstract should be restructured to make it concise and impactful and include all the key results. Please carefully go through the manuscript and SI to correct any typo and be consistent with the use of abbreviations and capitalization (ex. Pg. 12, Line 4-7/ Pg. 22, Line 11, "GLUT", Pg. 23 Line18 "IRF/H-GDx/Ca group")

In Supplementary Fig.4, TEM image shows the different size distribution of MNF after incubating with DMEM media for a week. This result does not align with the author's claim, "ensuring the stability of MNF in DMEM culture medium".

Fig 1 j – Please use ANOVA for the statistical analysis method with more than two experimental groups.

Please incorporate the information about MH-GDx/Ca MNF in Figure 1/Table S2 to support your claim regarding the impact of oligonucleotide length (not secondary structure or folding) on MNF formation.

Please reference the supplementary figures appropriately. Supplementary figures 21 and 22 and not referenced in the text. Please include the result of Supplementary Fig. 13 (Pg. 16, end of second paragraph) in the cellular uptake results (Pg. 16, first paragraph).

The data associated with Supplementary Fig. 18 and Supplementary Fig. 16c is key results to

assessing the cytotoxic properties of MNFs against the HER-2 expressing cancer cells. Please consider incorporating them in Fig 3.

In the results section, authors mostly mentioned the significant decrease and increase but authors often missed the group to make this comparison.

The discussion section needs to be expanded to explain the main results and how they form the final take home messages, including the comparison with other established literature. Consider moving some discussion contents from result to discussion part (ex. Pg 8, Line 10-14, Pg. 27, Line 15-17).

Reviewer #2 (Remarks to the Author):

The authors have satisfactorily responded to my concerns. The revised manuscript has been strengthened and should appeal to a broader audience as well.

Reviewer #4 (Remarks to the Author):

In this study, the authors reported an interferon regulatory factor-1 (IRF-1) loaded Ca²⁺/(aptamer-deoxyribozyme) MNF to target regulate glucose transporter (GLUT-1) expression in human epidermal growth factor receptor-2 (Her-2) positive gastric cancer cells, which could disrupt GSH/ROS homeostasis, suppress DNA repair, and augment ROS-mediated DNA damage therapy. The authors' work is quite good, but there are still some issues that need further explanation.

1. It is recommended to supplement the information by indicating the age of the mice at the time of euthanasia in Fig.4g and Supplementary Fig22.

2. It is suggested to specify the name of the cell line used in Fig4d-i.

3. It is recommended to supplement the effects of using the IFR-1 group alone on gastric cancer growth and GSH/ROS homeostasis to illustrate the superiority of the IRF/H-GDz/Ca⁺ nanodevice.

4. The Western blot bands in Supplementary Fig11c are obscured; Please make a revision.
5. The text is in 4d obscured; Please make a revision.

Point-by-point responses to reviewers

Dear reviewers, we sincerely appreciate the opportunity to revise our manuscript in light of your insightful comments. In the attached point-by-point responses file, you will find **your comments are highlighted in bold black text**, and **our responses are provided in blue**. **We have made revisions to the main text and Supporting Information, highlighted in yellow**, and modifications to the methods section have been incorporated to align with the updated figures.

REVIEWERS' COMMENTS

Reviewer #1 (Remarks to the Author):

The authors have incorporated most of the reviewers' suggestions that has improved the quality of this paper, there are still some deficiencies that should be addressed before this manuscript is published. The authors have included a lot of information to support their hypothesis, but logic and flow is not properly structured. The paper needs additional help in scientific editing and proofreading. In addition, the paper would be strengthened by incorporating of following points;

Acknowledging the feedback received, we're grateful for the recognition of the effort made in integrating most of the reviewers' suggestions, which have indeed enhanced the quality of our manuscript. We understand there remain areas requiring further refinement, particularly regarding the structure and logical flow of the document, alongside the need for comprehensive scientific editing and proofreading. We appreciate the constructive critique highlighting these aspects and agree that addressing them is crucial for the clarity and impact of our paper.

To resolve these issues, we committed to conducting a thorough revision. This will include a detailed review of the manuscript's organization to ensure that information is presented in a logical, coherent manner that systematically supports our hypothesis. Furthermore, we seek additional assistance for scientific editing and proofreading from our English-speaking co-authors. We understand the critical nature of these elements in conveying our research effectively.

Regarding the specific points suggested for incorporation, we plan to meticulously integrate these into our manuscript. Each recommended area will be addressed directly, ensuring that our claim is not only strengthened but also more comprehensive and robust. In summary, we are dedicated to addressing the identified deficiencies through careful revision, structured enhancement of logic and flow, and comprehensive editing for scientific accuracy and readability. We are confident these efforts will result in a significantly improved manuscript that meets the esteemed standards.

1. The abstract should be restructured to make it concise and impactful and include all the key results. Please carefully go through the manuscript and SI to correct any typo and be consistent with the use of abbreviations and capitalization (ex. Pg. 12, Line 4-7/ Pg. 22, Line 11, "GLUT", Pg. 23 Line18 "IRF/H-GDx/Ca group")

Thank you for your valuable feedback. We restructured the abstract to ensure conciseness and impact while incorporating all key results. Additionally, we meticulously reviewed the manuscript and supporting information to correct any typos and maintain consistency in the use of abbreviations and capitalization, as per your instructions.

Revision made:

For abstract:

..... Herein, we discover that longer oligonucleotides can enhance the synthesis efficiency and stability of MNFs by increasing DNA folding and entanglement probabilities during the reaction. Besides, longer oligonucleotides provide upgraded metal ions binding conditions, facilitating MNFs to load macromolecular protein drugs in room temperature. Furthermore, longer oligonucleotides facilitate functional expansion of nucleotide sequences, enabling disease-targeted MNFs. As a proof-of-concept, we build an interferon regulatory factor-1 (IRF-1) loaded Ca^{2+} /(aptamer-deoxyribozyme) MNF to target regulate glucose transporter (GLUT-1) expression in human epidermal growth factor receptor-2 (HER-2) positive gastric cancer cells. This MNF nanodevice disrupts GSH/ROS homeostasis, suppresses DNA repair, and augments ROS-mediated DNA damage therapy, with tumor inhibition rate up to 90%. Our work signifies a significant advancement towards an era of universal MNF application.

Through the main text:

.....In addition to specific high glucose uptake tissues, such as astrocytes, high levels of GLUT-1 expression have been an indicator of carcinogenesis (GLUT-1 can be detected in the necrotic areas of many human tumor types, but not in normal tissues)^{42, 43}. Hence, here we validated the detection capability of the H-GDz molecular beacon for GLUT-1 mRNA (Fig. 3a).....

.....Meanwhile, cysteine concentration significantly decreased for H-GDz/Ca and IRF/H-GDz/Ca group, which proof that the GLUT-1 inhibition will block the PPP metabolism and further affect the cysteine synthesis. This series of metabolic outcomes was also validated in another SNU216 gastric cancer cell line (Supplementary Fig. 16a).....

.....Of note, we found a remarkable suppression of RAD51 expression for IRF/H-GDz/Ca group, which could be attributed to the synergistic effect of GLUT-1 downregulation and the inhibitory action of IRF-1 on RAD51.....

To obtain molecular level insight into the aforementioned hypothesis, molecular dynamics (MD) simulations were employed, where two DNA systems of identical nucleotide counts were constructed (**Fig. 2b and c**)..... To further demonstrate this hypothesis, we calculated another parameter, persistence length (L_p), which is used to measure the mechanical bending stiffness of polymeric chains. A smaller L_p value indicates higher flexibility and, consequently, a higher degree of folding. The results indicate that the long-chain DNA, H-GDz, has an L_p value of 5.476 Å, while the short-chain DNA, GDz, has an L_p value of 9.527 Å (**Supplementary Fig.2**). Therefore, H-GDz is indeed softer than GDz, with a higher probability of folding. Moreover, in both systems, the calcium ions were observed to exhibit a notable affinity towards the DNA sequences, forming strong binding interactions, while the sodium and chloride ions inside medium did not exhibit a discernible binding tendency with the DNA sequences (**Fig. 2b,c and Supplementary Fig.3**).

It has been reported that the inhibition of GLUT-1 results in a decrease in the levels of the pentose phosphate pathway (PPP) intermediate, 6-phosphogluconate (6PG), and a specific increase in the $\text{NADP}^+/\text{NADPH}$ ratio. Furthermore, the NADPH downregulation will also block the cysteine synthesis. To investigate these alterations, we utilized the 6-phosphogluconate assay kit, $\text{NADP}^+/\text{NADPH}$ quantification kit, and the cysteine assay kit to assess the levels of 6PG, $\text{NADP}^+/\text{NADPH}$ ratio and cysteine concentration (**Supplementary Fig.15**) in N87 cell line. From the results, we found a significantly decreased of the 6PG concentration, while the $\text{NADP}^+/\text{NADPH}$ ratio was increased for the H-GDz/Ca and IRF/H-GDz/Ca groups. Meanwhile,

cysteine concentration decreased significantly for H-GDz/Ca and IRF/H-GDz/Ca groups, which proves that the GLUT-1 inhibition will block the PPP metabolism and further affects the cysteine synthesis. This series of metabolic outcomes was also validated in another SNU216 gastric cancer cell line (**Supplementary Fig. 16a**).

From the perspective of constructing GLUT-1 molecular beacons, the preparation of DNAzyme molecular beacons served three main purposes: confirming their target substrate recognition ability, demonstrating the sustained activity of functionalized DNAzymes in MNF materials' post-cellular internalization, and leveraging DNA nanotechnology for multifaceted MNF applications. While GLUT-1 is present in many cell types, our study provides a practical framework for detecting low-abundance genes in future MNF applications.

.....Both systems contained 0.15 M calcium chloride solution and roughly an identical count of nucleotides in explicit single-point-charge (SPC) water. System 1 consisted of ten GDz sequences ($10 \times 41 \text{ nt} = 410 \text{ nt}$) while system 2 comprised five H-GDz sequences ($5 \times 83 \text{ nt} = 415 \text{ nt}$) (**Figure 1B and C**). Moreover, the simulation systems were neutralized by adding an appropriate number of sodium ions. Periodic boundary conditions (PBC) were applied with an orthorhombic unit cell. As a reference, system 2 was simulated also with 0.15 M NaCl instead of CaCl₂.

2. In Supplementary Fig.4, TEM image shows the different size distribution of MNF after incubating with DMEM media for a week. This result does not align with the author's claim, "ensuring the stability of MNF in DMEM culture medium".

We appreciate the meticulous observations made by the reviewers. Regarding the stability tests, we conducted multiple trials to ensure the stability of MNF in DMEM culture medium. Regarding the uneven particle size observed in the images, we believe it is primarily due to two reasons.

Firstly, different selected-area may result in variations in particle size. Although many "small" and normal size particles were indeed present in the selected areas, likely due to a reaction time of only 5 minutes in the aqueous phase, leading to the formation of small nanoparticles. However, this observation is consistent with the initial TEM results on the 1st Day and corroborated by dynamic light scattering (DLS) measurements.

Secondly, the MNF material did not exhibit aggregation and remained uniformly distributed, with no increase in particle size. Thus, we consider the MNF material stable in DMEM culture medium for up to one week. Meanwhile, given our multiple verification experiments, we have replaced the previously selected views with alternative views from the same experimental group to address the reviewers' concerns.

Supplementary Figure 4. The TEM image of MNF material after storage in DMEM for one week.

3. Fig 1 j – Please use ANOVA for the statistical analysis method with more than two experimental groups.

We appreciate the suggestion regarding the statistical analysis method for Fig 1j. We utilized ANOVA for the comparison of data among more than two experimental groups, as recommended. This approach ensured robust statistical analysis and enhance the reliability of our findings. Thank you for your valuable input.

Revision made:

Fig. 2. The bold dashed line in the middle is the median, and the unbold line is the quartiles. (k) Energy-dispersive X-ray spectroscopy (EDS) and elemental mapping analysis of MNF nanoparticles. Statistics in (j) were calculated using one-way ANOVA, followed by the Tukey post-hoc test for multiple comparisons, and the experiments in (a, h, i, j, k) were repeated three times independently with similar results. The source data from (i, j, k) are provided as a Source Data file.....

4. Please incorporate the information about MH-GDx/Ca MNF in Figure 1/Table S2 to support your claim regarding the impact of oligonucleotide length (not secondary structure or folding) on MNF formation.

Thank you for your valuable suggestion. We have incorporated information about the MH-GDz/Ca MNF in Figure 3C to strengthen our claim regarding the impact of oligonucleotide length on MNF formation, specifically focusing on length rather than secondary structure or folding.

The yield of MH-GDz/Ca MNF material remains consistent with that of H-GDz/Ca MNF, indicating that under constant concentration ratios, only length influences the binding between DNA and calcium ions. This observation strengthens our assertion that oligonucleotide length is the primary determinant affecting MNF formation, independent of secondary structure or folding. This addition will provide comprehensive support for our assertion and enhance the clarity and credibility of our findings.

5. Please reference the supplementary figures appropriately. Supplementary figures 21 and 22 and not referenced in the text. Please include the result of Supplementary Fig. 13 (Pg. 16, end of second paragraph) in the cellular uptake results (Pg. 16, first paragraph).

Thank you for bringing this to our attention. We ensured that supplementary figures 21 and 22 are appropriately referenced in the text.

Additionally, it's indeed we should combine all the confocal results together for better illustration the relationships for the HER-2 aptamer to different cell lines. We include the results of Supplementary Fig. 13 at the end of the second paragraph on page 16, integrating it into the cellular uptake results described in the first paragraph on the same page. This adjustment provided a more comprehensive and cohesive presentation of our findings, and we thank the reviewer again.

Revision made:

Furthermore, tumor sections were prepared to examine the Cy5.5 fluorescence intensity of the nanomaterials (in cyan) within the tumor tissue (Fig. 5c and **Supplementary Fig. 21**). The results

suggested that the tumor tissue in the IRF/H-GDz/Ca NPs group exhibited a significant fluorescence signal from the nanomaterials.....

.....Later, eGFP was utilized for ex-vivo visualization of the gastric tumor (**Fig. 5g and Supplementary Fig. 22**).....

For the purpose of validating the robustness of the aforementioned findings, we conducted a replicate verification utilizing an additional HER-2 positive gastric cancer cell line SNU216. The results similarly confirm that H-GDz/Ca materials, when possessing the correct HER-2 sequence, exhibit significant internalization in SNU216 cells (**Supplementary Fig. 11a,b**). Furthermore, the internalization of MNF materials is notably higher compared to the individual H-GDz sequence. In addition to validating the impact of the aptamer sequence's accuracy on endocytosis, we also examined the internalization of MNF materials by N87 cells versus HER-2-negative normal human dermal fibroblasts (NHDF) (**Supplementary Fig. 12**). The results similarly demonstrated the selectivity of MNF materials for HER-2-positive cell lines.

6. The data associated with Supplementary Fig. 18 and Supplementary Fig. 16c is key results to assessing the cytotoxic properties of MNFs against the HER-2 expressing cancer cells. Please consider incorporating them in Fig 3.

We appreciate the reviewer's suggestions, as these two results indeed provide a clear visualization of the tumor-killing effect of MNF nanomaterials. Therefore, we have revised Figure 3 accordingly, incorporating Supplementary Fig. 18 and Supplementary Fig. 16c into Figure 3 as Figure 3l and Figure 3m, respectively. Additionally, to align with the modifications made to the figures, we have adjusted the order of supplementary material images to match the sequence in the main text. Thank you for your guidance in improving the clarity and completeness of our presentation.

(l, m) Cytotoxicity experiments of each MNF material group on N87 and SNU216 cells. The data presented in the figures represent the mean values with standard deviation (mean \pm SD).

7. In the results section, authors mostly mentioned the significant decrease and increase but authors often missed the group to make this comparison.

Thank you for highlighting this important aspect. We recognize the need to consistently specify the compared groups when discussing significant changes in our results section. We apologize for any confusion caused by this oversight. In the revised manuscript, we ensured to explicitly mention the groups involved in all significant changes to enhance clarity and interpretation.

In light of the reviewer's suggestions, we have removed several 'significant' words that lacked practical significance. Simultaneously, we have retained certain 'significant' where meaningful comparisons were made with previously mentioned groups. Additionally, we have addressed the issue of missing comparison groups by providing groups to clarify the context of 'significant'.

Thank you for your valuable feedback once again.

Revision made:

From the results in **Fig. 3e** and higher magnification images in **Supplementary Fig.9**, compared with MH-GDz/Ca NPs group, the H-GDz/Ca NPs group exhibited 2.95 times (**Supplementary Fig.10**) higher efficient GLUT-1 detection, suggesting great cell internalization of H-GDz/Ca NPs. In contrast, the cellular uptake of free DNA sequences was significantly diminished comparing with NPs groups, due to their high negative charge, as less GLUT-1 signal can be detected. However, remarkably, even in the presence of the highly negative charge, we observed a higher

For the purpose of validating the robustness of the aforementioned findings, we conducted a replicate verification utilizing an additional HER-2 positive gastric cancer cell line SNU216. The results similarly confirm that H-GDz/Ca materials, when possessing the correct HER-2 sequence, exhibit significant internalization in SNU216 cells (**Supplementary Fig. 11a,b**), comparing with the MH-GDz/Ca group. Furthermore, the internalization of MNF materials is n.....

8. The discussion section needs to be expanded to explain the main results and how they form the final take home messages, including the comparison with other established literature. Consider moving some discussion contents from result to discussion part (ex. Pg 8, Line 10-14, Pg. 27, Line 15-17).

Thank you for your feedback. We acknowledge the need to expand the discussion section to provide a comprehensive explanation of the main results and their implications, including comparisons with established literature. To address this, we incorporated relevant discussion content from the results section, specifically from Page 8, Line 10-14, and Page 27, Line 15-17, into the discussion part. Thank you for guiding us in enhancing the clarity and depth of our discussion

Revision made:

Discussion

The key advantage of MNFs over MOFs lies in the multifunctionality of the DNA sequences compared to the organic ligands. DNA sequences within MNFs can act as nucleic acid drugs with therapeutic functions, construct adapter sequences for targeted cell binding, and form molecular fluorescence beacons for disease gene diagnosis. Therefore, utilizing MNF materials as carriers for drug delivery not only leverages the functionality of the drug itself but also harnesses the multifunctionality of the DNA sequences simultaneously. However, it is imperative to note that MNFs may suffer from certain

limitations, including stability concerns compared to MOFs and challenges associated with achieving rapid synthesis at room temperature.

Our work has revealed a pivotal breakthrough in the synthesis of MNF systems by elucidating the crucial role of DNA sequence length, and also unlocked unprecedented avenues for loading large-molecule active drugs into these advanced nanomaterials. The observed length related results can be attributed to the inherent characteristics of oligonucleotides, which are known to lack strong base stacking forces and exhibit flexible spatial conformations¹⁸. Consequently, increasing the length of oligonucleotide sequences (which is different from augmenting the nucleotide concentration), holds the potential to enhance the folding and entanglement probability of the sequences during the reaction process^{39, 40}.

.....From a therapeutic perspective, the current scientific landscape is rich with endeavors to enhance ROS production within the tumor microenvironment for robust anticancer effects. In alignment with this research enthusiasm, our simplified MNF material presents distinctive advantages for ROS-induced tumor therapy. First, the MNF boasts a carrier-free therapeutic architecture, integrating therapeutic DNA functional sequences with mitochondrial-disrupting metal ions, thus avoiding the introduction of non-therapeutic components. Second, leveraging the innate targeting proficiency of DNA nanotechnology, MNF requires no exogenous cellular targeting modifications, streamlining the therapeutic approach. Third, protein-loading capability of MNF facilitates the IRF-1 loading and inhibition of nuclear repair proteins RAD51, leading to a significant enhancement of ROS-mediated nuclear DNA fragmentation and thereby amplifying the ROS-driven therapeutic cascade. Besides, we found the MNF metabolism and clearance pathway is kidney based, which could be attributed to the inherent renal interception of negatively charged nanomaterials or negatively charged oligonucleotides.....

Reviewer #2 (Remarks to the Author):

The authors have satisfactorily responded to my concerns. The revised manuscript has been strengthened and should appeal to a broader audience as well.

Thank you for your positive feedback. We are pleased that our revisions have addressed your concerns and strengthened the manuscript. It is our aim to ensure that our research is accessible and appealing to a broad audience, and we are glad to hear that you feel the revised manuscript achieves this. We appreciate your valuable input and support throughout the review process.

Reviewer #4 (Remarks to the Author):

In this study, the authors reported an interferon regulatory factor-1 (IRF-1) loaded Ca²⁺/(aptamer-deoxyribozyme) MNF to target regulate glucose transporter (GLUT-1) expression in human epidermal growth factor receptor-2 (Her-2) positive gastric cancer cells, which could disrupt GSH/ROS homeostasis, suppress DNA repair, and augment ROS-mediated DNA damage therapy. The authors' work is quite good, but there are still some issues that need further explanation.

Thank you for your insightful comments on our study. We appreciate your acknowledgment of our work and its potential significance. We understand that there are areas that require further explanation, and we are committed to addressing them thoroughly. We will carefully review your feedback and provide additional clarification where necessary to ensure the clarity and completeness of our findings.

1. It is recommended to supplement the information by indicating the age of the mice at the time of euthanasia in Fig.4g and Supplementary Fig22.

Thank you for your suggestion. We supplemented the information in Figure 4g and Supplementary Figure 22 by indicating the age of the mice at the time of euthanasia.

Revision made:

Gastric Cancer Orthotopic model

..... Humane endpoints include conditions such as tumor burden exceeding 10% of normal body weight, weight loss in animals exceeding 20% of their normal weight, and persistent self-harm by the animals. When humane endpoints are reached (when mice around 13-15 weeks in this study), euthanasia is performed through cervical dislocation under deep anesthesia.

2. It is suggested to specify the name of the cell line used in Fig4d-i.

Thank you for your suggestion. We specify the name of the cell line used in Figures 4d-i to provide clarity to the readers.

We mentioned the cell line used in methods:

First, a subcutaneous tumor model was established by injecting 1×10^6 **N87 gastric cancer cells labeled with Luciferase/GFP**, into the BALBc-nu strain (Female, 6–8 weeks old).

3. It is recommended to supplement the effects of using the IRF-1 group alone on gastric cancer growth and GSH/ROS homeostasis to illustrate the superiority of the IRF/H-GDz/Ca⁺ nanodevice.

Thank you for your valuable suggestion. We understand the importance of providing a comprehensive comparison to highlight the efficacy of our proposed nanodevice. However, the main objective of this work is to verify whether the functionality of the oligonucleotide sequences forming MNF remains after MNF formation, as well as their capability to load protein drugs. The existing groups have already sufficiently demonstrated the functionality of nucleic acids. Furthermore, by comparing the group loaded with IRF-1 with the group not loaded with IRF-1, we have also demonstrated the effective delivery of IRF-1. Therefore, introducing an additional IRF-1 group would not emphasize our core objective.

Of course, we respect the reviewer's suggestion, and indeed, introducing the IRF-1 group would highlight the material's delivery efficacy for proteins. Meanwhile, standalone IRF-1 protein cannot achieve tumor-targeted delivery and is susceptible to enzymatic degradation in the bloodstream, which would definitely highlight the advantages of MNF materials. We will consider introducing a control group with blank protein drugs in future MNF studies to fully explore the delivery advantages of MNF for proteins. We thank you for your kind suggestions.

4. The Western blot bands in Supplementary Fig11c are obscured; Please make a revision.

Thank you for bringing this to our attention. We apologize for any inconvenience caused by the obscured Western blot bands in Supplementary Figure 11c.

SNU216 cell line

5. The text is in 4d obscured; Please make a revision.

Regarding your comment about the text being obscured in 4d, we apologize for any confusion or difficulty in understanding. We carefully reviewed this section.